# Single-cell DNA methylome and 3D multi-omic atlas of the adult mouse brain

Hanqing Liu[1,14], Qiurui Zeng[1,2,14], Jingtian Zhou[1,3], Anna Bartlett[1], Bang-An Wang[1], Peter Berube[1,2], Wei Tian[1], Mia Kenworthy[1], Jordan Altshul[1], Joseph R. Nery[1], Huaming Chen[1], Rosa G. Castanon[1], Songpeng Zu[4], Yang Eric Li[4], Jacinta Lucero[5], Julia K. Osteen[5], Antonio Pinto-Duarte[5], Jasper Lee[5], Jon Rink[5], Silvia Cho[5], Nora Emerson[5], Michael Nunn[1], Carolyn O'Connor[6], Zhanghao Wu[7], Ion Stoica[7], Zizhen Yao[8], Kimberly A. Smith[8], Bosiljka Tasic[8], Chongyuan Luo[9], Jesse R. Dixon[10], Hongkui Zeng[8], Bing Ren[4,11,12], M. Margarita Behrens[5] & Joseph R. Ecker[1,13 ✉]

Cytosine DNA methylation is essential in brain development and is implicated in various neurological disorders. Understanding DNA methylation diversity across the entire brain in a spatial context is fundamental for a complete molecular atlas of brain cell types and their gene regulatory landscapes. Here we used single-nucleus methylome sequencing (snmC-seq3) and multi-omic sequencing (snm3C-seq)[1] technologies to generate 301,626 methylomes and 176,003 chromatin conformation–methylome joint profiles from 117 dissected regions throughout the adult mouse brain. Using iterative clustering and integrating with companion whole-brain transcriptome and chromatin accessibility datasets, we constructed a methylation-based cell taxonomy with 4,673 cell groups and 274 cross-modality-annotated subclasses. We identified 2.6 million differentially methylated regions across the genome that represent potential gene regulation elements. Notably, we observed spatial cytosine methylation patterns on both genes and regulatory elements in cell types within and across brain regions. Brain-wide spatial transcriptomics data validated the association of spatial epigenetic diversity with transcription and improved the anatomical mapping of our epigenetic datasets. Furthermore, chromatin conformation diversities occurred in important neuronal genes and were highly associated with DNA methylation and transcription changes. Brain-wide cell-type comparisons enabled the construction of regulatory networks that incorporate transcription factors, regulatory elements and their potential downstream gene targets. Finally, intragenic DNA methylation and chromatin conformation patterns predicted alternative gene isoform expression observed in a whole-brain SMART-seq[2] dataset. Our study establishes a brain-wide, single-cell DNA methylome and 3D multi-omic atlas and provides a valuable resource for comprehending the cellular–spatial and regulatory genome diversity of the mouse brain.

The mouse brain is a complex organ comprising millions of cells that form diverse anatomical structures and cell types[3–8]. Advances in single-cell transcriptome and epigenome technologies are revealing the intricate molecular diversity of the mammalian brain, which in turn are offering insights into epigenetic mechanisms central to orchestrating this biological diversity[9–13].

Cytosine DNA methylation (5mC), a covalent genome modification in post-mitotic cells throughout their lifespan[14], is associated with

neuronal function, behaviour and various diseases[15]. Although 5mC predominantly occurs at CpG sites (mCG) in mammalian genomes, non-CpG cytosine methylation (mCH, where H can be A, C or T) is also prevalent in neurons[14,16]. CpG and CpH methylation modulate transcription factor (TF) binding and gene transcription through dynamic occurrence at regulatory elements and gene bodies[17]. Both types of methylation directly influence the DNA binding of methyl-CpG binding protein 2 (MeCP2)[18–21], a crucial 5mC reader and the cause of Rett

[1]Genomic Analysis Laboratory, The Salk Institute for Biological Studies, La Jolla, CA, USA. [2]Division of Biological Sciences, University of California, San Diego, La Jolla, CA, USA. [3]Bioinformatics and Systems Biology Program, University of California, San Diego, La Jolla, CA, USA. [4]Department of Cellular and Molecular Medicine, University of California, San Diego School of Medicine, La Jolla, CA, USA. [5]Computational Neurobiology Laboratory, The Salk Institute for Biological Studies, La Jolla, CA, USA. [6]Flow Cytometry Core Facility, The Salk Institute for Biological Studies, La Jolla, CA, USA. [7]Sky Computing Lab, University of California, Berkeley, Berkeley, CA, USA. [8]Allen Institute for Brain Science, Seattle, WA, USA. [9]Department of Human Genetics, University of California, Los Angeles, Los Angeles, CA, USA. [10]Peptide Biology Laboratory, The Salk Institute for Biological Studies, La Jolla, CA, USA. [11]Center for Epigenomics, University of California, San Diego School of Medicine, La Jolla, CA, USA. [12]Institute of Genomic Medicine, University of California, San Diego School of Medicine, La Jolla, CA, USA. [13]Howard Hughes Medical Institute, The Salk Institute for Biological Studies, La Jolla, CA, USA. [14]These authors contributed equally: Hanqing Liu, Qiurui Zeng. ✉e-mail: ecker@salk.edu

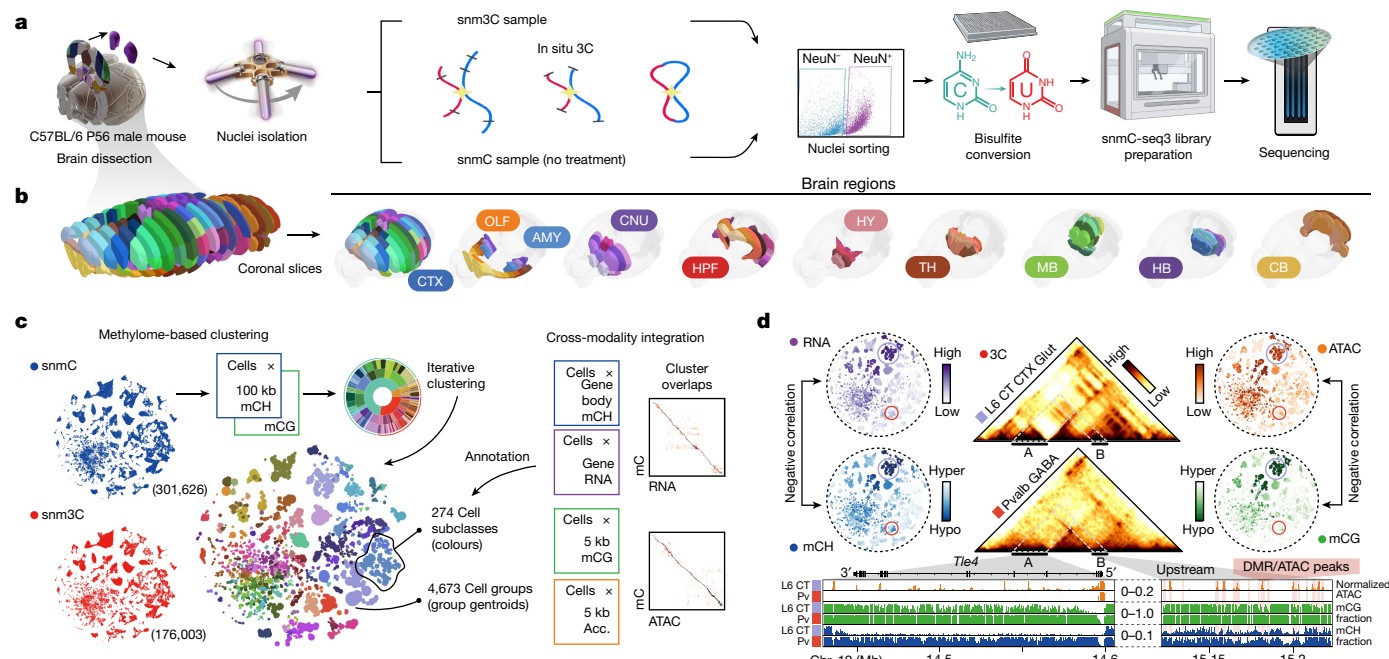

**Fig. 1 | Single-cell DNA methylome and multi-omic atlas chart the cellular and genomic diversity of the whole mouse brain. a**, The workflow of dissection, nuclei and library preparation for snmC-seq3 and snm3C-seq. P56, postnatal day 56. **b**, The 117 dissected regions from 18 coronal slices (600-μm thick) were grouped into 10 major brain regions (see Supplementary Table 10 for abbreviations). Each dissection region is registered to the 3D CCF[3]. **c**, The cell atlas: methylome-based iterative clustering of snmC and snm3C datasets. Left, *t*-distributed stochastic neighbour embedding (*t*-SNE) plot coloured by modality. Middle, plot aggregated into 4,673 cell group centroids and coloured

by 274 cell subclasses. Right, cross-modality integration of brain-wide datasets from BICCN, details in Fig. 2. RNA data from ref. 6. ATAC data from ref. 11. Acc., accessibility. **d**, The genome atlas: the *Tle4* gene exemplifies pseudo-bulk profiles of five modalities across the whole brain, with genome browser view of the 'L6 CT CTX Glut' and 'Pvalb GABA' subclasses in the bottom. Interactive browser available at tinyurl.com/fig1d. Schematic in **a** created using BioRender (www.biorender.com). Brain atlas images in **b** were created based on ref. 3 and the Allen Brain Reference Atlas (atlas.brain-map.org), © 2017 Allen Institute for Brain Science.

syndrome[22]. Genome-wide differential methylation analyses have predicted millions of regulatory elements and have produced a cellular taxonomy and a base-resolution genome atlas[9,23].

Furthermore, *cis*-regulatory elements in complex mammalian genomes can operate over long distances to regulate target genes[24]. Understanding the relationships between the physical contact frequency of enhancers and promoters and their collective impact on gene body epigenetics and transcriptomic status is crucial for decoding the molecular diversity of the mammalian brain. Our previous work used single-nucleus methylome (snm) and chromatin conformation capture (3C) sequencing (snm3C-seq) to concurrently examine these aspects[1]. However, a detailed brain-wide map of chromatin conformation remains to be charted.

In this study, we used enhanced single-nucleus methylation sequencing (snmC-seq3) and snm3C-seq technologies to analyse DNA methylomes and the 3D genome in detail[25,26]. We collected 301,626 methylomes and 176,003 m3C joint profiles from the entire mouse brain to produce a dataset comprising 786 billion methylation reads (snmC-seq3 plus snm3C-seq) and 33 billion *cis*-long-range chromatin contacts (snm3C-seq). This rich dataset identifies 4,673 cell groups, which aligned well with other BRAIN Initiative Cell Census Network (BICCN) data[6,11]. The methylome clusters were annotated using the nomenclature from companion transcriptomic studies[6], thereby offering a comprehensive multi-omic resource.

Our analysis underscore the spatial information in the epigenome, which was validated using a multiplexed error-robust fluorescence in situ hybridization (MERFISH)[27] dataset created using genes that showed distinct gene-body methylation patterns across brain regions. We also explore the regulatory landscapes of individual genes by examining thousands of aggregated epigenetic profiles. Notable connections

emerge between chromatin conformation diversity and gene-body methylation profiles across multiple genome scales. Intersecting this epigenetic dataset with a correlation-based analysis, we construct gene regulatory networks (GRNs) that connect TFs, differentially methylated regions (DMRs) and potential target genes. Finally, integration with a whole-brain full-length SMART-seq dataset[6] illuminates the interplay between epigenetic profiles and transcriptional dynamics within long neuronal genes.

To facilitate access to this resource, we introduce the mouse brain cellular and genomic browser (mousebrain.salk.edu), a user-friendly platform for data query and visualization. By unveiling the multifaceted complexities of the molecular architecture of the mouse brain, our study deepens insights into the epigenetic and transcriptomic intricacies that underpin brain function and diseases.

## The methylome and 3D genome atlas

We developed snmC-seq3, an optimized single-nucleus methylome sequencing method (Supplementary Methods), to profile genome-wide 5mC at base resolution (Fig. 1a) across 117 dissected regions in the whole brain from adult male C57BL/6 mice (Fig. 1b, Extended Data Fig. 1a and Supplementary Table 1). We also used snm3C-seq, a multi-omic technology[1], to jointly profile the DNA methylome and chromatin conformation from 33 dissected regions (Extended Data Fig. 1b), which added the 3D genome context across all brain cell types (Fig. 1a). Each dissected region is represented by two to three replicates, which were obtained from pooling the same region from at least six animals. Single nuclei were captured using fluorescence-activated nuclei sorting (FANS), which enriched for neurons that were positively labelled with a NeuN antibody (NeuN[+] neurons constituted 92% of snmC and

78% of snm3C data, with the remaining data being NeuN⁻ neurons or non-neurons; Methods). Collectively, we obtained 324,687 (301,626 passed quality control (QC)) DNA methylome profiles, including 102,783 nuclei from previous research[9]. On average, the snmC-seq dataset had 1.44 ± 0.50 million (mean ± s.d.) final reads that covered 72 ± 24 million (6.5% ± 2.2%) cytosine bases in the mouse genome. We also obtained 196,172 (176,003 passed QC) joint methylome and 3C profiles, with each cell having 1.99 ± 0.57 million final reads that covered 72 ± 20 million (6.5% ± 1.8%) cytosine bases. The 3C modality of each cell had 188,000 ± 81,000 (18.3% ± 5.7% of the total fragments) *cis*-long-range contacts and 108,000 ± 41,000 (10.4% ± 2.3%) *trans*-contacts (Extended Data Fig. 2, Methods and Supplementary Tables 2 and 3).

After QC and preprocessing, we analysed the data in cellular and genomic contexts (Fig. 1c,d). During the cellular analysis, we conducted iterative clustering of the mCH and mCG profiles in 100-kb bins throughout the genome to establish a methylome-based whole-brain cell-type taxonomy. At the highest level of granularity, we obtained a total of 4,673 cell groups. To validate and annotate the dataset, we integrated the methylome data with other brain-wide chromatin accessibility[11] and transcriptome datasets[6], which resulted in cluster-level mapping across modalities and annotations of these clusters into 30 class labels and 274 subclass labels shared with a companion transcriptome study[6] (Supplementary Table 4 and see below).

On the basis of the clustering and integrative annotations, we produced pseudo-bulk profiles of five modalities (mCH/mCG fraction, chromatin conformation, accessibility and gene expression) for each cell group, thereby providing a cell-type-specific, multi-omic atlas for the mouse genome (Fig. 1d). With more details covered in subsequent sections, we use the TLE family member 4 (*Tle4*) gene, a marker for the 'L6 CT CTX Glut' subclass, as an example to illustrate the power of this comprehensive dataset (details in Supplementary Note 1).

Overall, our study utilizes brain-wide single-cell mC and m3C datasets to achieve the following aims: (1) define cellular taxonomy based on the DNA methylome; (2) integrate with other atlas-level datasets from the BICCN; and (3) generate a multi-omic cell-type-specific genome atlas for the mouse brain. This resource enabled us to conduct several detailed analyses and make various discoveries, as we described below.

## Methylome-based cell-type taxonomy

Following QC and preprocessing (Methods), we used iterative clustering to classify methylome-based cell populations in the snmC-seq and snm3C-seq datasets, utilizing mCH and mCG profiles in 100-kb bins across the genome[9,25]. In the final iteration, we identified 2,573 clusters and further separated them on the basis of brain dissection regions into 4,673 cluster-by-spatial groups, which served as the finest granularity level for subsequent analyses (Fig. 2a and Extended Data Fig. 3). To establish a hierarchical structure for whole-brain cell types and to support multi-omic data analyses, we iteratively integrated the methylome datasets with a companion brain-wide single-cell transcriptome dataset (see next section). Following integration, we annotated the mC-based cell groups in agreement with 30 transcriptome-based classes and 274 subclasses[6] (Supplementary Table 4). The subsequent analyses relied on the cell-group and subclass levels of cell classifications (Fig. 2a and Extended Data Fig. 3a,d).

We organized our dissections into ten major brain regions (Fig. 2b,c and Extended Data Fig. 4) according to their specific cell-type composition and neuronal functionality (Fig. 2d) as follows: the isocortex (CTX); olfactory areas (OLF; including the olfactory bulb and piriform cortex); amygdala areas (AMY; including the cortical subplate (CTXsp) and the striatum-like amygdala nuclei (sAMY)); cerebral nuclei (CNU; including the striatum and pallidum, but excluding the sAMY); the hippocampal formation (HPF; including the hippocampus and parahippocampal cortex); the thalamus (TH); the hypothalamus (HY); the midbrain (MB);

the hindbrain (HB; including the pons and the medulla); and the cerebellum (CB). Most neuronal subclasses (218) were each derived from a single major region. Eighteen neuronal subclasses were situated across two adjacent regions, which could be due to imprecise dissections but may also represent neuronal types shared between neighbouring brain regions (Supplementary Table 5). In addition, marked cellular diversity was observed in non-telencephalic regions (the TH, the HY, the MB and the HB; Fig 2d,e and Extended Data Fig. 4), which is a common feature observed in other single-cell brain atlases that investigate various molecular modalities[6–8,11]. Notably, the global methylation level substantially changed across cells and dissection regions (Extended Data Fig. 3g,h), with subcortical neuronal subclasses exhibiting markedly increased mCH levels compared with cortical excitatory neurons (Extended Data Fig. 3i,j and Supplementary Note 2).

Last, our dataset extensively profiled non-neuronal cells and adult immature neurons (IMNs) throughout the brain (Extended Data Fig. 4 and Supplementary Tables 4 and 5). Consistent with other modalities[6,11], we detected spatial differences in astrocyte methylomes, particularly between telencephalic and non-telencephalic regions. Initially, IMNs clustered with astrocytes, but later iterations resolved one population in the subgranular zone of the dentate gyrus and another population in areas overlapping the rostral migratory stream[28]. Furthermore, the oligodendrocyte lineage demonstrated spatial distinctions between telencephalic and non-telencephalic regions at the cluster level (Extended Data Fig. 4). Our dataset also encompasses other immune and vascular cell types, including microglia, pericytes, endothelial cells, arachnoid barrier cells and vascular leptomeningeal cells.

## Consensus cell-type taxonomy

Developing a brain cell-type taxonomy requires integrating various molecular modalities, verifying cell clusters on the basis of multiple molecular information and applying a uniform nomenclature[29]. We began this endeavour by performing an integrative analysis with a brain-wide transcriptome dataset from the BICCN consortium[6]. After strict QC, this single-cell RNA sequencing (scRNA-seq) dataset established a cell taxonomy that categorized 4.3 million cells into 5,200 cell clusters, 1,045 supertypes, 306 subclasses, 32 classes and 7 divisions. Various aspects were incorporated into the cluster annotation, including spatial distribution[6,7], neurotransmitter identity, marker genes and existing cell-type knowledge[29].

We used an efficient framework (adapted from the Seurat package[30]; Methods) for iterative cross-modality integration to leverage this substantial effort. The initial integration effectively matched neuronal spatial distribution and high-level annotations (Fig. 2e), whereas subsequent iterations refined cluster matching within subclasses to greater detail (Fig. 2f). We utilized integration overlap scores (Methods) to map methylome cell groups to transcriptome clusters and to annotate methylome datasets into subclasses using consistent nomenclature (Supplementary Table 4). In summary, we matched all methylome cell groups with 4,669 (90%) transcriptomic clusters, which encompassed 4.19 million (97.4%) cells corresponding to 274 subclasses (Fig. 2f). The 531 unassigned transcriptomic clusters represented only 2.6% of cells, which were primarily rare populations (<0.03% of the total RNA dataset) that were insufficiently represented in the methylome dataset. We calculated the transcriptome profile for each cell group on the basis of these integration results (Methods).

The overlap score for the final iteration within each subclass revealed a high-granularity correspondence between methylome and transcriptome clusters (Fig. 2f, boxes). We further examined vital neural functional genes to demonstrate this accurate match between mC and RNA levels (Extended Data Fig. 5). Overall, this high-resolution cross-modality integration offers multi-omic evidence for identifying thousands of cell clusters in the adult mouse brain, and lays the groundwork for subsequent genomic and epigenomic analyses.

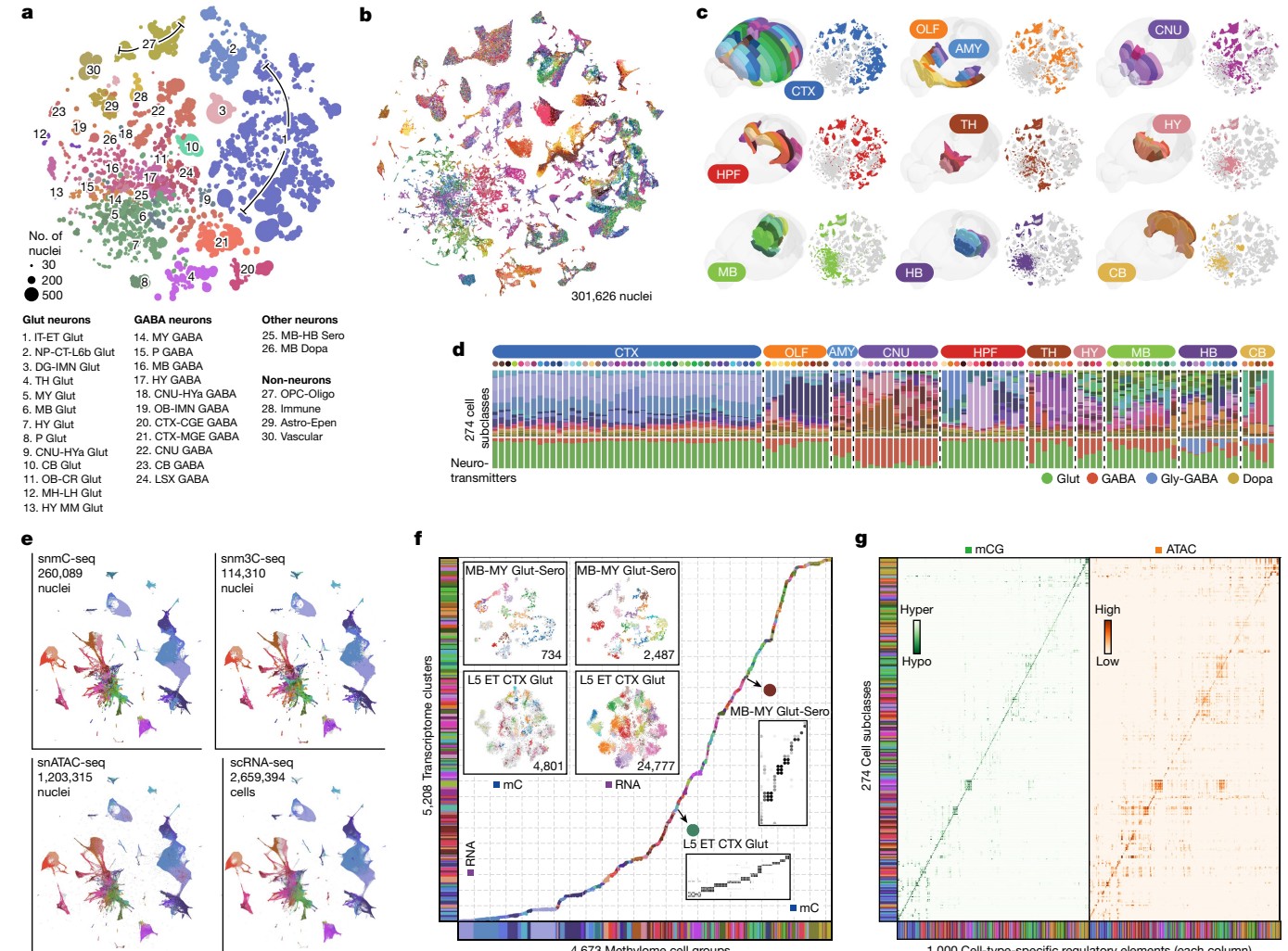

**Fig. 2 | Consensus cell-type taxonomy across molecular modalities.**
**a**, Cell-group-centroid *t*-SNE colour by cell class (*n* = 4,673). **b**, Cell-level *t*-SNE colour by 117 dissection regions. **c**, 3D CCF registration and cell *t*-SNE of each major region. **d**, Cell subclass (top row) and neurotransmitter composition (bottom row) of each brain dissection region (each upper dot) grouped by major region. Other neurotransmitters are not shown in the plot, but the information is provided in Supplementary Table 2. **e**, Integration *t*-SNE of all neurons from the snmC-seq, snm3C-seq, snATAC-seq and scRNA-seq datasets, coloured by matched cell subclasses. For each plot, the light grey cells in the background represent cells from the other three modalities. RNA data from ref. 6. ATAC data from ref. 11. **f**, Brain-wide cluster map between the snmC-seq and scRNA-seq datasets (Supplementary Table 4) based on iterative integration.

Each dot, coloured by subclasses, on the diagonal represents a link between the mC clusters (*x* axis) and RNA clusters (*y* axis). Two examples in floating panels demonstrate highly granular correspondence of cell clusters in the final integration round: integration *t*-SNE of 'MB-MY Glut-Sero' and 'L5 ET CTX Glut' cells from mC and RNA coloured by intramodality clusters and confusion matrix of overlap score between the intramodality clusters (see Extended Data Fig. 5 for more gene details). **g**, Dot plots of mCG fraction (left) and chromatin accessibility (right) of cell-type-specific CG-DMRs (columns) in each cell subclass (row). The colour of each dot represents an aggregated epigenetic profile of 1,000 DMRs in a cell subclass; deeper colour indicates that these DMRs are more hypomethylated or accessible in a subclass. See Extended Data Fig. 6 for more mC–ATAC integration details.

## Cell-type-specific regulatory elements

Having established a consensus cell taxonomy across the entire mouse brain, we further identified 2.56 million non-overlapping CpG DMRs between the subclasses of the whole brain or the clusters of each major brain region (Methods). These DMRs involved 44% of the total CpG sites in the genome, with an average length of 189 ± 356 bp (mean ± s.d.) and containing 3.9 ± 6.0 CpG pairs (each containing 2 bases). The CpG DMRs provide predictions about cell-type-specific *cis*-regulatory elements, and hypomethylation in the DMR region usually indicates the active regulatory status in adult brain tissue[9,23] (Fig. 2g). To annotate the accessibility status of the DMRs in a systematic manner, we performed iterative integration between the methylome and chromatin accessibility dataset from the BICCN[11], using non-overlapping chromosome 5-kb bins (Methods). This dataset, generated using single-nucleus assay for

transposase-accessible chromatin with sequencing (snATAC–seq) without NeuN enrichment by FANS, contains 1,372,646 neurons and 939,760 non-neuronal cells. As this dataset shares the same dissection samples with the snmC-seq dataset, we used this metadata information to assess the integration alignment score[30] between neurons analysed using mC and ATAC. Notably, the dissected regions were precisely aligned (score of 0.89 ± 0.11), which indicated extensive concordance in the cellular diversity of both epigenomic modalities (Extended Data Fig. 6a,b). After integration, we also calculated the chromatin accessibility profile for each cell group using their matched ATAC-analysed cells. The resulting mCG fractions and chromatin accessibility levels at DMR regions showed similar cell-type-specificity across brain cell subclasses. This result confirmed the correct match of cell-type identities (Fig. 2g and Extended Data Fig. 6c, d). By integrating the mC and ATAC datasets, we achieved high concordance in cellular diversity across both epigenomic

modalities, which further validates the accuracy of our approach in determining cell-type-specific regulatory elements and their activities.

## Spatial epigenomic diversity

Tens of millions of cells in the mouse brain accurately form complex anatomical structures that are controlled by their diverse gene expression and epigenetic regulation. Our clustering analysis demonstrated cell-type composition differences across brain regions (Fig. 2). To further explore the spatial information in the DNA methylome, we performed differentially methylated gene (DMG) and DMR analyses across anterior-to-posterior, dorsal-to-ventral and medial-to-lateral axes in the brain using representative dissection regions (Extended Data Fig. 7a–c). In all three axes, we identified hundreds of thousands of DMGs related to various neuronal functions and DMRs associated with these genes. This result highlights the marked spatial diversity encoded in the methylome.

To increase the spatial resolution of the analysis and to investigate whether the observed methylation spatial pattern corresponds to actual transcriptomic diversity, we used MERFISH technology, which enables in situ profiling of the expression level of hundreds of genes in brain sections[7,27,31]. We designed a 500-gene panel (Supplementary Table 6) selected on the basis of cell type and spatial diversity in gene-body hypomethylation across the brain (Methods). We then profiled six coronal sections corresponding to our mC and m3C brain slices (Extended Data Fig. 7d). After QC, we obtained 266,903 MERFISH cells and annotated their cell subclasses by integrating with the scRNA-seq dataset[6] (Extended Data Fig. 7e,f and Supplementary Table 7). We then performed cross-modality integration between the neurons in the methylome and MERFISH datasets, imputing the spatial location of each methylation nucleus (Fig. 3a and Supplementary Table 8). Notably, the predicted spatial coordinates of the methylation nuclei closely matched the dissected regions (Fig. 3b). For example, glutamatergic cells showed arealization[32] among cortical areas within each slice and dorsal–ventral separation was observed among medium spiny neurons dissected from the caudoputamen (CP) and nucleus accumbens (ACB) regions. Moreover, many subcortical dissection borders were faithfully preserved in the imputed spatial embedding. The spatial location imputation also assigned many cell subclasses to fine anatomical structures, which were considerably smaller than our dissection regions (Fig. 3c and Extended Data Fig. 7g). For instance, laminar layer information was mapped among cortical excitatory cells (for example, 'L2/3 IT CTX Glut', 'L5 ET CTX Glut' and 'L2/3 IT ENTI-PIR Glut'). In addition, many subcortical neurons were allocated to specific brain nuclei (for example, 'STN-PSTN Pitx2 Glut' and 'ZI Pax6 GABA'), which highlights the correspondence between the cell-subclass identity and anatomical structure in these areas.

The high spatial resolution in the imputation was attributed to the strong association between cell location and DNA methylation of crucial genes and regulatory elements. For example, the *Elavl2* gene, which encodes a RNA-binding protein involved in post-transcriptional regulation functions in neurons[33], exhibited a dorsal–ventral increased expression pattern in subcortical neurons in slice 10, which was also observed as a decrease in gene-body mCH methylation of *Elavl2* and a nearby mCG methylation of a DMR (Fig. 3d). Notably, the chromatin interactions between the DMR and *Elavl2* showed stronger contacts in regions where *Elavl2* was highly expressed. Likewise, *Rasgrf2*, which encodes a guanosine nucleotide exchange factor for Ras GTPases, displayed differential expression and methylation across cortical layers. DMRs near *Rasgrf2* were highly correlated, with chromatin conformation data supporting physical proximity when both the DMR and *Rasgrf2* were active (Fig. 3e). *Negr1* also showed similar correspondence among modalities in cortical dissected regions (Extended Data Fig. 7h). These findings demonstrate a clear spatial pattern in DNA methylation that aligns with the spatial transcriptome, which implies that epigenetic

regulation exerts precise control over the cellular spatial location. To extend this spatial annotation to the entire brain, we comprehensively integrated the MERFISH dataset from a companion study[6] that contained 51 coronal slices and 3.9 million cells. The integration helped us to position each nucleus from the methylome dataset into a specific spatial location, which facilitated the interpretation of epigenetic profiles in brain-wide anatomical structures (Extended Data Fig. 8 and Supplementary Table 9).

## Chromosomal conformation dynamics

The annotated multi-omic datasets enabled us to leverage the cell-type diversity across the entire brain to understand the chromatin conformation landscape of individual genes. Here we systematically evaluated the variability of different 3D genome features (chromatin compartment, topologically associated domain (TAD) and highly variable interactions) across cell subclasses. We associated them with gene activity by correlating chromatin contact strengths with methylation fractions.

We initiated this effort by examining the chromatin compartment, a genome topology feature that brings together genomic regions tens to hundreds of megabases away[34]. The genomes are organized into two major compartments, A and B, corresponding to active chromatin and silent chromatin, respectively[34]. After calculating the compartment score of cell subclasses at the 100-kb resolution, we observed numerous A/B compartment switches in megabase-long regions (Fig. 4a). For instance, the chromosome 2 region spanning 3.5 million bases to 10.6 million bases exhibited a strong negative compartment score (B compartment) in mature oligodendrocytes ('Oligo NN'), but positive scores (A compartment) in cortical excitatory neurons such as 'L2/3 IT CTX Glut' (Fig. 4b). Notably, this compartment-switching region overlapped with *Celf2*, a gene that encodes a vital RNA-binding protein that modulates alternative splicing in neurons[35].

Given these observations, we sought to determine whether compartment switching correlated with DNA methylation changes within the same regions. After calculating the Pearson correlation coefficient (PCC) values across cell subclasses, we found a negative correlation between the compartment score and mCG or mCH fraction of 100-kb chromatin bins, with mCG exhibiting a stronger correlation than mCH (Extended Data Fig. 9a). We also observed that the compartment score of negatively correlated bins demonstrated greater variability across cell subclasses than the positively correlated bins (Fig. 4c and Extended Data Fig. 9b,c), which suggested that these negatively correlated bins exhibit wide activity change across a wide range of cell subclasses.

We then discovered that genes overlapping with the negatively correlated bins were enriched[36] in numerous neuronal functions, including nervous system development (Fig. 4d). To explore this aspect further, we examined another scRNA-seq atlas of mouse brain development[37] and found that the negatively correlated bins overlapped with genes that display a substantial increase in expression during prenatal brain development. By contrast, uncorrelated or positively correlated bins demonstrated no such trend (Extended Data Fig. 9d). These results suggest that large chromosomal conformation changes might be established during early development and subsequently maintain cellular specificity in adult brain nuclei[38].

## TAD and long gene boundary association

We also investigated TADs[39] and their boundaries at a 25-kb resolution (Methods). By first identifying boundaries in individual cells and subsequently using the domain boundary probability at the cell-subclass level, we were able to represent the strength of domain boundaries at each 25-kb bin (Extended Data Fig. 9e). To evaluate the variability of boundary probabilities across the genome, we performed a Chi-square test on each bin and identified 83,518 bins with significant variability across subclasses (false discovery rate (FDR) < 1 × 10$^{-3}$; Methods). For

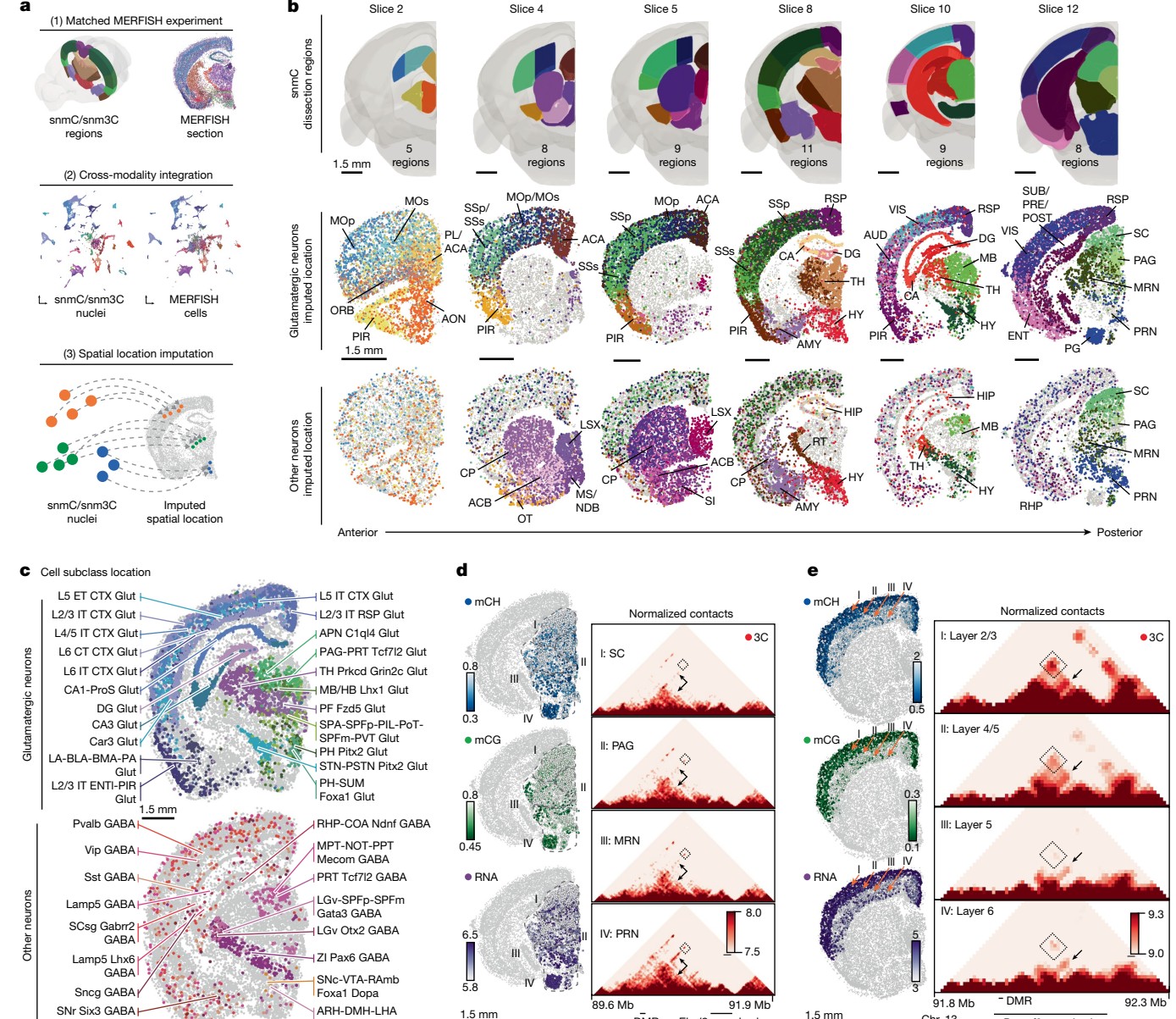

**Fig. 3 | Coherent spatial epigenomic and transcriptomic diversity in the brain. a**, Workflow of mC–MERFISH integration and spatial embedding of methylome cells. **b**, Spatially mapped methylation cell atlas. The first row displays CCF-registered brain dissection regions. The second and third rows show imputed spatial locations for glutamatergic and other neurons coloured by dissection regions. **c**, Spatial distribution of cell subclasses for glutamatergic neurons and other neurons on slice 10. **d**, Spatial epigenetic pattern of neuronal genes and their associated DMRs. The *Elval2* gene represents the spatial pattern among subcortical regions. The left column shows the gene-body mCH fraction, the DMR (chromosome 13: 91164342–91165792) mCG fraction and RNA expression. The right column displays a heatmap of normalized contacts between the DMR and the gene. **e**, The *Rasgrf2* gene and associated DMR (chromosome 13: 92027775–92028983) exhibit cortical layer differences in the same layout as **d**.

example, we observed that at the *Lingo2* locus—a 'L2/3 IT CTX Glut' hypomethylated gene linked to essential tremor and Parkinson's disease[40]—the TAD boundaries aligned with the transcription start site (TSS) and the transcription termination site (TTS) of the gene (Fig. 4e). Across all the neuronal subclasses, the boundary probability of the 25-kb bin at the *Lingo2* TSS exhibited a negative correlation with the transcript body mCH fraction (PCC = −0.65, FDR < 0.001, permutation-based test; Methods and Fig. 4f).

To generalize this observation, we calculated the average boundary probability at all gene TSSs and TTSs in the genome, separating them by transcript length (<100 kb as short, >100 kb as long[19]). Long genes displayed increased levels of boundary probability at the TSSs and TTSs (Fig. 4g), which suggested that TADs are more likely to form

around the gene body (that is, gene body domains). Our analysis then focused on the relationship between variable domain boundaries and gene bodies, particularly long genes (>100 kb) implicated in neuronal pathogenicity and potentially regulated by mCH and MeCP2 (ref. 19). We next calculated the PCC of gene transcript body mCH or mCG fractions with the boundary probabilities of all 25-kb bins within transcript ±2 Mb distances (Extended Data Fig. 9f). The top negatively correlated boundaries were predominantly located at the TSSs and TTSs of the corresponding gene transcripts (Fig. 4h and Extended Data Fig. 9f,g). We also observed a few significantly positively correlated boundaries to the transcript body mCH or mCG, although they lacked clear TSS–TTS colocalization (Extended Data Fig. 9f,g). Functional enrichment analysis[36] revealed that genes with strongly negatively correlated

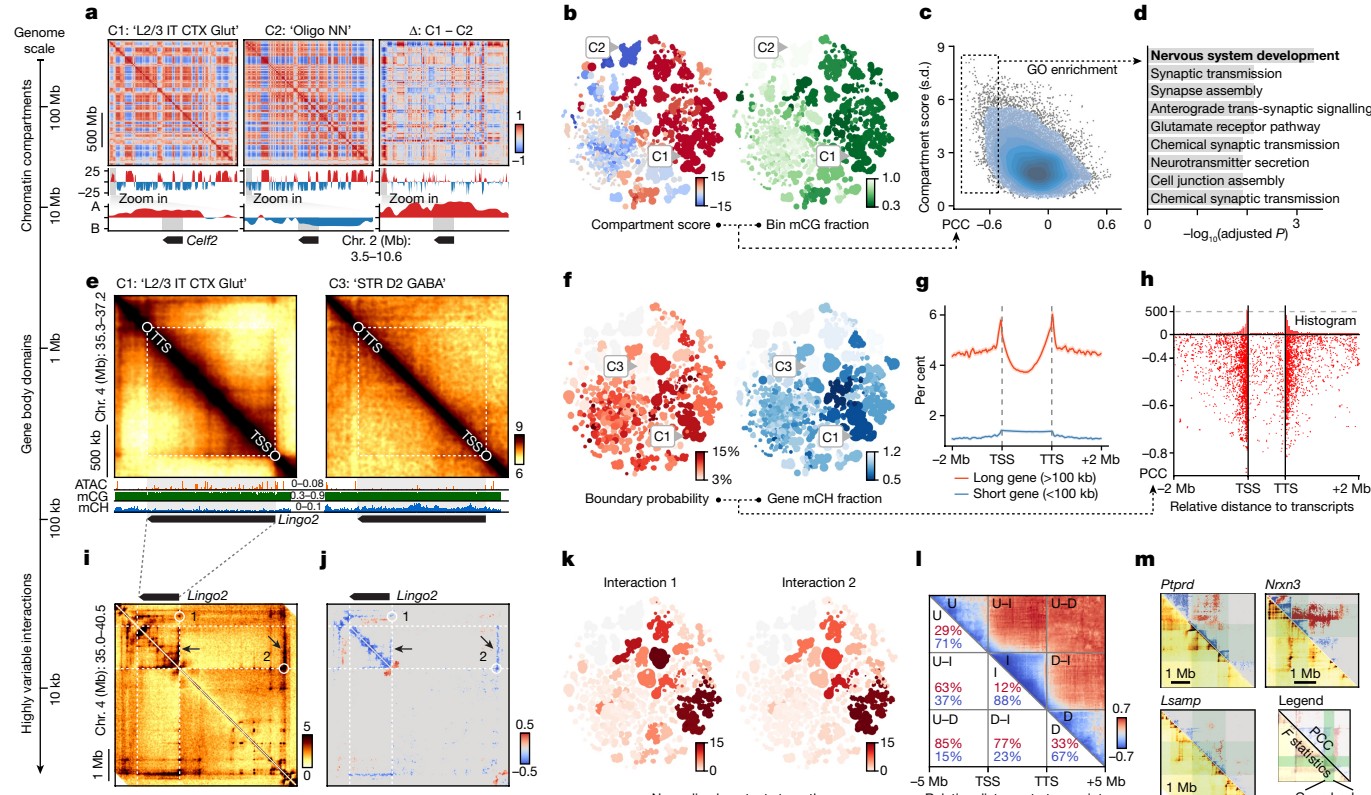

**Fig. 4 | Highly dynamic chromatin conformation features correlate with DNA methylation around neuronal genes. a**, Top, PCC chromatin conformation matrices for chromosome 2. Middle, compartment scores (red for A compartments, blue for B compartments). Bottom, zoom-in view of the *Celf2* locus. Columns represent 'L2/3 IT CTX Glut' (C1), 'Oligo NN' (C2) and Δ values (C1 − C2). **b**, Cell-group-centroid *t*-SNE for the bin chromosome 2 (6800000–6900000; *Celf2*) coloured by compartment score and the bin mCG fraction. **c**, Scatterplot of chrom100k bins, showing PCC values between compartment score and chrom100k mCG fraction (*x* axis) and compartment score s.d. values across cell subclasses (*y* axis). Blue contours indicate the kernel density of the dot. **d**, Functional enrichment for neuronal genes intersected with negatively correlated chrom100k bins (boxed in **c**). Adjusted *P* values obtained from one-side Fisher's exact test after FDR correction. **e**, Top, normalized chromatin contact matrices around *Lingo2* for C1 and 'STR D2 Gaba' (C3). Bottom, pseudo-bulk ATAC and methylome genome tracks. **f**, *t*-SNE coloured by the *Lingo2* TSS (bin: chromosome 4 (36950000–36975000)) boundary probability and mCH fraction. **g**, Mean boundary probabilities for

25-kb bins around long and short genes; error bands represent ±s.d. **h**, Scatterplot showing the location of the most negatively correlated boundary for each long gene transcript. The *y* axis is the PCC between the 25-kb bin boundary probabilities and transcript body mCH fractions; the *x* axis is the relative genome location to the transcripts. **i**,**j**, Heatmap indicates variance in contact strength across cell subclasses using *F* statistics from one-way ANOVA (**i**) and the PCC between the *Lingo2* mCH fraction and contact strength of highly variable interactions (**j**). White circles identify two loop-like, highly variable interactions. Arrows point to strips between interactions and gene bodies. **k**, *t*-SNE coloured by contact strengths of interactions 1 and 2 from **j**. **l**, Pileup view of the relative genome location of correlated interactions from all genes (using long genes (>100 kb)). The colours in the upper triangle are average PCCs. Abbreviations indicate intragenic (I), upstream (U) and downstream (D) and their combinations. **m**, Gene-specific chromatin landscape of megabase-long genes. Green marks gene bodies; the lower triangle shows *F* statistics as in **i**, and the upper triangle depicts PCC values similar to **j**.

gene-body domains were significantly enriched for crucial neuronal and synaptic functions. By contrast, positively correlated TAD boundaries were not associated with genes enriched for specific functions (Extended Data Fig. 9h). Together, these results indicate that TAD boundaries are closely associated with the TSS and TTS of long genes implicated in neuronal pathogenicity and pivotal functions.

## Diverse chromatin interaction landscapes

To thoroughly profile the chromatin conformation diversity at high resolution and to link genes to their potential regulatory elements, we analysed chromatin interactions at 10-kb resolution (Extended Data Fig. 10a). We first performed a one-way analysis of variance (ANOVA) across cell subclasses, using *F* statistics to summarize the variability of all interactions. Highly variable interactions corresponded to dot or strip-like patterns around genes (Fig. 4i).

Subsequently, we calculated the PCC values between transcript body mCH fraction and the contact strength of highly variable interactions within ±5 Mb of the transcript body (Fig. 4j). Highly variable and

gene-correlated interactions were assigned to a gene if any anchors of the interaction overlapped with the gene body. Through this assignment, the majority (95%) of gene-associated interactions were located within 1.2 Mb of the TSS of the gene (Extended Data Fig. 10b). Genes with numerous correlated interactions exhibited crucial neuronal and synaptic functions, overlapping with those genes that displayed a negatively correlated gene-body domain boundary as described in the previous section (Extended Data Fig. 10c,d). For instance, in the *Lingo2* locus, highly variable interactions were identified within the gene body, at gene body domain boundaries or corresponding to distal loop structures[41] (Fig. 4j, circles). The correlation analysis further stratified interactions as positively or negatively correlated with the methylation change of the gene. Notably, the correlated interaction anchors can be up to 1.6 Mb downstream (interaction 1) or 3.2 Mb upstream of the *Lingo2* TSS (interaction 2) while associating with strips along the entire gene body (Fig. 4j,k).

We then summarized the distribution of significantly correlated interactions surrounding all long genes by categorizing them into six groups on the basis of their relative location to the gene: intragenic,

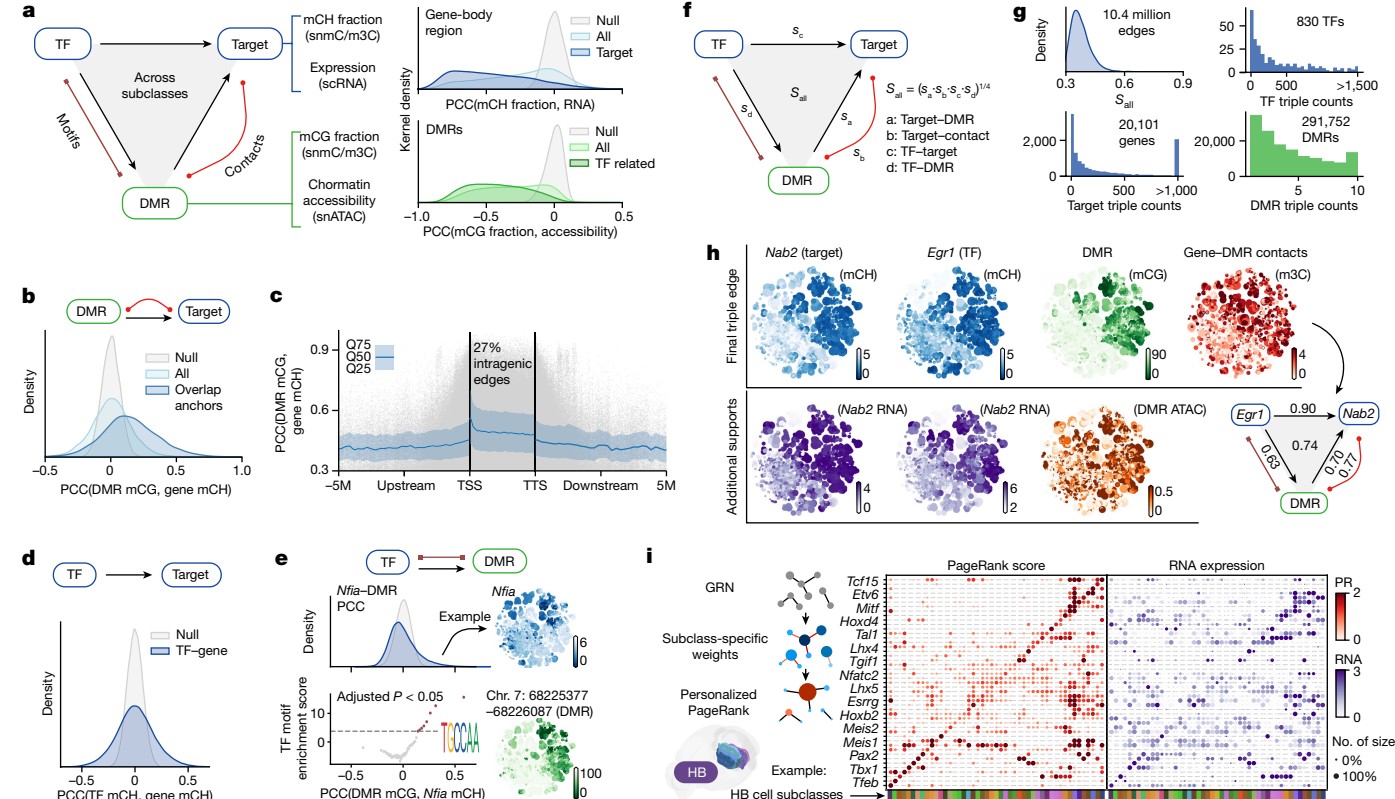

**Fig. 5 | GRNs predict binding elements, downstream targets and the cell-type importance of TFs. a**, Schematic depicting components of the GRN. The adjacent density plots show PCC values between gene mCH fractions and RNA expression levels; and between DMR mCG fractions and chromatin accessibilities. **b**, Density plot presents PCC values between DMR mCG and mCH fractions of the target gene. Grey indicates null distribution; light blue, all correlations; blue, correlations between DMRs overlapping with the correlated interaction anchors of the target gene. **c**, Scatterplot depicting PCC values between the DMR mCG and the target gene mCH and the relative location of the DMR in $1.2 \times 10^6$ DMR–gene edges. Grey dots represent DMR–target edges; blue line indicates the median PCC with the error band representing 25–75% quantile. **d**, Density plot showing PCC values between the mCH fraction of TFs and the target gene. **e**, Top, PCC between the *Nfia* mCH fraction and the DMR mCG fraction. Bottom, cisTarget[58] motif enrichment score in 50 DMR groups ordered and grouped by the *Nfia* DMR PCC value above. The example *t*-SNE plots are

coloured by the *Nfia* mCH fraction and mCG fraction of a positively correlated DMR. **f**, Schematic of the TF–DMR–target triple and the final score. **g**, Distribution of the final scores of all triples (from **f**) in the final network. Histograms show the number of triples that each TF, gene and DMR is involved in. **h**, Example triple comprising *Egr1* (TF), *Nab2* (target) and DMR (chromosome 10: 127578032–127578186). *t*-SNE plot colour by the mCH fraction and RNA level of the gene; mCG fraction and chromatin accessibility of the DMR; and the gene–DMR contact score. **i**, Left, schematic explaining the PageRank (PR) score calculation. Right, dot plots of the normalized PageRank score and RNA expression of TFs in hindbrain subclasses, with red dots coloured and sized by PageRank score; purple dots coloured by RNA counts per million (CPM) and sized by the percentage of cells in the subclass with gene expression. All the PCC values were calculated across cell subclasses ($n = 274$), and adjusted $P$ values were obtained using permutation test and FDR correction (Methods).

upstream, downstream, upstream–intragenic, downstream–intragenic and upstream–downstream (Fig. 4l). Our results revealed that the contact strength of intragenic, upstream and downstream interactions were mostly negatively correlated with gene body methylation (per cent negative PCC, intragenic = 88%, upstream = 71%, downstream = 67%), a result consistent with the observation that the gene-body domain forms between the TSS and TTS, insulating the interactions between intragenic, upstream and downstream while increasing their interaction within each group. Moreover, the upstream–intragenic and downstream–intragenic interactions were primarily positively correlated with gene-body methylation (per cent positive PCC, upstream–intragenic = 63%, downstream–intragenic = 77%). However, the negatively correlated interactions probably remain crucial as they potentially link distal regulatory elements to intragenic regions (Fig. 4j). Upstream–downstream interactions exhibited the least negative correlations (per cent negative PCC, upstream–downstream = 15%) and did not directly interact with the gene body, which potentially relates to their higher-level chromatin conformation regulation.

Despite these general trends, the specific chromatin conformation landscapes of individual genes were highly diverse (Fig. 4m).

In addition to the notable upstream–intragenic and downstream–intragenic interactions observed in *Lingo2*, many megabase-long genes displayed complex intragenic subdomain patterns (for example, *Ptprd*, *Nrxn3* and *Lsamp*; Fig. 4m and Extended Data Fig. 10e–j). These patterns may correspond to more subtle gene activity regulation, including alternative TSS and exon usage, which will be explored below.

## The multi-omic GRNs

Numerous important TFs orchestrate the intricate spatial and cell-type-specific gene expression patterns within GRNs, which can be elucidated using multi-omic information[42,43]. Here we present a framework that connects TFs with DMRs and their potential downstream target genes, leveraging DNA methylome and chromatin conformation signals to construct GRNs for whole-brain neurons (Fig. 5a, left, and Methods). Our approach uses mCH fractions as proxies for gene status and mCG fractions as indicators of regulatory element activity. To further support our findings, we incorporated integrated transcriptome and accessibility profiles as complementary evidence

because of their strong negative correlation with gene mCH and DMR mCG fractions, respectively (Fig. 5a, right).

We built connections between the following elements: (1) DMRs and their potential target genes (DMR–target edge); (2) TFs and their potential target genes (TF–target edge); and (3) TFs and their potential binding DMRs (TF–DMR edge). We established DMR–target edges by accounting for the correlation of methylation fractions between the DMR and surrounding genes and the chromatin conformation landscape of the gene as discussed above (Fig. 5b and Methods). This approach intersected the diversity of both modalities measured in our snm3C-seq assay by limiting correlation-based edges to genome regions displaying distinct chromatin conformation changes. This step generated $1.2 \times 10^6$ edges between $5.7 \times 10^5$ DMRs and $2.1 \times 10^4$ genes, with 27% of edges connecting intragenic DMRs to genes and 73% linking distal DMRs (Fig. 5c). For instance, the edges of *Psd2* and *Celf2* demonstrated highly concordant cell-type-specificity of DNA methylation and chromatin interaction between gene bodies and their associated DMRs (Extended Data Fig. 11a). Next, we proceeded to connect TF–target edges on the basis of their correlated methylation fractions (Fig. 5d). We identified a total of $4.6 \times 10^6$ edges between 1,705 TFs and $2.6 \times 10^4$ genes.

As the TF–target edge alone is insufficient to discern gene regulation relationships[43], we also quantified the TF–DMR edges by considering the correlation of methylation fractions between the DMR and TF gene body and the enrichment of TF binding motifs in the correlated DMR sets (Extended Data Fig. 11b and Methods). In the motif enrichment analysis, we discovered that many TFs have their motifs solely enriched in the DMRs that positively correlated with TF gene-body methylation, such as *Nfia*, *Onecut2* and *Rfx1* (Fig. 5e and Extended Data Fig. 11c). This finding implies that the binding of these TFs potentially activates the underlying regulatory elements of these genes. We also observed some TFs with motifs enriched in negatively correlated DMRs, such as for FOXP2 (Extended Data Fig. 11d), which has been reported to have transcription repression functions[44], which is potentially achieved by repressing active enhancers. We identified $1.2 \times 10^7$ edges between 843 TFs and $4.6 \times 10^5$ DMRs (Extended Data Fig. 11e).

We combined all three types of edges (DMR–target, TF–target and TF–DMR) to construct the final GRN with TF–DMR–target triples. Each triple was assigned a final score that represented the overall correlation of cell-type specificity between the three components (Fig. 5f and Methods). The resulting network comprises $1.04 \times 10^7$ triples, involving 830 TFs, 20,101 genes and 291,752 non-overlapping DMRs (Fig. 5g). The different combination of correlations in a triple provides insights into regulatory relationships among the TF, DMR and target gene (Extended Data Fig. 11f,g and Supplementary Note 3).

In addition, the individual TF–DMR–gene triples predicted numerous TF and gene relationships, pinpointing their intermediate regulatory elements. These relationships were supported by the DNA methylome and chromatin conformation data, as well as the integrated transcriptome and chromatin accessibility data. For example, one high-scoring triple (0.74) linked the crucial neuronal TF EGR1 to its downstream target gene *Nab2* through a distal DMR (Fig. 5h). Expression of *Nab2* is known to be induced by EGR1, and the NAB2 protein in turn represses EGR1 activation function, thereby forming a negative feedback loop[45]. Two additional triples (*Egr1–Erf* and *Egr1–Synpo*) are illustrated in Supplementary Note 4 (Extended Data Fig. 12a–d). These examples demonstrate the power of our approach to identify new and biologically relevant gene regulatory relationships by leveraging multi-omic data.

## Key TFs in the GRN

TFs play a crucial part in regulating cell identity[46]. To demonstrate the importance and specificity of TFs within each cell subclass, we utilized our comprehensive GRN with the Taiji framework[47]. Using the PageRank algorithm, this framework identifies key TFs by propagating gene and regulatory element information on the GRN with node and edge weights specific to each cell subclass.

Focusing on the hindbrain (Fig. 5i) and midbrain (Extended Data Fig. 12g) as examples, we discovered key TFs that exhibited highly specific PageRank scores among cell subclasses within these complex brain regions. The combination of TF PageRank scores uniquely identified each cell subclass in these regions, aligning with their respective transcription specificities. Notably, the PageRank score was able to capture the specificity of TFs that were expressed at extremely low levels (Fig. 5i and Extended Data Fig. 12h), which is probably due to the gene-body methylation measurements.

Furthermore, we noted multiple instances of TFs within the same family, such as the RFX family, that displayed distinct cell-type-specific PageRank scores despite having nearly identical DNA-binding motifs (Extended Data Fig. 12i and Supplementary Note 5).

The comprehensive GRN and the PageRank algorithm effectively identified key TFs with high cell-type specificity in diverse brain regions. This approach generated numerous predictions about TF functions in determining cell identity and paves the way for future perturbation experiments[48].

## Epigenetic and RNA isoform heterogeneity

Alternative splicing leads to the production of different isoforms from the same gene, and dysfunction of this process in the brain has been associated with various neurodevelopmental disorders[49]. It is regulated by various RNA-binding proteins and has recently been associated with DNA methylation[50]. The diversity of isoform expression has been reported in several cortical cell types[51]. However, their diversity in a considerably wider range of cell types across the entire mouse brain and their relationship with the epigenome remains to be elucidated. To investigate these questions, we integrated the snmC and snm3C-seq datasets with a companion full-length scRNA-seq (SMART-seq) dataset from the BICCN, which contains 195,680 cells covering the entire adult mouse brain[6] (Methods). This integration enabled us to explore the intragenic diversity of DNA modification and topology in conjunction with RNA transcript and exon level measurement at cell-group resolution (Fig. 6a and Methods).

To exemplify this framework, we first examined the methylation pattern of the gene encoding neurexin 3 (*Nrxn3*), a crucial presynaptic gene that regulates synapse recognition through alternative isoforms[52]. We observed that *Nrxn3* is broadly expressed across neurons, with its isoforms (*α-Nrxn3* and *β-Nrxn3*) showing diverse patterns among cell subclasses (Fig. 6b). Notably, the expression patterns of these isoforms also matched with the methylation fraction of single CpGs located around the *Nrxn3* gene body (Extended Data Fig. 13a), with two particularly highly correlated regions positioned downstream of the first exon on *α-Nrxn3* and *β-Nrxn3* (Fig. 6c, regions 1 and 2, respectively). Similarly, the neuron-specific antioxidant gene *Oxr1* also exhibited intragenic methylation heterogeneity that matched the diversity of several transcripts and exons (Extended Data Fig. 13b).

To systematically analyse this phenomenon, we conducted a machine-learning-based analysis to quantify the predictability of alternative splicing using intragenic DNA methylome or chromatin conformation features in each cell group (Methods). Specifically, we assessed the level of improvement that can be obtained by incorporating high-resolution intragenic features to predict isoform expression levels compared with using whole gene-body measurements as a proximate averaged activity of isoforms. To that end, we trained two models for each gene (Fig. 6d): one with the true intragenic features and another using within-sample shuffled features that disrupted intragenic correspondence but preserved the sample-level information for each gene. We calculated PCC scores between the predicted and true values across cell groups for both models. The ΔPCC value between the true and shuffled models represented

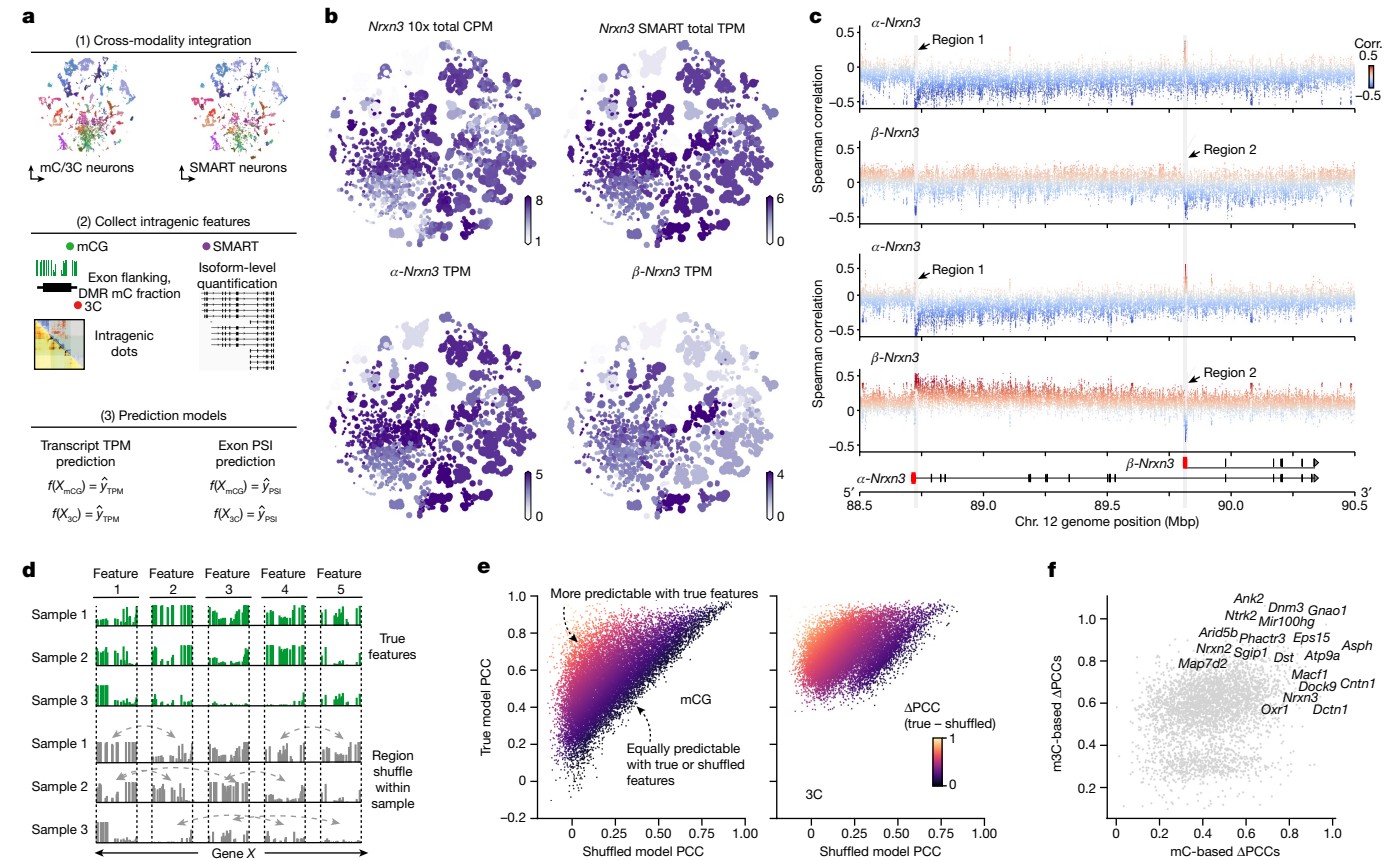

**Fig. 6 | Epigenetic heterogeneity predicts gene isoform diversity. a**, Workflow for the integrative analysis between epigenome and transcriptome datasets. **b**, Cell-group-centroids *t*-SNE plot coloured by *Nrxn3* CPM in scRNA-seq (10x) data, *Nrxn3* transcript per million (TPM) in SMART-seq (sum up all transcripts), *α-Nrxn3* TPM (Ensembl database identifier ENSMUST00000163134) and *β-Nrxn*3 TPM (Ensembl database identifier ENSMUST00000110130). **c**, Scatterplots of the correlation (Corr.) between transcript expression (*y* axis) and the methylation level of adjacent single CpG sites (dot) at the *Nrxn3* gene body. The arrows point to two most correlated regions (region 1 and region 2). From top to bottom, the scatterplots show the correlation information for CpG mCG fractions with *α-Nrxn3* and *β-Nrxn3* transcript TPM and the per cent spliced in

(PSI) values of the first exon of *α-Nrxn3* and *β-Nrxn3*. Interactive browser for region 1 available at tinyurl.com/fig6c-region1, and for region 2 at tinyurl.com/fig6c-region2. **d**, Schematic of the process for constructing the prediction model with true or shuffled features. For each gene, we used the exon, exon-flanking region and intragenic DMRs as the mC features. The 3C features are all the intragenic highly variable interactions (Methods). **e**, Scatterplot of the PCC values between predicted TPM and true TPM for each highly variable transcript (dot), using methylation features (mCG; left) and chromatin contact interactions (3C; right) for prediction. **f**, Scatterplot of the ΔPCC in mC models (*x* axis) and m3C models (*y* axis) for highly variable transcripts (dot). Top transcripts with large ΔPCC values are listed by their corresponding gene names.

the gain in predictability through adding intragenic features (Fig. 6e and Extended Data Fig. 13c). Many crucial neuronal and synaptic genes known to express functional alternative isoforms exhibited a large ΔPCC in their highly variable transcripts and exons (for example, *Nrxn1*, *Nrxn2* and *Nrxn3* (ref. 52), *Ntrk2* (ref. 53) and *Oxr1* (ref. 54); Fig. 6f and Extended Data Fig. 13d). Notably, chromatin conformation features demonstrated better overall prediction accuracy than DNA methylation in these alternatively spliced genes (Fig. 6f and Extended Data Fig. 13c), which is possibly because these features account for genome 3D interaction, whereas methylation features only consider 1D. This observation aligns with the understanding that many alternative splicing events involve nuclear compartmentalization and long-range genome interactions[55].

Finally, the prediction models prioritized specific transcripts and exons for which alternative usage is more likely under epigenetic regulation. We evaluated several representative examples in the genome browser, such as the promoters for *α-Nrxn3* and *β-Nrxn3* or the first exon of *Orx1* (Supplementary Note 6 and Extended Data Fig. 13e,f). Together, these results highlight the complex interplay between epigenetic regulation and alternative splicing, unveiling potential cell-type-specific regulatory mechanisms contributing to the post-transcriptional diversity of neuronal and synaptic genes in the brain.

## Discussion

This study presented a single-cell DNA methylation and 3D multi-omic atlas of the entire mouse brain. By utilizing methylome-based clustering and cross-modality integration with additional BICCN companion datasets[6,11], we established a cell-type taxonomy consisting of 4,673 cell groups and 274 subclasses. Our integrative approach combined five molecular modalities—gene mCH, DMR mCG, chromatin conformation, accessibility and gene expression—to create a multi-omic genome atlas featuring thousands of cell-type-specific profiles. Furthermore, we identified 2.6 million DMRs at two clustering granularities, which offers a large pool of candidate regulatory elements for various analyses. Notably, the intricate cellular diversity within the mouse brain exhibited extensive concordance across all molecular modalities, as evidenced by the aligned cell-type-specific patterns observed in numerous essential neuronal genes (Extended Data Fig. 5) and groups of regulatory elements (Extended Data Fig. 6). These findings underscore the fundamental interplay between epigenetics and transcriptomics in shaping the cellular diversity of the brain and serve as a foundation for incorporating additional complementary molecular modalities, such as histone modification, 5hmC, translatome and proteome, in future analyses. However, the comprehensive scope of

this study presented challenges in addressing additional biological aspects such as intrahemispheric differences, individual variability and sex differences. Future research endeavours are anticipated to explore these areas to contribute to a more comprehensive molecular atlas of the brain.

We also observed extensive spatial diversity encoded within the DNA methylome across the entire mouse brain. This epigenetic spatial pattern demonstrated high concordance with spatial transcriptional diversity, as evidenced through integration with a MERFISH dataset generated from spatially diverse methylated genes. By leveraging whole-brain MERIFSH datasets from a companion study[6], we achieved a detailed spatial map of DNA methylation and chromatin conformation profiles within delicate brain structures. The results offer a valuable anatomical context for methylation and 3D multi-omic cell data and emphasize the considerable influence of epigenetic regulation on spatial cell organization within the brain.

Building on the foundation of our high-resolution, spatially annotated multi-omic brain cell atlas, we expanded our investigation to the mouse genome to explore the underlying gene regulatory diversity across multiple scales. At the whole-chromosome level, the chromatin compartment identity of megabase-long regions can undergo significant alterations among different brain cell types. These changes were negatively correlated with DNA methylation, particularly at mCG sites. Genes within these regions play important parts in neuronal functions, especially in neurodevelopment. We also observed that TAD boundaries tended to form around neuronal long genes, with a negative correlation identified between boundary probability and the transcript body mCH fraction. A recent discovery of a similar gene boundary feature termed the transcription elongation loop offers a potential explanation for the higher gene domain boundary probability observed[56]. However, the mechanism by which the diversity of this feature arises across various brain cell types remains to be elucidated.

We also conducted an unbiased investigation of the chromatin conformation context surrounding individual genes by performing ANOVA and correlation analyses using whole-brain populations. This approach produced gene-specific chromatin conformation landscapes that offer predictions about the importance of individual chromatin interaction pixels at 10-kb resolution. These results offer numerous candidate loci that can potentially elucidate the causal relationships between DNA methylation statuses and chromatin structures. It paves the way for using advanced technologies such as epigenetic editing[57] in future investigations.

Integrating the extensive gene, DMR and chromatin conformation data enabled us to construct a comprehensive GRN for gene regulation in the mouse brain. This network predicted regulatory relationships between TFs and their target genes through the precise DMRs containing TF-binding motifs. Furthermore, numerous TF motifs were strongly enriched in DMRs, at which mCG fractions correlated positively or negatively with the TF mCH fraction. This result indicated dominant activation or repression roles for the corresponding TFs. Personalized PageRank analysis of the GRN identified the most influential TFs for each cell subclass in subcortical regions characterized by vast cellular diversity. The GRN also revealed diverse cell-type-specific patterns among members of the same TF family. Finally, the high-resolution methylome and chromatin conformation data enabled us to examine the relationship between epigenetic modalities and alternative isoforms. Our findings suggest that extensive intragenic epigenetic heterogeneity may contribute to the regulation of alternative promoter and exon splicing in these genes. The predictive model identified top candidates for further investigation into their causal relationships.

In summary, our analyses underscored the potential of this whole-brain dataset to characterize cellular, spatial and epigenomic diversity at high resolution. Furthermore, this resource, as demonstrated in our web application (mousebrain.salk.edu/), offers valuable insights into the fundamental gene regulation principles that shape the complexity of the mammalian brain and lay the groundwork for a deeper understanding of the molecular underpinnings of the human brain.

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

## Methods

### Mouse brain tissues

All experimental procedures using live animals were approved by the Salk Institute Animal Care and Use Committee under protocol number 18-00006. Adult (P56) C57BL/6J male mice were purchased from the Jackson Laboratory at 7 weeks of age and maintained in the Salk animal barrier facility on 12-h dark–light cycles with food ad libitum for up to 10 days (housing conditions: temperature of 21–23 °C, relative humidity of 61–63%). Brains were extracted (between P56 and P63), sliced and dissected in an ice-cold dissection buffer as previously described[9]. For snmC-seq3 samples, brains were sliced coronally at 600-µm intervals from the frontal pole across the whole brain, producing 18 slices, and dissected brain regions were obtained according to the Allen Brain Reference Atlas CCF (version 3)[59] (CCFv3) (Extended Data Fig. 1a) For all the snm3C-seq samples, brains were sliced coronally at 1,200 µm, which resulted in a total of 9 slices and dissected 2–6 combined brain regions according to the CCFv3 (Extended Data Fig. 1b). For nuclei isolation, each dissected region was pooled from 3–30 animals, and 2–3 biological replicas were processed per region. Comprehensive brain dissection metadata are provided in Supplementary Table 1. No statistical methods were used to predetermine sample sizes. We empirically determined the use of two to three biological experiments for all single-cell epigenomic experiments to achieve minimum reproducibility for this large-scale project. Blinding and randomization was not performed during handling of the tissue samples. Additionally, all dissected regions were digitally registered into CCFv3 using ITK-SNAP[60] (v.4.0.0) at 25 µm resolution (details of the annotated voxel file available in the Data availability section).

### Isolation of nuclei and FANS

For snmC-seq3 samples, the nuclei were isolated and sorted into 384-well plates using previous methods[9] with modifications described in Supplementary Methods (sections I and III). In brief, single nuclei were stained with AlexaFluor488-conjugated anti-NeuN antibody (A60, monoclonal, MAB377X, Millipore, 1:500 dilution) and Hoechst 33342 (62249, ThermoFisher) followed by FANS using a BD Influx sorter in single-cell (one drop single) mode. For each 384-well plate, NeuN+ (488+) nuclei were sorted into columns 1–22, whereas NeuN− (488−) nuclei were sorted into columns 23 and 24, achieving an 11:1 ratio of NeuN+ to NeuN− nuclei (Supplementary Note 7). The snm3C-seq protocol included additional in situ 3C treatment steps during preparation of the nuclei, which allowed the chromatin conformation modality to be captured. These steps were performed using an Arima-3C BETA kit (Arima Genomics), with a detailed protocol provided in Supplementary Methods (section II).

### Library preparation and Illumina sequencing

Both snmC-seq3 and snm3C-seq samples followed the same library preparation protocol (detailed in Supplementary Methods). This protocol was automated using a Beckman Biomek i7 and Tecan Freedom Evo instrument to facilitate large-scale applications. The snmC-seq3 and snm3C-seq libraries were sequenced on an Illumina NovaSeq 6000 instrument, using one S4 flow cell per 16 384-well plates and using 150 bp paired-end mode. The following software were used during this process: BD Influx (v.1.2.0.142; for flow cytometry), Freedom EVOware (v.2.7; for library preparation), Illumina MiSeq control (v.3.1.0.13) and NovaSeq 6000 control (v.1.6.0/RTA, v.3.4.4; for sequencing), and Olympus cellSens Dimension 1.8 (for image acquisition).

### Mapping and primary QC

The snmC-seq3 and snm3C-seq mapping was conducted using the YAP pipeline (cemba-data package, v.1.6.8) as previously described[9]. Specifically, the main mapping protocol included the following steps: (1) demultiplexing FASTQ files into single cells (cutadapt[61], v.2.10); (2) read-level QC; (3) mapping (one-pass mapping for snmC, two-pass mapping for snm3C) (bismark[62], v.0.20; bowtie2 (ref. 63), v.2.3); (4) BAM file processing and QC (samtools[64], v.1.9; Picard, v.3.0.0); (5) methylome profile generation (allcools, v.1.0.8); and (6) chromatin contact calling (snm3C-seq only). Snakemake[65] pipeline files with detailed mapping steps are provided in the Code availability section. All reads were mapped to the mouse mm10 genome. The gene and transcript annotation used in this study was based on a modified version of the GENCODE vm23 GTF file generated by the BICCN consortium in accordance with a previous study[6].

Primary QC for DNA methylome cells included the following criteria: (1) overall mCCC level of <0.05; (2) overall mCH level of <0.2; (3) overall mCG level of >0.5; (4) total final reads of >500,000 and <10,000,000; and (5) bismark mapping rate of >0.5. Note that the mCCC level estimates the upper bound of the cell-level bisulfite non-conversion rate. Additionally, we calculated lambda DNA spike-in methylation levels to estimate the non-conversion rate for each sample. All samples demonstrated a low non-conversion rate (<0.01; Extended Data Fig. 2i). We chose loose cut-off values for the primary filtering step to prevent potential cell or cluster loss. The clustering-based QC described below accessed potential doublets and low-quality cells. For the 3C modality in snm3C-seq cells, we also required cis-long-range contacts (two anchors >2,500 bp apart) >50,000.

### Analysis infrastructures

The whole-brain dataset comprised nearly 0.5 million single-cell or 5,000 pseudo-bulk mC profiles and 0.2 million single-cell or 2,500 pseudo-bulk 3C profiles. The dataset size was much larger than previous bulk and single-cell studies of mC or 3C[1,9]. To enable efficient whole-brain data analysis, we formatted the entire multidimensional epigenomic data into three primary tensor datasets and used them as inputs for analysis at two different stages.

The first stage was cellular analysis. We used a cell-by-feature tensor called methylome cell dataset (MCDS) to carry out methylome-based clustering and cross-modality integration, as illustrated in Figs. 2 and 3. Here we focused on individual cells with aggregated genomic features, such as kilobase chromosome bins and gene bodies. This analysis enabled us to aggregate single-cell profiles into pseudo-bulk levels by clustering and annotation. The pseudo-bulk merge increased genome coverage while eliminating the need to frequently access hundreds of terabytes of single-cell files in the subsequent analysis stage.

The second stage was genomic analysis, for which we used a pseudo-bulk-by-base tensor for mC, called base-resolution dataset (BaseDS), and a pseudo-bulk-by-2D-genome tensor for 3C, termed cooler dataset (CoolDS), to perform methylome and chromatin conformation analysis at flexible genomic resolutions, as depicted in Figs. 4–6. These pseudo-bulk tensors were generated at cell-group (thousands of profiles) and subclass (hundreds of profiles) levels to support multiple cellular granularities required by different analyses.

The large tensor datasets were stored using the chunked and compressed Zarr format[66], hosted within the object storage of the Google Cloud Platform. Data analysis was conducted using ALLCools[9], Xarray[67] and dask[68] packages. To facilitate large-scale computation, the Snakemake package[65] was used to construct pipelines, whereas the SkyPilot package[69] was utilized to set up cloud environments. Additionally, the ALLCools package (v.1.0.8) was updated to perform methylation-based cellular and genomic analyses, and the scHiCluster[70] package (v.1.3.2) was updated for chromatin conformation analyses. In the Data and Code availability sections, we provide information for these tensor storage and reproducibility-related details (package version, analysis notebook and pipeline files). For simplicity, the description below focused mainly on key analysis steps and parameters.

### Methylome clustering analysis

After mapping, single-cell DNA methylome profiles of the snmC-seq and snm3C-seq datasets were stored in the 'all cytosine' (ALLC) format,

a tab-separated table compressed and indexed by bgzip/tabix[71]. The 'generate-dataset' command in the ALLCools package helped generate a methylome cell-by-feature tensor dataset (MCDS). We used non-overlapping chromosome 100-kb (chrom100k) bins of the mm10 genome to perform clustering analysis; gene body regions ±2 kb for clustering annotation and integration with the companion transcriptome dataset; and non-overlapping chromosome 5-kb (chrom5k) bins for integration with the chromatin accessibility dataset. Details about the integration analysis are described below.

**Pre-clustering.** We performed two iterative clustering analyses for both the snmC and snm3C datasets. The first was a four-round pre-clustering step for QC purposes. The pre-clusters defined in this round contained potential doublets or low-quality cells (corresponding to debris or debris clumps in sorting). We started with all cells passing the primary QC filters and used the plate-normalized cell coverage (PNCC) metric to mark problematic pre-clusters. This metric was calculated using the final mC reads of each cell divided by the average final reads of cells from the same 384-well plate. We reasoned that cells at the same plate underwent all the library preparation steps inside the same PCR machine, thereby sharing the closest batch conditions. We observed some pre-cluster aggregating cells mostly showing extreme PNCC values (<0.5-fold or >2-fold) compared with most other clusters, which is a hallmark of problematic cells (Extended Data Fig. 2i). For each pre-cluster, we performed a permutation-based statistical test to call this abnormality. First, we randomly sampled null-population cells with the cluster size, stratified on sample composition 10,000 times. We then calculated *P* values for the observed PNCC mean (two-tailed test, larger or smaller) and standard deviation (s.d., one-tailed test, larger) compared with the null PNCC mean and s.d. distribution. After calculating the FDR using the Benjamini–Hochberg procedure[72], we marked pre-clusters as low-quality with absolute($\log_2$(PNCC)) > 0.8 and FDR < 0.01 (for mean or s.d.). In total, 8,979 (2.77%) snmC and 737 (0.38%) snm3C cells were removed from further analyses.

**Methylome clustering.** We then performed iterative clustering using the DNA methylome to determine whole-brain cell clusters. For both the snmC and snm3C datasets, we performed four rounds of iteration with the mCH and mCG fractions of chrom100k matrices. The clustering analysis within each iteration has been described in a previous study[9]. We also provide information about annotated Jupyter notebooks in the Code availability section, detailing the functions and parameters used in each step. Most functions were derived from the allcools[9], scanpy[73] and scikit-learn[74] packages. In summary, a single iteration consisted of the following main steps:
1) Basic feature filtering based on coverage and the ENCODE blacklist[75].
2) Highly variable feature (HVF) selection.
3) Generation of posterior chrom100k mCH and mCG fraction matrices, as used in the previous study[9] and initially introduced in ref. 76.
4) Clustering with HVF and calculating cluster enriched features (CEFs) of the HVF clusters. This framework was adapted from the cytograph2 (ref. 37) package. We first performed clustering based on variable features and then used these clusters to select CEFs with stronger marker gene signatures of potential clusters. The concept of CEF was introduced in ref. 77. The CEF calling and permutation-based statistical tests were implemented in ALLCools. clustering.cluster_enriched_features, for which we selected for hypomethylated genes (corresponding to highly expressed genes) in methylome clustering.
5) Calculate principal components (PCs) in the selected cell-by-CEF matrices and generate the *t*-SNE[78] and UMAP[79] embeddings for visualization. *t*-SNE was performed using the openTSNE[80] package using previously described procedures[81].
6) Consensus clustering. We first performed Leiden clustering[82] 200 times, using different random seeds. We then combined these result

labels to establish preliminary cluster labels, which were typically larger than those derived from a single Leiden clustering owing to its inherent randomness[82]. Following this, we trained predictive models in the PC space to predict labels and compute the confusion matrix. Finally, we merged clusters with high similarity to minimize confusion. The cluster selection was guided by the R1 and R2 normalization applied to the confusion matrix, as outlined in the SCCAF package[83].

The iterative process of training and merging continued until the performance of the model on withheld test data achieved a specified accuracy (0.95 for the first round, >0.9 for all subsequent rounds). The resolution parameter of the Leiden algorithm significantly influenced cluster number and randomness (that is, variation in cluster membership as random seeds changed). Therefore we used relatively small resolution values during each clustering stage (0.25 for the first iteration, 0.2–0.5 for the remaining iterations; the Scanpy default is 1). This approach substantially reduced randomness while also underestimating cluster numbers. However, during the four rounds of iteration, any under-split clusters were further delineated in subsequent rounds. This framework was incorporated in ALLCools.clustering. ConsensusClustering.

For each clustering round, we assessed whether a cluster required additional clustering in the next iteration based on two criteria: (1) the final prediction model accuracy exceeded 0.9, and (2) the cluster size surpassed 20. In total, we executed four iterative clustering rounds, which produced the following cluster numbers: 61 (L1), 411 (L2), 1,346 (L3) and 2,573 (L4). We further separated cells within L4 clusters in the final round by considering their brain dissection region metadata. We first divided all dissection regions with more than 20 cells in an L4 cluster while combining other regions with fewer than 20 cells with their nearest regions based on the average Euclidean distance in the PC space of L4 clustering. The final 4,673 cell groups combined L4 clusters and dissection regions. Incorporating dissection region data, which offered insights into the physical location of a cell, enhanced the flexibility of the analysis, such as enabling spatial region comparisons. Furthermore, we acknowledged that generating pseudo-bulk profiles from cell-level data demanded substantial computational resources. Aggregating cells at a higher granularity initially facilitated more straightforward merging later, such as combining them at the subclass level during subsequent analyses.

**Cluster-level DNA methylome analysis**
After clustering analysis, we merged the single-cell ALLC files into pseudo-bulk level using the allcools merge-allc command. Next, we used allcools generate-base-ds to generate the BaseDS from multiple ALLC files. The BaseDS was a Zarr dataset storing sample-by-base tensors for the entire dataset and allowed querying cytosines by genome position and methylation context (CpG and CpH). Next, we performed DMR calling as previously described[9,23,84] using the ALLCools.dmr.call_dms_from_base_ds and ALLCools.dmr.DMSAggregate functions that were reimplemented for BaseDS. In brief, we first calculated CpG differential methylated sites using a permutation-based root mean square test[84]. The base calls of each pair of CpG sites were combined before analysis. We then merged the differential methylated site into a DMR if they were within 250 bp and had PCC > 0.3 across samples. Because the genome coverage was unbalanced between samples, we proportionally downsampled the coverage at each base in each sample to base call coverage of 50 and a total base call coverage across samples of 3,000.

We applied the DMR calling framework across subclasses of the whole mouse brain and cell clusters within each major region. The two sources of DMRs were combined to capture the CpG fraction diversity in different cell-type granularities. There were around 10 million unique yet overlapping DMRs after the combination. We then merged the DMRs to obtain a final non-overlapping DMR list (bedtools merge -d 0),

which included 2.56 million DMRs. We report all the overlapping DMRs and non-overlapping DMRs in the Data availability section. In the subsequent analysis, when DMR was used to calculate correlation or scan motif occurrence, we started with the 10 million overlapping DMRs. We selected the DMR with the strongest value (that is, most significant PCC or highest motif score) among the overlapping ones. The DMRs in the final results were non-overlapping.

## Atlas-level data integration and cluster annotation

We established a highly efficient framework based on the Seurat R package[30] integration algorithm to perform atlas-level data integration with millions of cells. The integration framework consisted of three major steps to align two datasets onto the same space: (1) using dimension reduction to derive embedding of the two datasets in the same space; (2) using canonical correlation analysis (CCA) to capture the shared variance across cells between datasets and find anchors as five mutual nearest neighbours between the two datasets; (3) aligning the low-dimensional representation of the two datasets together with the anchors. We used genes to integrate methylome and transcriptome profiles, and chrom5k bins to integrate methylome and chromatin accessibility profiles.

**Integration of methylome and transcriptome profiles.** To integrate our snmC-seq dataset with scRNA-seq data[6], the gene expression levels of RNA cells were normalized by dividing the total unique molecular identifier count of the cell and multiplying the average total unique molecular identifier count of all cells and then log-transformed. For mC cells, the posterior gene-body mC level was used. The cluster-enriched genes (CEGs, similar to CEFs described above) were identified in each cell subclass and cluster using mC data. We checked the variance of the mC CEGs among the mC cells and RNA cells and only used the CEGs with mC variance values of >0.05 and expression variation values of >0.005 for the analyses. We reversed the sign of mC levels before integration owing to the negative correlation between gene-body DNA methylation and gene expression (Fig. 1d). We fit a principal component analysis (PCA) model with the mC cells and transformed the RNA cells with the model. The PCs were normalized by the singular value of each dimension to avoid the embedding being driven by the first few PCs.

To find anchors between mC and RNA cells, we first $Z$-score-scaled the mC matrix and expression matrix of CEGs across cells, and the resulting matrices were represented as $X$ (mC: cell-by-CEG) and $Y$ (RNA: cell-by-CEG), respectively. CCA was used to find the shared low-dimensional embedding of the two datasets, which was solved using singular value decomposition of their dot product $USV^T = XY^T$. $U$ and $V$ were normalized by dividing the L2-norm of each row, and were used to find five mutual nearest neighbours as anchors and to score anchors using the same method as Seurat[30].

The original CCA framework of Seurat (v.3) is difficult to scale up to millions of cells owing to the memory bottleneck, whereby the mC cell-by-RNA matrix was used as the input to CCA. To handle this limitation, we randomly selected 100,000 cells from each dataset ($X_{ref}$ and $Y_{ref}$) as a reference to fit the CCA and transformed the other cells ($X_{qry}$ and $Y_{qry}$) onto the same CC space. Specifically, the canonical correlation vectors (CCVs) of $X_{ref}$ and $Y_{ref}$ (denoted as $U_{ref}$ and $V_{ref}$, respectively) were computed using $U_{ref}SV_{ref}^T = X_{ref}Y_{ref}^T$, where $U_{ref}^TU_{ref} = I$ and $V_{ref}^TV_{ref} = I$. Then the CCV of $X_{qry}$ and $Y_{qry}$ (denoted as $U_{qry}$ and $V_{qry}$, respectively) were computed using $U_{qry} = X_{qry}(Y_{ref}^TV_{ref})/S$ and $V_{qry} = Y_{qry}(X_{ref}^TU_{ref})/S$, respectively. The embeddings from the reference and query cells were concatenated for anchor identification.

The PCs derived from the first step were then integrated using the same method as Seurat[30] through these anchors. Rather than working on the raw feature space in Seurat, our integration step projected the PCs of scRNA-seq (query, denoted as $U_r$) to the PCs of the snmC-seq (reference, denoted as $U_m$) while keeping the PCs of the reference dataset unchanged. This approximation considerably reduced the time and memory consumption for computing the corrected high-dimensional matrix and redoing the dimension reduction. For anchor $k$ pairing mC cell $k_m$ and RNA cell $k_r$, $B_k = U_{m_{k_m}} - U_{rmr_{k_r}}$ was considered the bias vector between mC and RNA. Then for each RNA cell as a query, we used its 100 nearest anchors to compute a weighted average bias vector representing the distance to move a RNA cell into the mC space. The distance between the query RNA cell and an anchor was defined as the Euclidean distance on the RNA dimension reduction space between the query RNA cell and the RNA cell of the anchor. The weights for the average bias vector depended on the distances between the query RNA cell and the anchors, for which close anchors received high weights.

**Integration of methylome and chromatin accessibility profiles.** PCA on gene-body signals was insufficient to capture the open chromatin heterogeneity in snATAC-seq data[10,30]. Latent semantic indexing (LSI) applied to binarized cell-by-5-kb bin matrices had demonstrated promising results for snATAC-seq data embedding and clustering[30]. Therefore, to align snATAC-seq data with snmC-seq data at a high resolution, we developed an extended framework based on the previously described approach[30] to utilize binary sparse cell-by-5-kb bin matrices as input.

We first derived a cell-by-5-kb bin matrix to represent the snmC-seq data. In a single cell $i$, we modelled its mCG base call $M_{ij}$ for a 5-kb bin $j$ using a binomial distribution $M_{ij} \sim Bi(cov_{ij}, p_i)$, where $p$ represented the global mCG level of the cell (and '$\sim$' indicates 'distributed as'). We then computed $P(M_{ij} > mc_{ij})$ as the hypomethylation score of cell $i$ at bin $j$. The less likely to observe smaller or equal methylated base calls, the more hypomethylated the bin was. We next binarized the hypomethylation score matrix by setting the values greater than 0.95 as 1, otherwise 0, to generate a sparse binary matrix $A$. We selected the columns with more than five non-zero values, then computed the column sum of the matrix (colsum$_j = \sum_{i=1}^{No.\ cells} A_{ij}$) and kept only the bins with $Z$-scored $\log_2$(colsum) values between $-2$ and 2. The snATAC-seq data were also represented in a binary cell-by-5-kb bin matrix, where 1 represented at least one read detected in a 5 kb bin in a cell. The features were filtered in the same way as the mC matrix, and the bins remaining in both datasets were used for further analysis.

LSI with log term frequency was used to compute the embedding. Term frequency–inverse document frequency (TF–IDF) transformation was applied to convert the filtered matrix $B$ to $X$. Specifically, $B$ was normalized by dividing the row sum of the matrix to generate the term frequency matrix TFreq, and further converted to $X$ by multiplying the inverse document frequency vector IDF.

$X_{ij} = \log(\text{TFreq}_{ij} \times 100,000 + 1) \times \text{IDF}_j$, where $\text{TFreq}_{ij} = B_{ij}/\sum_{j'=1}^{No.\ bins} B_{ij'}$ and $\text{IDF}_j = \log(1 + \text{no. cells}/\sum_{i'=1}^{No.\ cells} B_{i'j})$. The embedding of single cells $U$ was then computed using singular value decomposition of $X$, where $X = USV^T$. We fit the LSI model with mCG data $B_m$ to derive $U_m$. The intermediate matrices $S$ and $V$ and vector IDF were used to transform the ATAC data $B_a$ to $U_a$, by

$$\text{TFreq}_{a_{ij}} = \frac{B_{a_{ij}}}{\sum_{j'=1}^{No.\ bins} B_{a_{ij'}}}$$

$$X_{a_{ij}} = \log(\text{TFreq}_{a_{ij}} \times 100,000 + 1) \times \text{IDF}_j$$

$$U_a = X_a V / S$$

CCA was also performed with the downsampling framework using 100,000 cells from each dataset as a reference and the others as query, but taking the TF–IDF transformed matrices as input. The query cells were projected to the same space using the IDF and CCV of the reference cells. Specifically, $B_{m_{ref}}$ and $B_{a_{ref}}$ were converted to $X_{m_{ref}}$ and $X_{a_{ref}}$, respectively, with TF–IDF, and the CCVs (denoted as $U_{ref}$ and $V_{ref}$) were

computed using $U_{\mathrm{ref}}SV_{\mathrm{ref}}^{T} = X_{\mathrm{m_{ref}}}X_{\mathrm{a_{ref}}}^{T}$. Then $B_{\mathrm{m_{qry}}}$ and $B_{\mathrm{a_{qry}}}$ were converted to $X_{\mathrm{m_{qry}}}$ and $X_{\mathrm{a_{qry}}}$, respectively, with TF–IDF using the IDF of reference cells, and the CCVs (denoted as $U_{\mathrm{qry}}$ and $V_{\mathrm{qry}}$) were computed using $U_{\mathrm{qry}} = X_{\mathrm{m_{qry}}}(X_{\mathrm{a_{ref}}}^{T}V_{\mathrm{ref}})/S$ and $V_{\mathrm{qry}} = X_{\mathrm{a_{qry}}}(X_{\mathrm{m_{ref}}}^{T}U_{\mathrm{ref}})/S$. The subsequent steps to find anchors and align $U_{\mathrm{m}}$ and $U_{\mathrm{a}}$ were the same as integrating the mC and RNA data.

**Iterative integration group design.** Similar to the clustering analysis, we integrated two datasets iteratively to match cell or cell clusters at the highest granularity. We first separated the pass-QC datasets into integration groups based on independent cell-type annotation (described above or provided by data generators) and dissection information. For instance, non-neuronal cells, IMNs and granule cells ('DG Glut' and 'CB Granule Glut') were separated from neurons because they were showing large global methylation differences from other neurons and unbalanced in cell numbers across datasets owing to different sampling and sorting strategies. Within each integration group, we performed the integration iteratively. We used the co-clustering from the integrated low-dimensional space to match cells or clusters between the two datasets (see below). We then performed the next round of integration until the matched cells or clusters fulfilled the stopping criteria. We list details about each pair of iterative integrations below. The resulting cluster map between datasets and mC and m3C cluster annotation is provided in Supplementary Table 4. Information about a set of Jupyter Notebooks for a single integration process between each pair is provided in the Code availability section.

**Integration between snmC-seq and scRNA-seq or SMART-seq datasets.** We used the gene body ±2 kb as features to integrate mC and RNA datasets[6], mapping the RNA clusters to mC cell groups. We used the mCG fraction of the gene bodies for non-neuronal cells, IMNs and granule cells and the mCH fraction of the gene bodies for other neurons. In each iteration, we calculated a confusion matrix between 4,673 mC cell groups and 5,200 RNA clusters (provided by data generators) using the overlap score as previously described[9,85]. We then built a weighted graph using the confusion matrix as the adjacency matrix and performed a Leiden clustering (resolution = 1) to bicluster mC and RNA clusters. This step puts similar mC and RNA clusters into integration groups based on their overlap score. The RNA and mC clusters in the same integration group were further integrated to match at finer granularity in the next iteration unless any of the following stop criteria were met: (1) there was only one integration group from this round; (2) there was only one mC or RNA cluster in the integration group; (3) the mC cell number was <30; or (4) the RNA cell number was <100 for the scRNA-seq dataset or <30 for the SMART-seq dataset. After integration, we obtained a mC to RNA cluster map for each mC cell group, which we used as the reference to annotate cell subclasses and remaining hierarchies in the transcriptomic taxonomy. We also evaluated the spatial location and marker genes (neurotransmitter-related genes or other markers provided in the transcriptome annotation). We manually resolved conflicts when the RNA clusters corresponded to more than one subclass by checking the dissection metadata and marker genes. We combined all RNA cells assigned to each mC cell group to generate the matched transcriptome profile.

**Integration between snmC-seq and snATAC-seq datasets.** The snmC-seq dataset and snATAC dataset[11] shared the same dissection tissues. We utilized this experimental design to integrate cells from the mC and ATAC datasets within each major region. Of note, the snmC-seq data were enriched for NeuN[+] by FANS, whereas the snATAC data unbiasedly profiled all cells. Therefore, we also separated neurons from non-neuronal cells and IMNs to balance the integration, especially in the first round. We used the mCG hypomethylation score of chromosome non-overlapping 5-kb bins to perform the integration. The cluster assignment and stop criteria were similar to the mC–RNA integration method. The alignment score (Extended Data Fig. 6a) was calculated as previously described[86], using $k = 1\%$ cells of the dissection region or $k = 20$, whichever is larger. We combined all ATAC cells assigned to each mC cell group to generate the matched chromatin accessibility profile.

**Integration between snmC-seq and snm3C-seq datasets.** We used the non-overlapping chromosome 100-kb bin as features to integrate snmC-seq and snm3C-seq datasets. The cluster assignment and stop criteria were similar to the mC–RNA integration method. After integration, we also annotated the snm3C cell groups with the transcriptomic taxonomy.

## MERFISH experiment

**MERFISH gene panel design.** The genes in the GTF file were first filtered on the basis of lengths of >1 kb. We then selected genes using methods from a previous study[31] but used the snmC-seq dataset and gene-body mCH fraction to perform the calculation. In brief, we used two approaches to prioritize genes. The first approach was to use mutual information between gene-body mCH fraction and neuron subclass labels, which aims to select genes that are differentially methylated between groups of cell subclasses. The second approach was to perform pairwise differentially methylated gene analysis (ALLCools.clustering.PairwiseDMG) among clusters within the same major region and select genes being identified as DMGs in most cluster pairs. For the first approach, we selected the top 100 genes. We selected the top 50 genes from each major region for the second approach. Owing to the overlaps, there were 325 genes after this selection. In addition to the cell-type markers, we selected spatial markers by calculating the mutual information between the major region label of a cell and the mCH fraction across the brain, or between the dissection region label and mCH fraction within each major region. We added another 175 non-overlapping genes to obtain a total of 550 genes. We then performed the same analysis using a previously published scRNA-seq dataset[6] to obtain the RNA-based prioritization lists. We selected 500 final genes as the gene panel based on rank in the RNA list to ensure that these genes are also expressed and highly diverse in the transcriptome. Encoding probes for these genes were designed and synthesized by Vizgen (Supplementary Table 6).

**MERFISH tissue preparation and imaging.** Fresh P56–P63 whole mouse brains were sliced coronally at 1,200-μm intervals, and each slice was then embedded in OCT, rapidly frozen in isopentane and dry ice, and stored at −80 °C until ready for slicing. Coronal section (12-μm thick) were obtained from each OCT-blocked tissue using a Leica CM1950 cryostat, immediately fixed in 4% formalin (warmed to 37 °C) for 30 min, and permeabilized in 70% ethanol following the manufacturer's procedures. Sample preparation, including probe hybridization and gel embedding, was performed using a sample preparation kit from Vizgen (10400012) following the manufacturer's protocol. Each section was imaged using a MERSCOPE 500 Gene Imaging kit (Vizgen, 10400006) on a MERSCOPE (Vizgen).

**MERFISH data preprocessing and annotation.** MERFISH data analysis, including imaging, spot detection, cell segmentation and cell-by-gene matrix generation, was conducted using MERSCOPE instrument software (v.2023-01). We removed abnormal cells (artificial segmentation and doublets) from the cell-by-gene matrix in each experiment based on the following criteria: (1) cell volume <30 μm³ or > 2,000 μm³; (2) total RNA count <10 or >4,000; (3) total RNA counts normalized by cell volume <0.05 or >5; (6) total gene detected <3; and (5) cells with >5 blank probes detected (negative control probe included in the gene panel). We then integrated the pass-QC MERFISH cells with the scRNA-seq datasets[6] to annotate the MERFISH cells with transcriptome nomenclature using the ALLCools integration functions described above.

**Integration between MERFISH and snmC and snm3C datasets.** We integrated the snmC and snm3C datasets with the MERFISH dataset to evaluate whether the spatial pattern observed in the DNA methylome matched the spatial diversity observed in the gene expression data. Integration was similar to the mC–RNA integration described above. To utilize the dissection region metadata, we grouped the snmC-seq and snm3C-seq data by the slice and integrated them with a matched MERFISH slice. We also separated neurons and other cells, similar to the mC–RNA integration method described above. We used the 500 genes in the MERFISH gene panel to perform the integration. After integration, we imputed the spatial location of each methylation nucleus on the integrated low-dimensional space. We calculated the ten nearest MERFISH neighbours for each mC nucleus in each integration group. We assigned the coordinate of centroids of these MERFISH cells as the mC spatial location of the nucleus.

## Cell and cluster-level chromatin conformation analysis

**Generation of the chromatin contact matrix and imputation.** After snm3C-seq mapping, we used the *cis*-long range contacts (contact anchors distance of >2,500 bp) and *trans* contacts to generate single-cell raw chromatin contact matrices at three genome resolutions: chromosome 100-kb resolution for the chromatin compartment analysis; 25-kb bin resolution for the chromatin domain boundary analysis; and 10-kb resolution for the chromatin interaction analysis. The raw cell-level contact matrices were saved in the scool format[87]. We then used the scHiCluster package (v.1.3.2) to perform contact matrix imputation as previously described[70]. In brief, the scHiCluster imputed the sparse single-cell matrix by first performing a Gaussian convolution (pad = 1) followed by using a random walk with restart algorithm on the convoluted matrix. For 100-kb matrices, the whole chromosome was imputed, whereas for 25-kb matrices, we imputed contacts within 10.05 Mb. For 10-kb matrices, we imputed contacts with 5.05 Mb. The imputed matrices for each cell were stored in cool format[87]. The cell matrices were aggregated into cell groups or subclass levels identified in the previous section. These pseudo-bulk matrices were concatenated into a tensor called CoolDS and stored in Zarr format for brain-wide analysis.

**Compartment analysis.** We used the imputed subclass-level contact matrices at the 100-kb resolution to analyse the compartment. We first filtered out the 100-kb bins that overlapped with ENCODE blacklist (v.2)[75] or showed abnormal coverage. Specifically, the coverage of bin $i$ on chromosome $c$ (denoted as $R_{c,i}$) was defined as the sum of the $i$-th row of the contact matrix of chromosome $c$. We only kept the bins with coverage between the 99th percentile of $R_c$ and twice the median of $R_c$ minus the 99th percentile of $R_c$. Contact matrices were normalized by distance, and the PCC of the normalized matrices was used to perform the PCA[34]. The IncrementalPCA class from the sklearn package[74], which allows fitting the model incrementally, was used to fit a single PCA model incrementally for each chromosome using all the cell subclass matrices. We then transformed all the cell subclasses with the fitted model so that the PCs for each subclass were transformed from the same loading and eased the cross-sample correlation analysis. We also calculated the correlation between PC1 or PC2 and 100-kb bin CpG or gene density. We use the component with higher absolute correlation as the compartment score and assigned the compartment with higher CpG density with positive scores (A compartment).

**Compartment score and mC fraction correlation.** We first performed quantile normalization along subclasses using the Python package qnorm (v.0.8.0)[88] to normalize the mC fractions and compartment scores. We then calculated the PCC between the compartment scores of non-overlapping chromosome 100-kb bins, with the corresponding mCH or mCG fraction of the bin across cell subclasses. Because the negatively correlated compartment score of the bins had a much higher standard deviation among cell types (Fig. 4c), we selected the 300 most negatively correlated chrom100k bins and used their overlapped genes to perform gene ontology (GO) enrichment analysis (Fig. 4d) using Enrichr[36]. We randomly selected gene-length-matched background genes to adjust the long-gene bias in all the GO enrichment analyses[36]. To investigate the developmental relevance indicated by the GO enrichment result, we used the developmental mouse brain scRNA-seq atlas[37] at the subtype level (approximate granularity of subclass in this study). Genes overlapping 300 of the most negatively correlated bins, 300 of the mostly positively correlated bins and 300 of the low-correlation bins were used to construct the plot in Extended Data Fig. 9d.

**Domain boundary analysis.** We used the imputed cell-level contact matrices at 25-kb resolution to identify domain boundaries within each cell using the TopDom algorithm[89]. We first filtered out the boundaries that overlapped with ENCODE blacklist (v.2)[75]. The boundary probability of a bin was defined as the proportion of cells having the bin called a domain boundary among the total number of cells from the group or subclass. To identify differential domain boundaries between $n$ cell subclasses, we derived an $n \times 2$ contingency table for each 25-kb bin. The values in each row represent the number of cells from the group that has the bin called a boundary or not as a boundary. We computed the Chi-square statistic and $P$ value for each bin and used FDR < $1 \times 10^{-3}$ as the cut-off for calling 25-kb bins with differential boundary probability.

**Domain boundary probability and transcript body mC fraction correlation.** We first performed quantile normalization along subclasses using the Python package qnorm (v.0.8.0)[88] to normalize the transcript body mC fractions and chromosome 25-kb bin boundary probabilities. We then calculated the PCC between the differential boundary probabilities of 25-kb bins with the transcript body mCH and mCG fractions. We grouped transcripts with >90% overlap within a gene and used their longest range. We calculated the transcript-body mCH and mCG fraction at the subclass level for each transcript. We then calculated the PCC between the mC fractions and boundary probabilities of bins overlapping the transcript body ±2 Mb. We used a permutation-based test to estimate the significance of the correlation[90]. Specifically, we shuffled the boundary probability and mC fraction values within each sample (subclass), disrupting the genome relationship between the bins while preserving the sample-level global difference. We calculated the PCC using the shuffled matrices 100,000 times and used a normal distribution to approximate the null distribution for more precise $P$ value estimation in FDR correction. We then used FDR < $1 \times 10^{-3}$ as the significance cut-off value for each PCC between a transcript and a 25-kb bin. In Fig. 2g, we used deeptools[91] (v.3.5.1) to profile the boundary probability at transcript ±2 Mb 25-kb bins. In Fig. 2h and Extended Data Fig. 2f–h, we selected the top positively correlated bin and top negatively correlated bin for each long gene (transcript body length of >100 kb) and performed the GO analysis using length-matched background genes, as described above (Extended Data Fig. 2h).

**Highly variable interaction analysis.** We used the imputed cell-level contact at the 10-kb resolution to perform the highly variable interaction analysis, for which the interaction represented one 10 kb-by-10 kb pixel in the conformation matrix. We filtered out any interactions that had one of the anchors overlapping with ENCODE blacklist (v.2)[75]. We then performed ANOVA for each interaction to test whether the single-cell contact strength of that interaction displayed significant variance across subclasses. The $F$ statistics of ANOVA represented an overall variability of the interaction across the brain. To select highly variable contacts, we used $F > 3$ as the cut-off value, which was decided by visually inspecting the contact maps as well as fulfilling the FDR < 0.001 criteria. ANOVA was only performed on interactions

for which anchor distance was between 50 kb and 5 Mb, given that increasing the distance only led to a limited increase in the number of significantly variable and gene-correlated interactions (Extended Data Fig. 10b).

**Interaction strength and mC fraction correlation.** To investigate the relationship between gene status and the surrounding chromatin conformation diversity, we first performed quantile normalization along subclasses using the Python package qnorm (v.0.8.0)[88] to normalize the transcript body mCH fractions and contact strengths of highly variable interactions. We then calculated the PCC between the transcript body mCH fraction and the highly variable interactions if any anchor of the interactions had overlapped with the gene body. Similar to the domain boundary correlation analysis, we shuffled the contact strengths and mCH fractions within each sample and used the shuffled matrix to calculate null distribution and estimate FDR. We select FDR < 0.001 as a significant correlation.

## GRN analysis

We presented a framework for building a GRN based on the DNA methylome and chromatin conformation profiles at the cell subclass level. We used 212 neuronal cell subclasses requiring them to have >100 cells in both snmC and snm3C datasets. Notably, the same framework can be applied to other brain cell types or a subset of cells (such as certain brain regions or cell classes based on specific questions). The GRN was composed of relationships between TFs, their potential binding elements (represented by DMRs) and downstream target genes. Pairwise edges were constructed between DMRs and target genes (DMR–target), TFs and target genes (TF–target) and TFs and DMRs (TF–DMR). The basis of each pairwise edge was the correlation between the methylation fractions of the two genome elements across cell subclasses. We performed quantile normalization along subclasses using the Python package qnorm (v.0.8.0)[88] to normalize the two matrices involved in calculating the correlation. Gene-body mCH fraction was used as a proxy for TF and target gene activity, and mCG fractions were used to represent DMR status. Variable genes and TFs were selected if they were identified as CEFs (described in the clustering steps) in any subclass.

For the DMR–target edges, we selected the highly variable and positively correlated chromatin contact interactions of the gene based on the results in the previous section, and included DMRs situated in any anchor regions of the interactions. We then calculated the PCC between DMR mCG and gene mCH fraction. For a group of overlapping DMRs, we selected the one with the highest absolute PCC value to represent that group, making the edges of the DMRs non-overlapping. Similar to the domain boundary and interaction correlation analysis, we shuffled the DMRs and genes within each sample to calculate the null PCC and to estimate the FDR. We filtered DMR–target edges with FDR < 0.001. For the TF–target and TF–DMR edges, we calculated the PCC between TF and all CEF genes or between TF and all DMRs, respectively, and applied the same FDR < 0.001 cut-off value to filter edges. For the TF–DMR edge, we further performed motif enrichment analysis on the significantly correlated DMRs (explained in the next section). We only kept TF–DMR edges when the TF had any motif significantly enriched in the correlated DMR set, and the particular DMR had that motif occurrence.

After obtaining the three pairwise edges, we intersected the edges together into triples based on shared genes (including TFs and targets) and DMR identifiers. We calculated a final edge score $S_{all} = \sqrt[4]{|S_a S_b S_c S_d|}$ for each triple by taking the geometric mean of the absolute values of four correlations, where $S_a$ was the correlation of the DMR–target edge, $S_b$ was the correlation of the TF–DMR edge, $S_c$ was the TF–target edge and $S_d$ was the correlation between target gene mCH fraction and gene–DMR contact strength. If multiple gene-correlated interactions had anchors overlapping with DMR and gene body, we selected the one with the lowest negative correlation.

**DMR motif scan and TF motif enrichment analysis.** We used an ensemble motif database from SCENIC+ (ref. 43), which contained 49,504 motif position weight matrices (PWMs) collected from 29 sources. Redundant motifs (highly similar PWMs) were combined into 8,045 motif clusters through clustering based on PWM distances calculated using TOMTOM[92] by the authors of SCENIC+ (ref. 43). Each motif cluster was annotated with one or more mouse TF genes. To calculate motif occurrence on DMRs, we used the Cluster-Buster[93] implementation in SCENIC+, which scanned motifs in the same cluster together with hidden Markov models.

To perform motif enrichment analysis in the TF–DMR edge analysis, we used the recovery-curve-based cisTarget algorithm[43,58]. In brief, the cisTarget algorithm performed motif enrichment on a set of DMRs by calculating a normalized enrichment score for each motif based on all other motifs in the collection. For each TF gene, we applied the cisTarget algorithm to positively correlated or negatively correlated DMRs separately. We used the package default cut-off (normalized enrichment score > 3) to select enriched motifs for a DMR set. A leading-edge analysis was performed using cisTarget to assign motif occurrence in DMRs with Cluster-Buster scores passing a cut-off value in enriched cases[43].

**PageRank analysis on weighted networks.** We adopted the Taiji framework[94] to perform TF analysis on a weighted GRN for each cell subclass. This framework uses the personalized PageRank algorithm to propagate node and edge weight information across the network, calculating the importance of each TF. To add subclass information as network weights, we simplified the network by including only TF and target gene nodes and weighting the gene node by inverted gene-body mCH value in the subclass. Specifically, we first performed quantile normalization across all subclasses. We then performed a robust scale of the matrix using sklearn.preprocessing.RobustScaler with quantile_range = (0.1, 0.9). We then inverted the scaled mCH fraction by

$$W_i = (\max(CH_i) - CH_i)/(\max(CH_i) - \min(CH_i)),$$

where $CH_i$ and $W_i$ denoted the scaled gene mCH fractions and inverted values, respectively, for subclass $i$.

We also added DMR mCG fraction into the edge weights. Specifically, we performed the same quantile normalization and robust scale on all the mCG fractions of DMRs involved in the network and calculated the inverted DMR mCG value by

$$V_i = (\max(CG_i) - CG_i)/(\max(CG_i) - \min(CG_i)),$$

where $CG_i$ and $V_i$ denote the scaled DMR mCG fractions and inverted values, respectively, for subclass $i$. The edge weight between a TF and a target gene in subclass $i$ was calculated as $e = \frac{1}{n}\sum_{t=0}^{n} S_{i,t} \times V_{i,t}$, where $n$ denotes the number of DMRs that connect the TF to target gene, $S_{i,t}$ is the final score of one TF–DMR–target triple, and $V_{i,t}$ is the inverted DMR mCG value.

## Intragenic epigenetic and transcriptomic isoform analysis
### Integration and isoform quantification of the SMART-seq dataset.
Preprocessing and gene-level quantification using STAR[95] (v.2.7.10) was performed with AIBS data generators as previously described[5]. We used gene-level counts to perform cross-modality integration iteratively as described in previous sections. We used kallisto[96] with steps described in a previous study[51] to quantify the SMART-seq at the isoform level with the same GTF file used in transcriptome and methylome analysis above. We calculated cell-group-level transcript per million (TPM) values based on the integration result. We also calculated the exon PSI from the transcript counts in each gene. The SMART-seq browser tracks (Extended Data Fig. 13a,b) were constructed using STAR-aligned BAM files.

**Prediction model training.** First, we quantified mC and m3C intragenic features for predicting the alternative isoform and exon usage. We used the exon, exon-flanking region and intragenic DMRs as the mC features of each gene. The exon-flanking region was defined as upstream or downstream 300 bp of each exon. We removed features with variance <0.01 and combined features with >90% overlap in their genome coordinates. For 3C features, we used all the intragenic highly variable interactions ($F$ statistics > 3) from the above section as features.

After collecting all the features, we selected genes with highly variable transcripts and exons among cell groups for model training. Highly variable transcripts were selected on the basis of the following criteria: (1) mean TPM across cell groups of >0.2; (2) TPM standard deviation of >0.3; and (3) transcript body (TSS to TTS) length of >30 kb. Highly variable exons were selected based on PSI standard deviation of >0.02 and PSI 90% quantile and 10% quantile difference of >0.05. We trained four models for each gene, including predicting transcript TPMs using mC or 3C features and predicting exon PSIs using mC or 3C features. The training contains two steps. First, we used sklearn.feature_selection.SelectKBest with the score function f_regression to select the top 100 features for each transcript or exon. We then used all features that had been selected at least once. We performed fivefold cross-validation to train random forest models using selected features and sklearn.ensemble.RandomForestRegressor. In each cross-validation run, we calculated the PCC between predicted values and true values as the model performance. We also shuffled the selected features within each sample (Fig. 6c) to train the model and calculate PCC values again as the shuffled model performance.

## Reporting summary

Further information on research design is available in the Nature Portfolio Reporting Summary linked to this article.

## Data availability

The snmC-seq2 and snmC-seq3 data and MERFISH dataset are accessible through the Neuroscience Multi-omic Data (NeMO) Archive (assets.nemoarchive.org/dat-sig83t9). The snm3C-seq data are accessible through NeMO (assets.nemoarchive.org/dat-sig83t9) and the NCBI's Gene Expression Omnibus (GEO) database (identifier GSE213262). The whole-brain snATAC-seq dataset is from ref. 11. The whole-brain scRNA-seq MERFISH and SMART-seq datasets are from ref. 6. All the processed data related to results and method sections are available from the GitHub repository at github.com/lhqing/wmb2023. The Allen Brain Reference Atlas and CCF is from ref. 3. A detailed description of the data availability is provided at mousebrain.salk.edu/download.

## Code availability

The mapping pipeline for snmC-seq3 and snm3C-seq is available at hq-1.gitbook.io/mc/. Single-cell DNA methylome data analysis tools are available at ALLCools (v.1.0.8) Python package (lhqing.github.io/ALLCools/intro.html). Single-cell chromatin conformation data analysis tools are available at the scHiCluster (v.1.3.2) Python package (github.com/zhoujt1994/scHiCluster). Other codes and Jupyter Notebooks related to results and method sections are available from the GitHub repository at github.com/lhqing/wmb2023.

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

**Acknowledgements** We thank members of the BICCN community for discussions and the SkyPilot developers from UC Berkeley Sky computing laboratory for helping us establish cloud computing frameworks. This work utilized the Stampede2 supercomputing resources at the Texas Advanced Computing Center through allocation MCB130189 from the Extreme Science and Engineering Discovery Environment. This work was supported by grants from NIMH, U19MH114831 and U19MH114831-04S to J.R.E., B.R. and E.Callaway, and U19MH114830 to H.Z., all under the BRAIN Initiative of the National Institutes of Health (NIH). The Flow Cytometry Core Facility of the Salk Institute is supported by grant NIH-NCI CCSG: P30 014195 and Shared Instrumentation Grant S10-OD023689. H.L. is supported by the Pioneer Fund Postdoctoral Scholar Award. J.R.E. is an investigator of the Howard Hughes Medical Institute.

**Author contributions** J.R.E., H.L., B.R. and M.M.B. conceived the study. H.L., Q.Z. and J.Z. analysed the data and drafted the manuscript. J.R.E., C.L., B.R., J.R.D., M.M.B., H.Z. and B.T. edited the manuscript. J.R.E., H.L., M.M.B., A.B., B.R., J. Lucero and M.N. coordinated the research. J.R.E., M.M.B., C.L., C.O., M.N., A.P.-D., J.K.O., J. Lucero, R.G.C., J.R.N., J.A., M.K., W.T., A.B. and H.L. generated the snmC-seq3 data. M.M.B., J.R.D., C.O., R.G.C., J.R.N., J.A., M.K., P.B., B.-A.W., A.B. and H.L. generated the sn-m3C-seq data. M.M.B., N.E., S.C., J.R., J. Lee, H.L. and Q.Z. generated the MERFISH data. B.R., Y.E.L. and S.Z. generated the snATAC-seq data. H.Z., K.A.S., B.T. and Z.Y. generated the 10x scRNA-seq, SMART-seq, whole-brain MERFISH data and the whole-mouse-brain transcriptomic cell-type taxonomy and atlas. H.L., H.C., Z.W. and I.S. contributed to the data archive, visualization and cloud infrastructure. J.R.E. and M.M.B. supervised the study.

**Competing interests** J.R.E. serves on the scientific advisory board of Zymo Research. B.R. is a shareholder of Arima Genomics and Epigenome Technologies. H.Z. is on the scientific advisory board of MapLight Therapeutics.

**Additional information**
**Correspondence and requests for materials** should be addressed to Joseph R. Ecker.

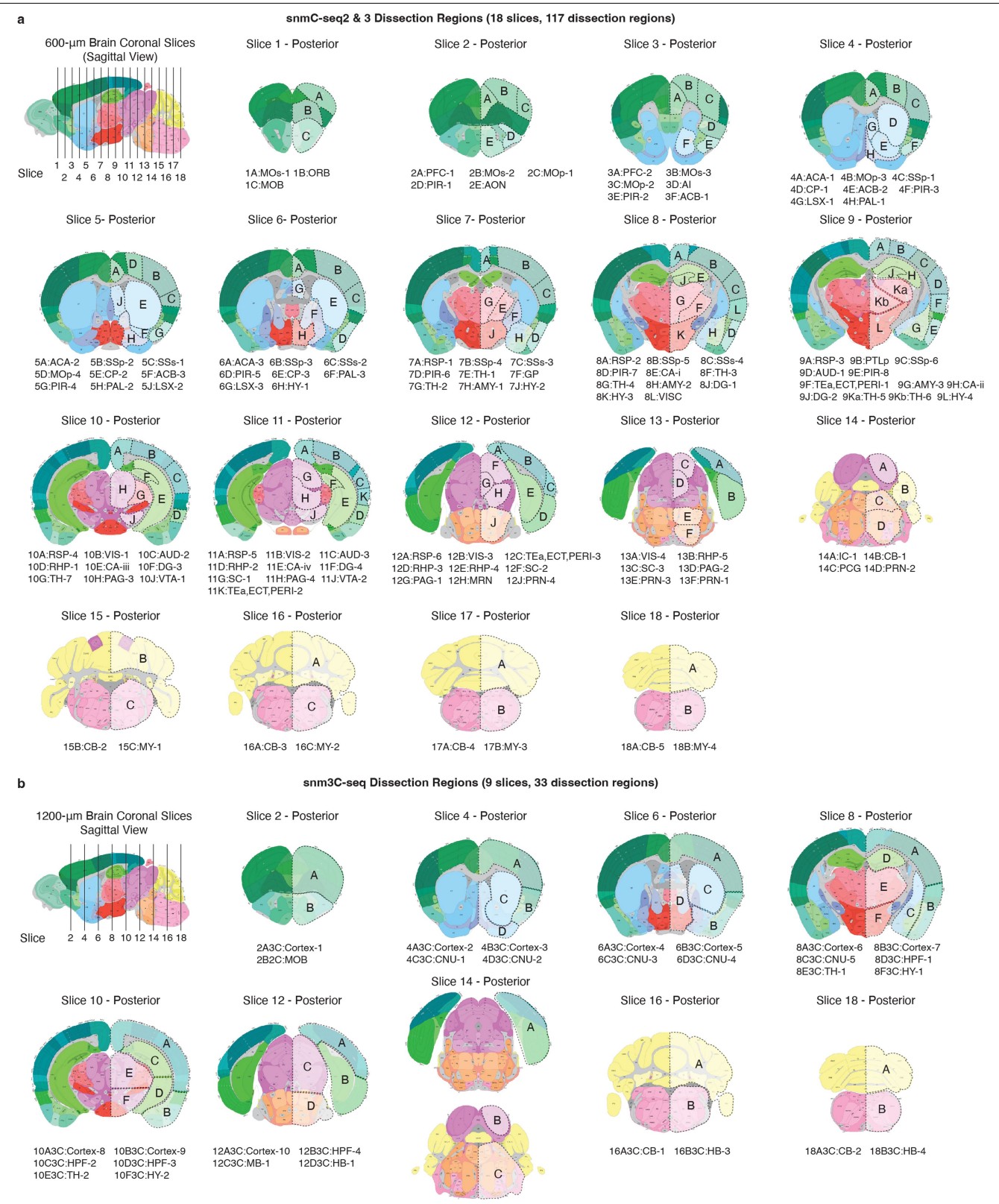

**Extended Data Fig. 1 | Brain dissection regions.** Schematic of brain dissection steps. Each male C57BL/6 mouse brain (age P56) was dissected into 600-µm slices for snmC-seq3 (**a**) and 1,200-µm slices for snm3C-seq3 (**b**). We then dissected brain regions from both hemispheres within a specific slice. Brain atlas images were created based on Wang et al.[3] and © 2017 Allen Institute for Brain Science. Allen Brain Reference Atlas. Available from: atlas.brain-map.org.

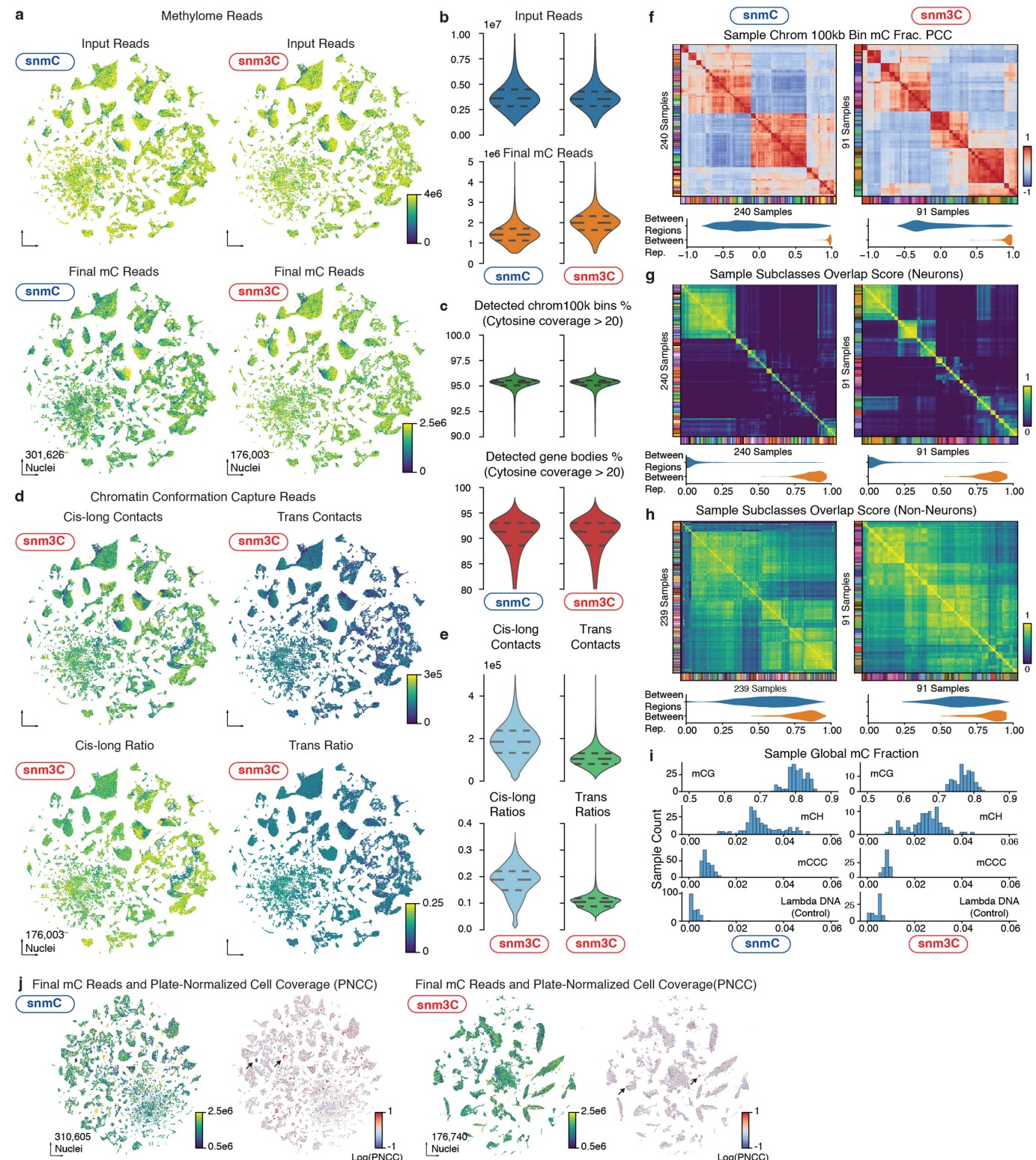

**Extended Data Fig. 2 | Quality Control for snmC and snm3C dataset. a-b**, The number of input reads and final pass QC reads in snmC-seq3 and snm3C-seq shown by t-SNE (**a**) and violin plot (**b**) **c**, The percentage of non-overlapping chromosome 100-kb bins or genes detected per cell in snmC-seq3 and snm3C-seq. Gray lines from top to bottom indicate the 75%, 50%, and 25% quantiles. **d-e**, The number and ratio of cis-long and trans contacts in snm3C-seq, depicted by t-SNE (**d**) and violin plot (**e**). **f**, Heatmap of PCC between the average methylome profiles (mean mCH and mCG fraction of all chromosome 100-kb bins across all cells belonging to a replicate sample). The violin plot below summarizes the

values between replicates within the same brain region or between different brain regions. **g-h**, Pairwise overlap score (measuring co-clustering of two replicates) of neuronal subtypes and (**g**) non-neuronal subtypes (**h**). The violin plots summarize the subtype overlap score between replicates within the same brain region or between different brain regions. **i**, Distribution of the mCG, mCH, mCCC, and Lambda DNA fraction (non-conversion rate) at sample level in snmC-seq3 and snm3C-seq. **j**, Pre-clustering t-SNE of snmC and snm3C dataset colored by final mC reads and plate-normalized cell coverage. Arrows indicate typical low-quality clusters filtered out from the further analysis.

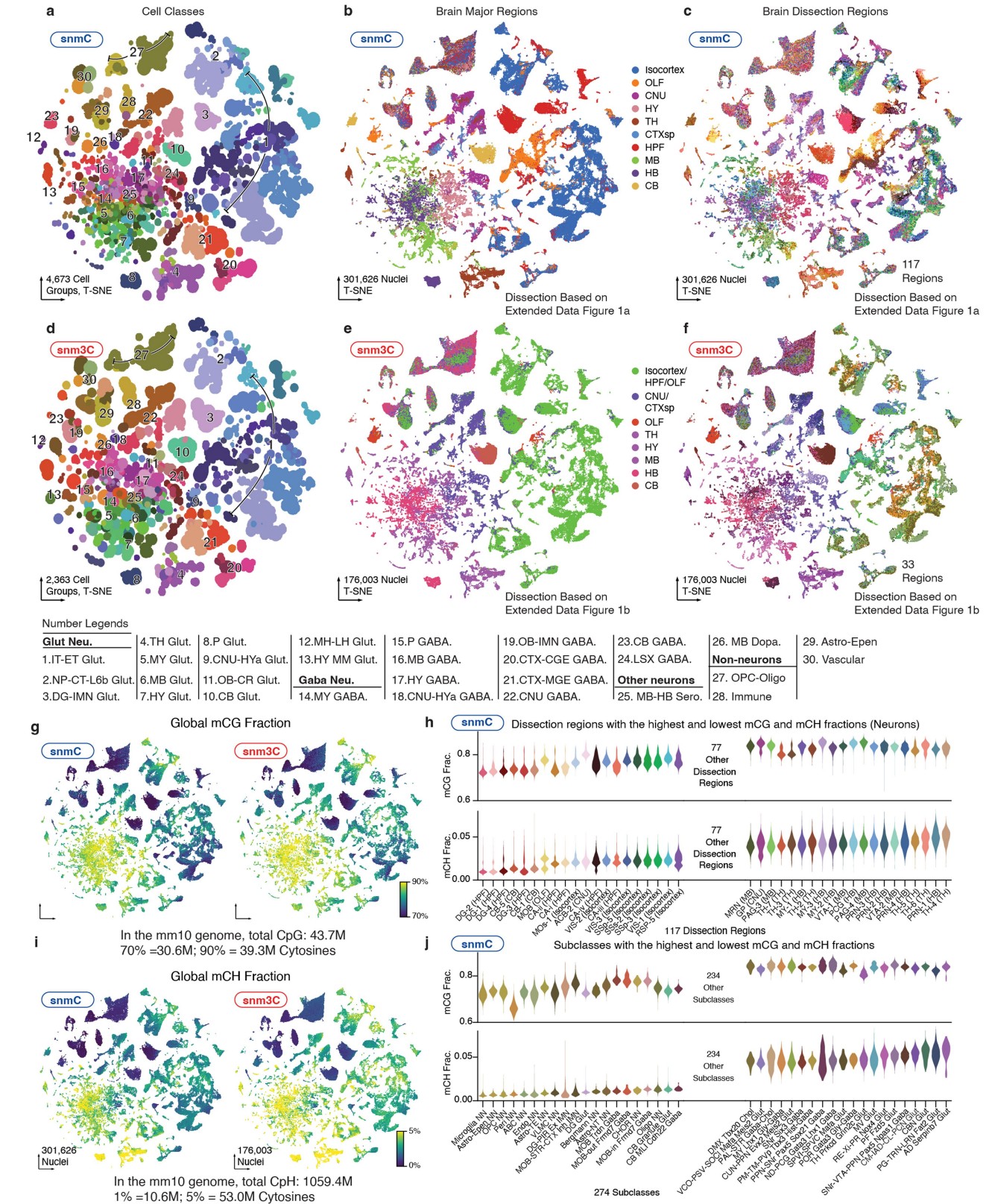

**Extended Data Fig. 3 | Metadata of snmC-seq and snm3C-seq dataset.**
**a-c**, t-SNE of snmC-seq color by cell subclass (**a**), major regions (**b**), and dissection regions (**c**). **d-f**, t-SNE of snm3C-seq color by cell subclass (**d**), major regions (**e**), and dissection regions (**f**). **g,h**, Cell-level t-SNE of snmC-seq and snm3C-seq color by global mCG (**g**) and global mCH (**h**) fraction. **i**, The average global mCG and

mCH fractions for neurons in different dissection regions. Regions are ordered by the global mCH fractions, and only the top and bottom 20 regions are shown. **j**, The average global mCG and mCH fractions for all cell subclasses. Subclasses are ordered by the global mCH level, and only the top and bottom 20 subclasses are shown.

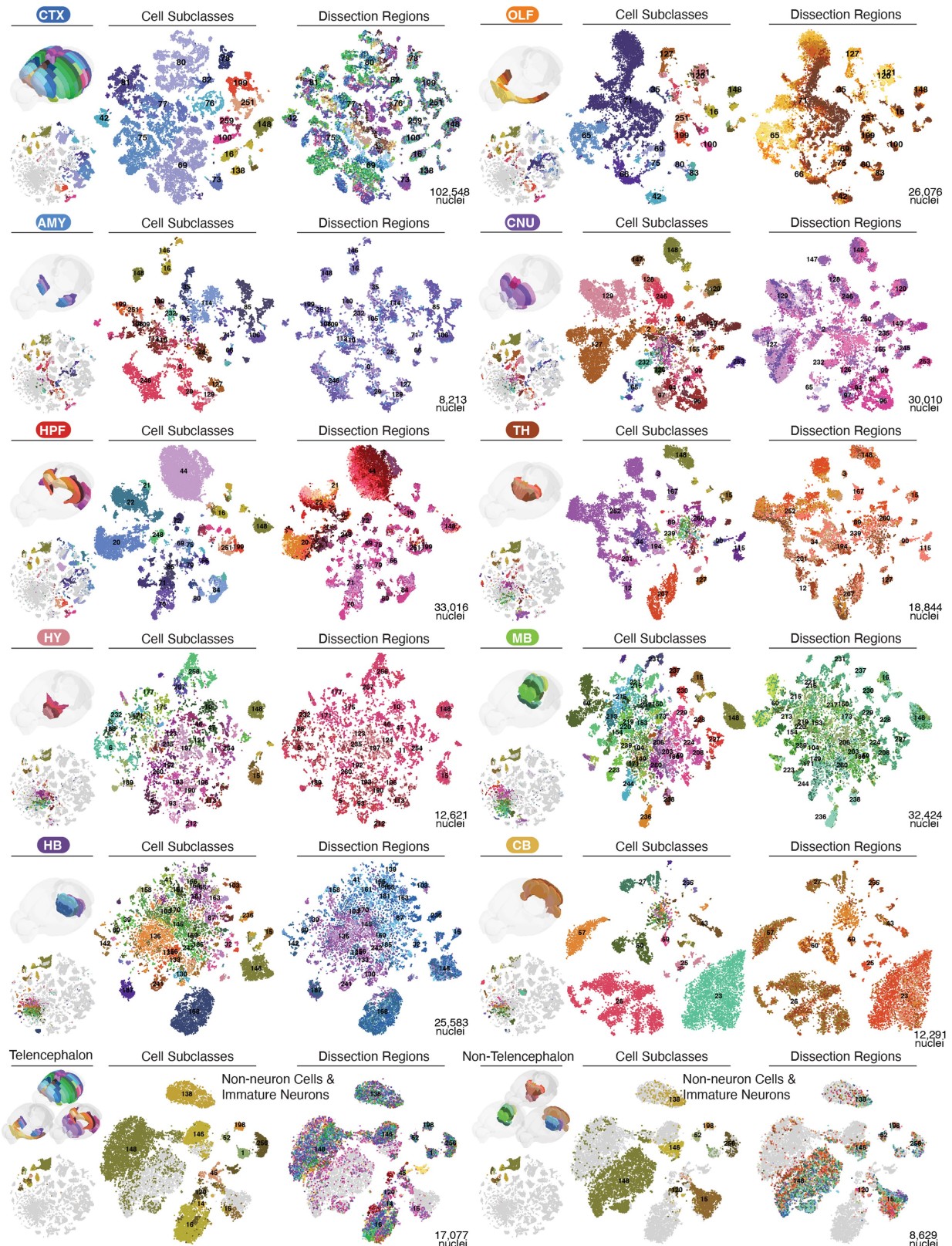

**Extended Data Fig. 4 | t-SNE embedding by major regions.** This figure groups cells by major regions (first five rows), including isocortex (CTX), olfactory bulb (OLF), amygdala (AMY), cerebral nuclei (CNU), hippocampus (HPF), thalamus (TH), hypothalamus (HY), midbrain (MB), hindbrain (HB), and cerebellum (CB). Each section comprises three columns. The left column displays the CCF-registered 3D brain dissection regions and the corresponding cell on the whole brain t-SNE. The middle and right columns show the t-SNE embedded by cells from this major region, colored by cell subclasses and dissection regions, respectively. The numbers on the t-SNE plot indicate the cell subclass ID, which refers to in Supplementary Table 4. The final row groups non-neuron cells into two sections based on telencephalon and non-telencephalon dissection regions.

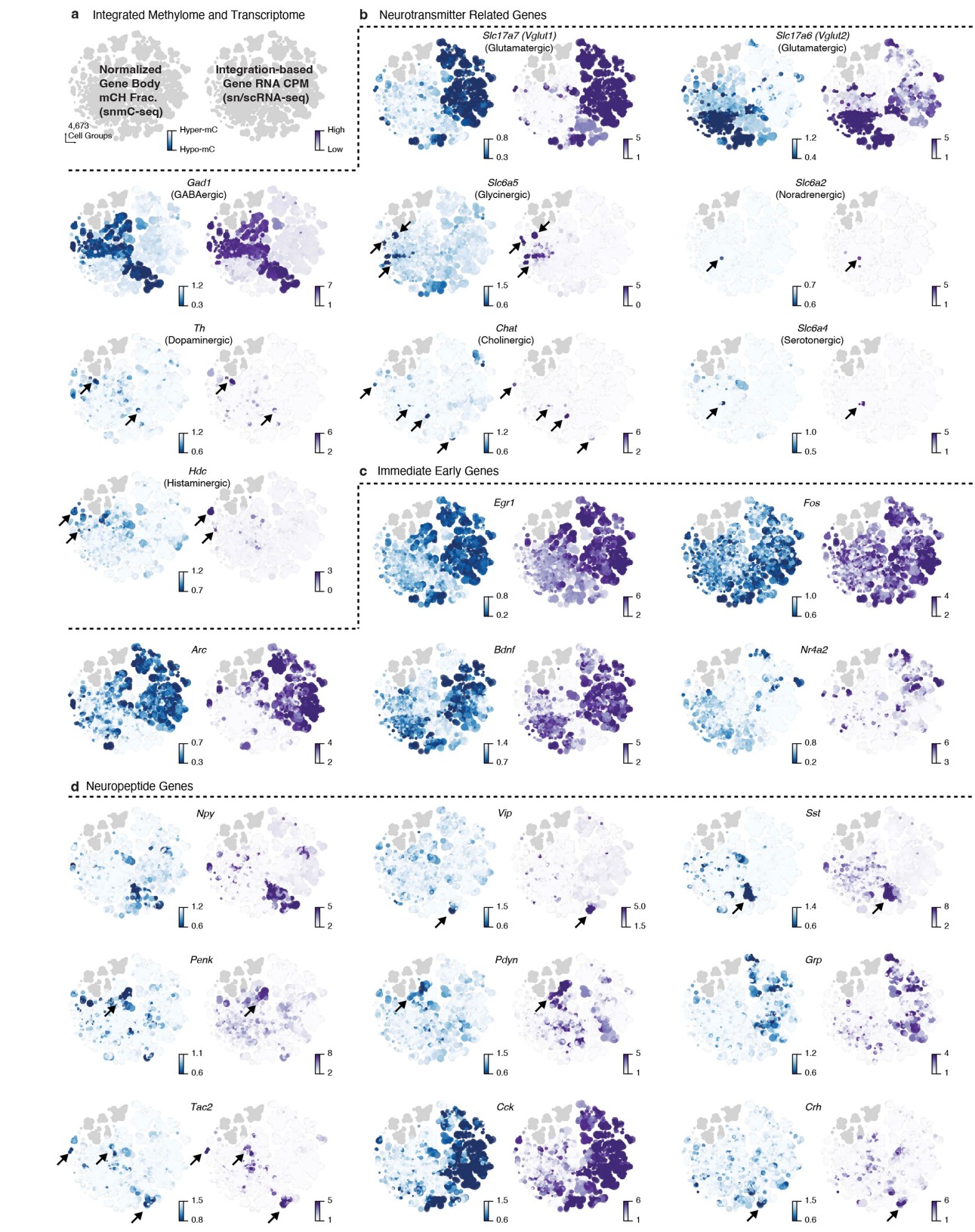

**Extended Data Fig. 5** | See next page for caption.

**Extended Data Fig. 5 | Example genes illustrating high-granularity correspondence between methylome and transcriptome.** All t-SNE embeddings in this figure are based on the methylome clustering shown in Fig. 2a. Gene expression of non-neuronal cell subclasses is not plotted here. **a**. Schematic representation of the normalized gene body mCH fraction (left panel) and RNA CPM value (right panel) at the cell-group-centroids t-SNE plot for each gene. **b**. Pairwise plots of neurotransmitter-related genes. These genes provide crucial information about cell type identities and display a highly similar specificity between gene body mCH fractions and mRNA expression. Genes include *Slc17a7* and *Slc17a6* for glutamatergic, *Gad1* for GABAergic, *Slc6a5* for glycinergic, *Slc6a2* for noradrenergic, *Th* for dopaminergic, *Chat* for cholinergic, *Slc6a4* for serotonergic, and *Hdc* for histaminergic. **c**. Pairwise plots of immediate early genes (*Fos*, *Egr1*, *Arc*, *Bdnf*, *Nr4a2*) are also expressed in many adult brain cell types[6,8]. Their expression levels are also anti-correlated with mCH fractions. **d**. Another gene category includes neuropeptides (*Npy*, *Vip*, *Sst*, *Penk*, *Pdyn*, *Grp*, *Tac2*, *Cck*, *Crh*), many of which are canonical cell type markers with vital signaling functions[97]. Their specificity is detectable in the gene body mCH that aligns with transcription.

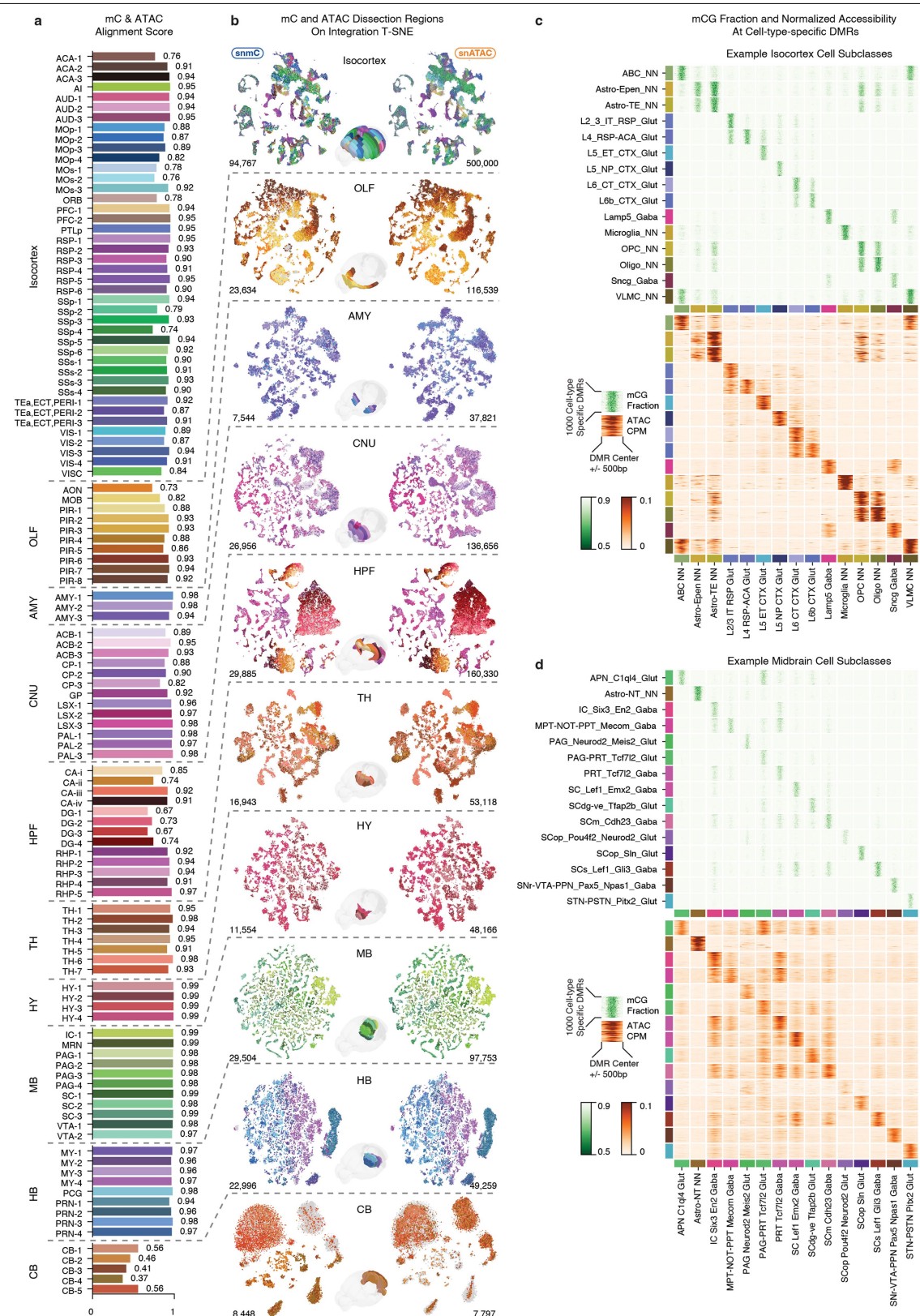

**Extended Data Fig. 6 | Integration of snATAC-seq and snmC-seq3 data.**
**a,** Barplot displays the alignment scores of each dissection region calculated in the low dimensional space of snATAC-seq and snmC-seq integration. **b,** t-SNE shows the co-embedding of snmC-seq and snATAC-seq data, grouped by major regions and colored by dissection regions. **c-d,** Heatmap visualization of 15 ×15 small heatmaps. Each small heatmap represents the mCG fractions (green) and the corresponding accessibility level of 1,000 cell-type-specific CG-DMRs. Columns display hypo-DMRs of that cell subclass while rows show their mCG fraction/ATAC CPM values. Take the top-right mini heatmap as an example, rows represent VLMC_NN hypo-DMRs, with color indicating mCG fraction in ABC_NN. Cell subclasses from isocortex (**c**) and midbrain (**d**) are shown as examples.

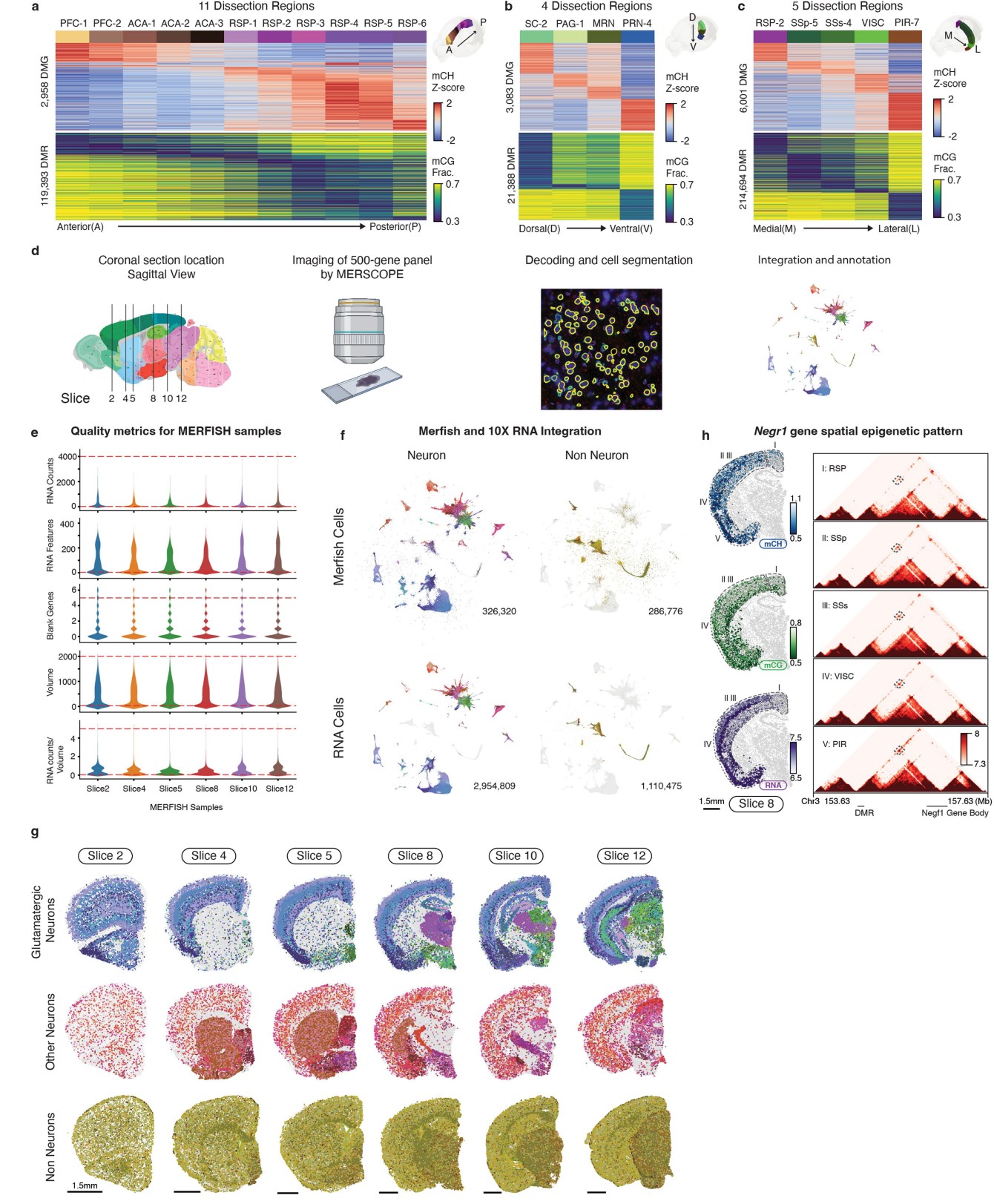

**Extended Data Fig. 7 |** See next page for caption.

**Extended Data Fig. 7 | MERFISH data processing and annotation. a-c**, Spatial methylation patterns of DMGs (genes with differential mCH levels on gene body ± 2 kb among different brain regions) and DMRs across three brain axes (anterior to posterior (**a**), dorsal to ventral (**b**), medial to lateral (**c**). **d**, Workflow illustrating the generation of MERFISH data, including sample preparation, imaging, and data analysis steps. **e**, Quality control assessment for each MERFISH sample, where the red lines represent the filtering cutoff for various quality metrics, including RNA total counts, RNA feature counts, blank gene number, cell volume (μm³), and RNA counts per volume. **f**, Integration t-SNE plot of MERFISH and scRNA dataset[6] color by cell subclasses. **g**, MERFISH cells colored by cell subclasses, with labels obtained from the integration with the RNA dataset. From top to bottom, the cells are displayed by glutamatergic neurons, other neurons, and non-neurons. **h**, Spatial epigenetic patterns of *Negr1* and its associated DMRs. Brain slices in the left column are color-coded by normalized gene body mCH fraction, mCG fraction of the DMR (chr3:154,927,600-154,929,099), and RNA expression. The right column displays the normalized contacts heatmap between the DMR and gene. Microscope objective and slide in d were created using BioRender (www.biorender.com).

**a** GLUT Neurons Color by Dissectin Region

Slice23　Slice35　Slice 47　Slice 51

Slice63　Slice67　Slice73　Slice 75

Slice 89　Slice 95　Slice 101　Slice 123

**b** Slice 67 -Color by Cell Subclass

CA1-ProS Glut　DG Glut　L2/3 IT RSP Glut　L2/3 IT CTX Glut　L4/5 IT CTX Glut

L5 IT CTX Glut　L6 CT CTX Glut　STN-PSTN Pitx2 Glut　TH Prkcd Grin2c Glut　ZI Pax6 Gaba

**Extended Data Fig. 8 | Integration of snmC-seq and AIBS whole-mouse-brain MERFISH datasets. a**, Imputed spatial locations of glutamatergic neurons colored by dissection regions. 12 coronal slices were selected to represent 51 total MERFISH slices. Additional data for the remaining slices can be accessed through our interactive browser: https://mousebrain.salk.edu/dynamic_browser. **b**, AIBS MERFISH Slice 67 color by individual cell subclasses.

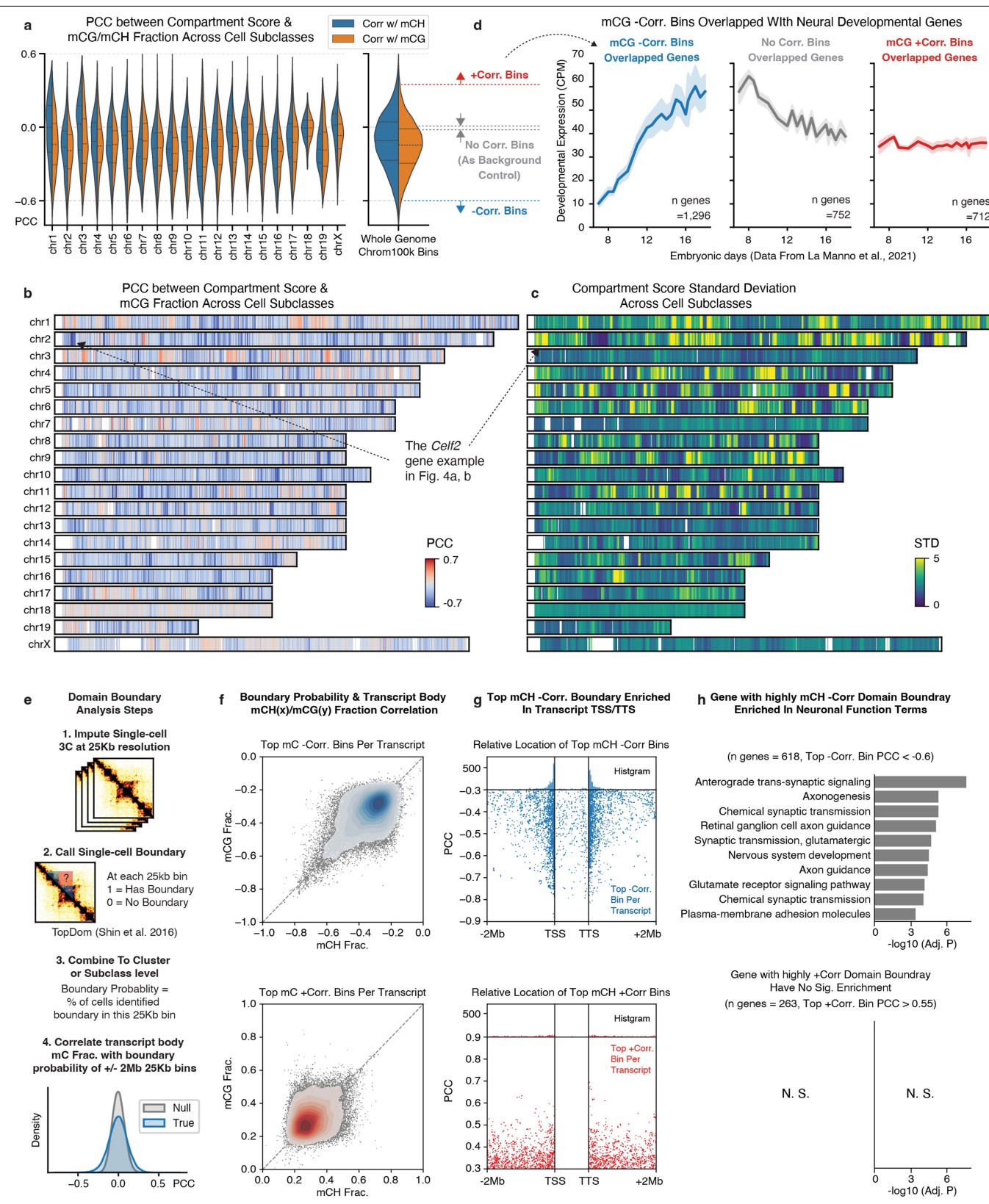

**Extended Data Fig. 9** | See next page for caption.

**Extended Data Fig. 9 | Chromatin conformation analysis at compartment and domain level. a**, PCC between compartment score and mCG (orange)/ mCH (blue) fractions of all 100 kb bins on each chromosome (left panel) or whole genome (right panel). The dot lines inside each violin plot are 75%, 50%, and 25% quantiles from top to bottom. **b-c**, chromosome 1-D heatmaps show PCC between compartment score and mCG fraction (**b**) and the compartment score STD across cell subclasses (**c**) for each chromosome at a 100-Kb resolution. Arrows indicate the location of the *Celf2* gene used as an example in Fig. 4a,b. **d**, The line plot (mean±s.d.) shows the developmental gene expression level among subtypes defined in La Manno et al.[37] across embryonic days. The genes in each subpanel are selected by overlapping with top negatively correlated (left), positively correlated (right), or uncorrelated (middle) chrom100k bins in (**a**). **e**, Workflow for gene body domain boundary analysis. **f**, The scatter plots of the most negatively (top) or positively (bottom) correlated boundary to each long gene transcript. Both the x and y axis is the PCC between 25 Kb bin boundary probability and transcript body mCH (x-axis) or mCG (y-axis) fractions. **g**, The scatterplot shows the location of each long gene transcript's most negatively (top) or positively (bottom) correlated boundary. The y-axis is the PCC between the 25 Kb bin boundary probabilities and transcript body mCH fractions; the x-axis is the relative genome location to the transcripts. **h**, Functional enrichment for genes associated with negatively correlated domain boundaries (upper) or positively correlated boundaries (lower). Adjusted p-values obtained from one-side Fisher's exact test after FDR correction.

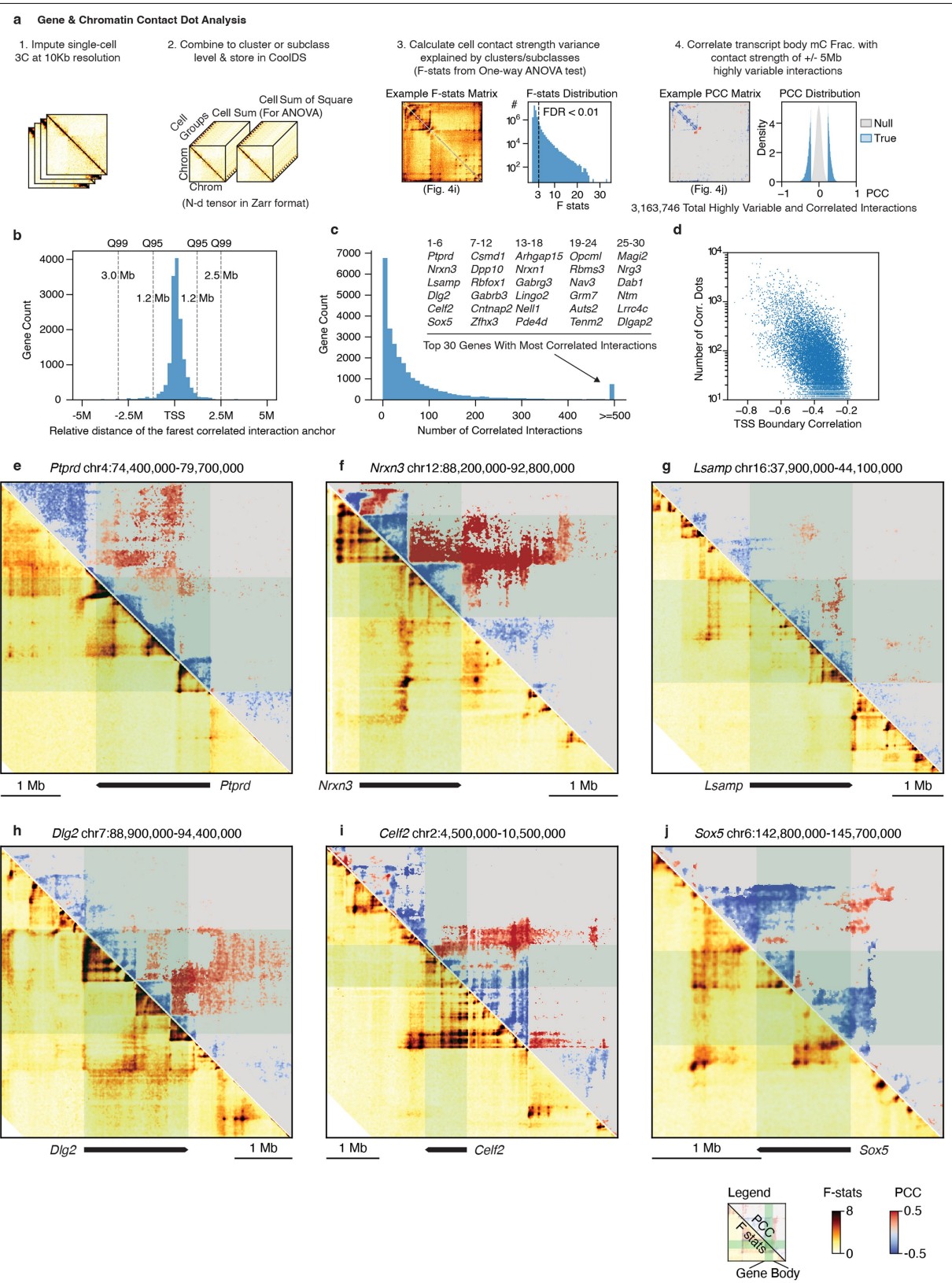

**Extended Data Fig. 10 | Correlation between gene expression and chromatin contacts. a**, Workflow for highly variable and gene correlated interaction analysis. **b**, The distribution of the distance between the furthest correlated interaction and gene TSS. Q95 and Q99 stand for the quantile of all interactions ordered by the distance to TSS. **c**, Distribution of the number of highly variable and correlated interactions per gene; top 30 gene names are listed. **d**, Scatterplot shows each gene's number of correlated interactions (y-axis) and TSS boundary probability correlation (x-axis, PCC between mCH and TSS boundary probability,

from Extended Data Fig. 9e). **e-j**, Compound heatmaps display the chromatin conformation landscape of megabase-long genes, including *Ptprd* (**e**), *Nrxn3* (**f**), *Lsamp* (**g**), *Dlg2* (**h**), *Celf2* (**i**), and *Sox5* (**j**). For each panel, green rectangles indicate the location of the gene body, the lower triangle shows the F statistics from ANOVA analysis analyzing the variance of contact strength across all cell subclasses (similar to Fig. 4i), and the upper triangle shows the PCC between contact strength and mCH fraction (similar to Fig. 4j).

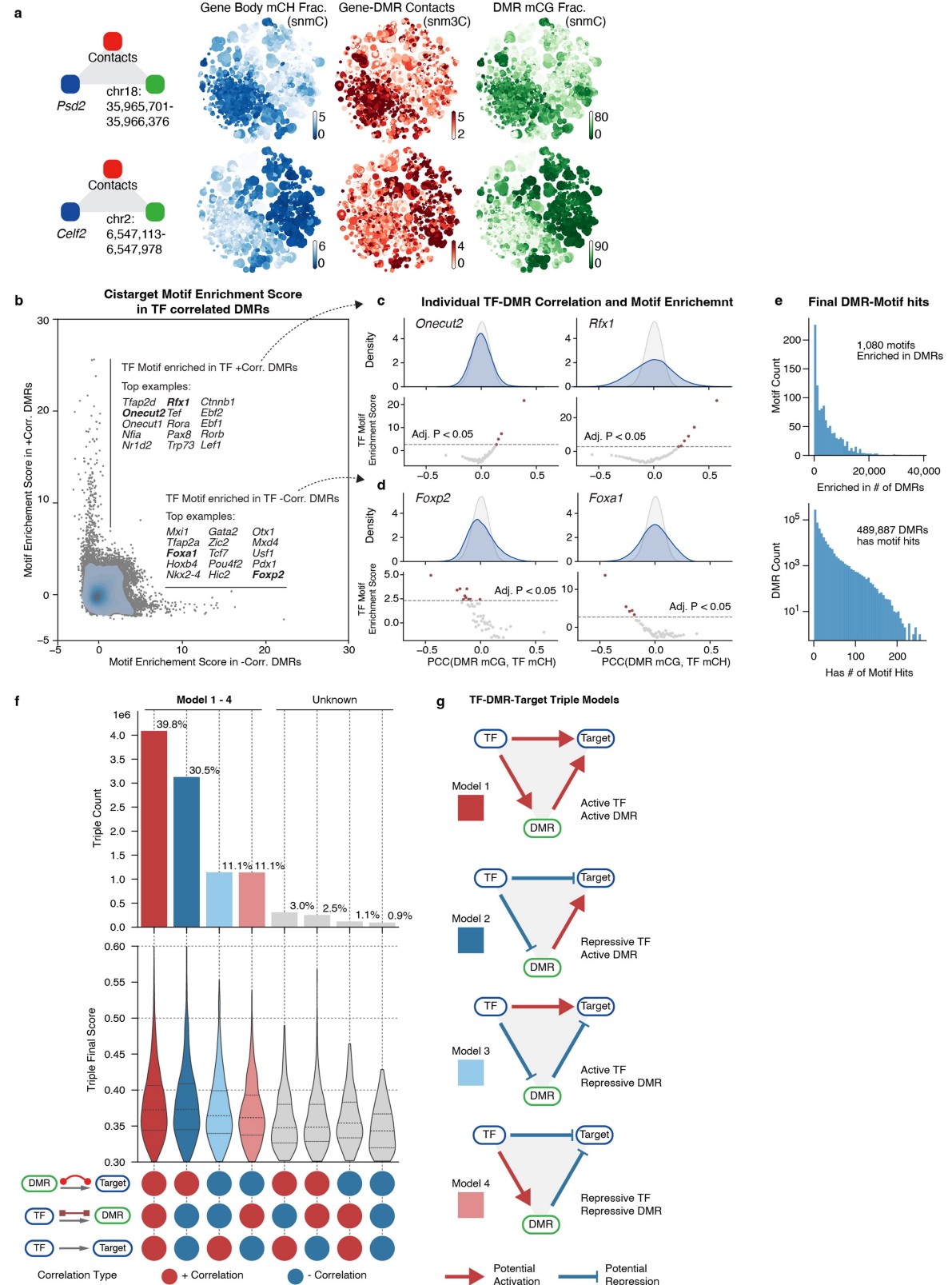

**Extended Data Fig. 11** | See next page for caption.

**Extended Data Fig. 11 | Construction of TF-DMRs-Target regulatory networks. a**, Schematic of the DMR-Target edge for Psd2 (top row) and Celf2 (bottom row). From left to right, the t-SNE plot is colored by gene mCH fraction, gene-DMR contacts, and DMR mCG fraction. **b**, Scatterplot shows the motif enrichment scores in negatively correlated DMRs (x-axis) and positively correlated DMRs (y-axis) for each TF. The top TFs with the highest motif enrichment scores are listed. Blue contours are the kernel density of the dots. **c**-**d**, Example TFs with motifs enriched in positively correlated DMRs or negatively correlated DMRs are shown in more detail (similar to Fig. 5f). The *Onecut2* and *Rfx1* gene (**c**) are examples of having motifs enriched in positively correlated DMRs, the Foxp2 and Foxa1 gene (**d**) are examples of having motifs enriched in negatively correlated DMRs. Adjusted p-values obtained from the z-test of the motif enrichment score from pycistarget[43] (Method) after FDR correction. **e**, The top histogram shows the distribution of the number of DMRs each motif is enriched in. The bottom histogram shows the distribution of the number of motif occurrences each DMR has. **f**, The TF-DMR-Target triples are separated into eight categories (columns) based on their PCC sign between Gene-DMR, TF-DMR, and TF-Gene. The top bar plot is the triple distribution in each category. The middle violin plot is the triple final score distribution within each category. Lines inside the violin plot are 25%, 50%, and 75% quantiles, respectively. The bottom dots show the correlation sign combination of each category. Column colors match the schematic in (**f**). **g**, The schematic displays the potential regulatory model for the four most common (based on **e**) TF-DMR-Target triple categories.

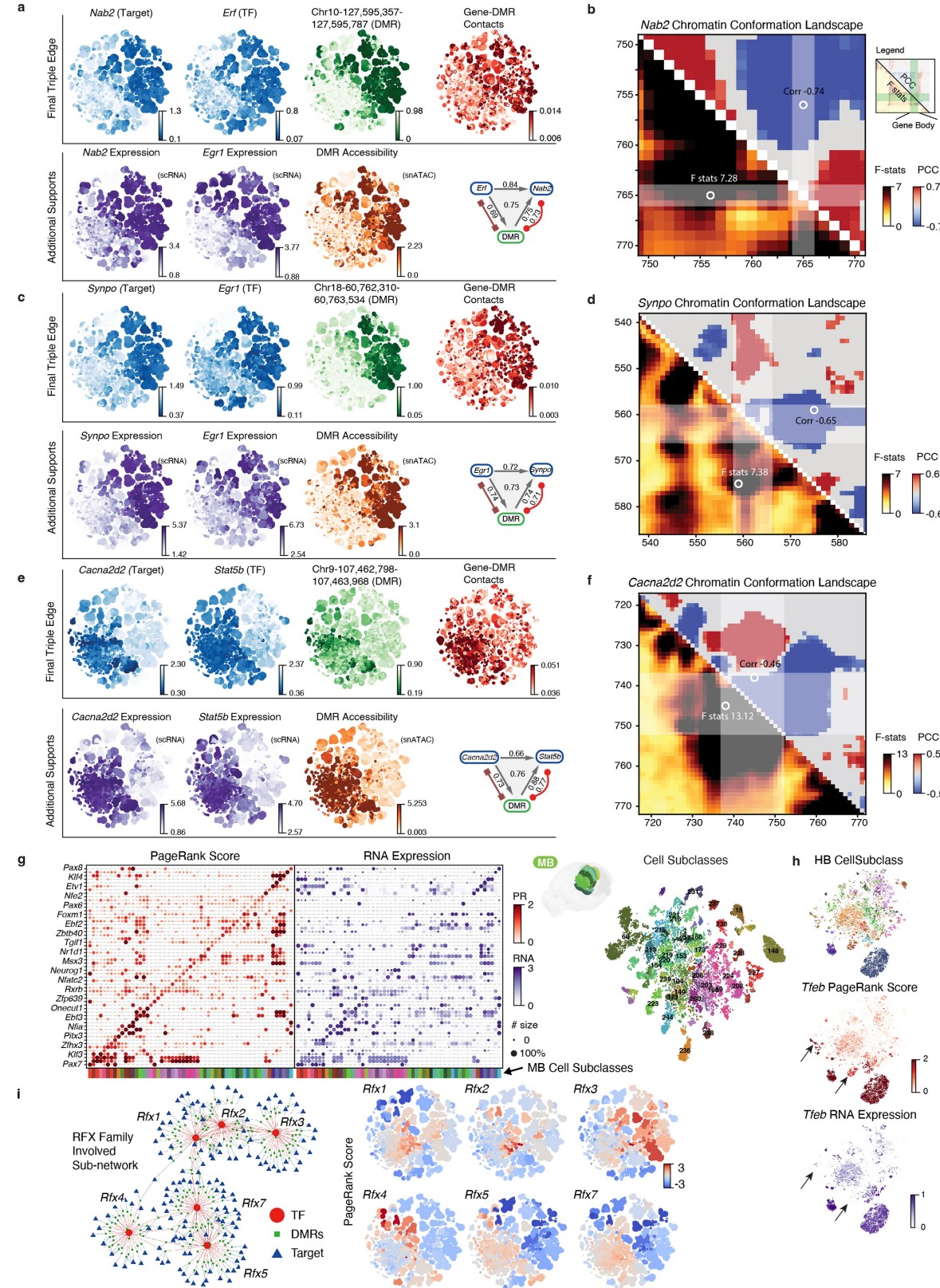

**Extended Data Fig. 12 |** See next page for caption.

**Extended Data Fig. 12 | TF-DMR-Gene triple predict TF and gene relationships. a-f**, Example TF-DMR-Target triple, including 1: *Erf* (TF), *Nab2* (target) and DMR (Chr10:127,595,357-127,595,787) (**a-b**); 2: *Egr1* (TF), *Synpo* (target) and DMR (Chr18:60,762,310-60,763,534) (**c-d**); 3: *Cacna2d2* (TF), *Stat5b* (target) and DMR (Chr9:107,462,798-107,463,968) (**e-f**); For each example, left are t-SNE plot colored by the mCH fraction (blue) or RNA level (purple) for target and TF; mCG fraction (green) and chromatin accessibility (orange) for DMR; and gene-DMR contact score (red) (**a,c,e**). The compound heatmaps on the right show the chromatin landscape of target genes, including *Nab2* (**b**), *Synpo* (**d**), and *Cacna2d2* (**f**); the layout is similar to Extended Data Fig. 10e–j.

**g**, The dot plots represent TF's normalized PageRank Score and RNA expression for cell subclasses in the hindbrain (MB). Red dots are colored and sized by PageRank Score. Purple dots are colored by RNA CPM, sized by the percentage of cells in that subclass expressing this gene. Right, the t-SNE plot of snmC-seq cells from MB colored by dissection region and the CCF-registered 3D brain dissection regions. **h**, From top to bottom, t-SNE plot colored by HB cell subclasses, *Tfeb* PageRank Score and *Tfeb* RNA expression. Arrows point to two cell subclasses with high PageRank score but low RNA level. **i**, Left, schematic of RFX family sub-networks. Right, t-SNE plot color by normalized PageRank Score of RFX family genes.

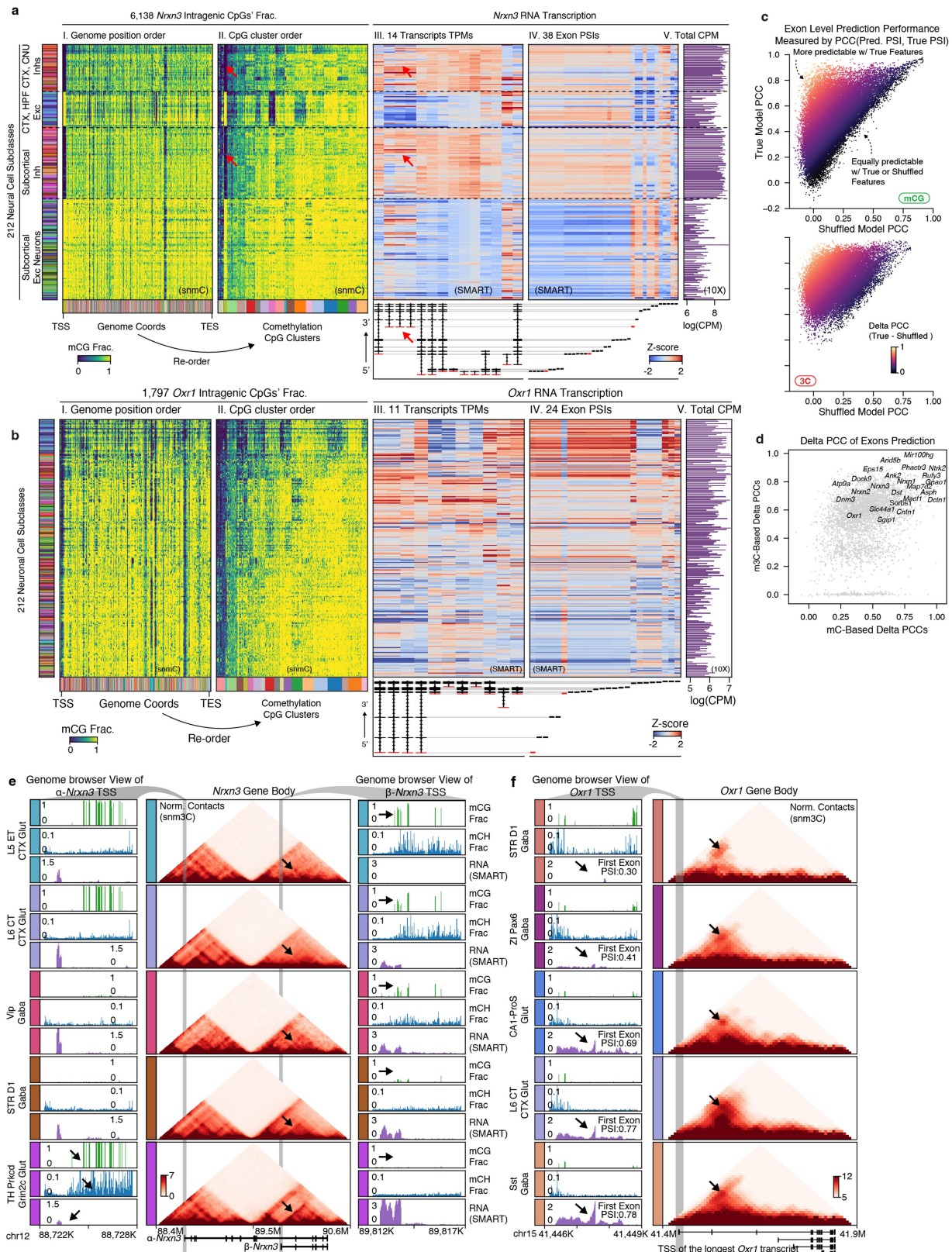

**Extended Data Fig. 13** | See next page for caption.

**Extended Data Fig. 13 | Epigenetic heterogeneity and gene exon usage.**
**a**, Compound heatmaps illustrate the similarity between the *Nrxn3* intragenic methylation heterogeneity and alternative isoform expression patterns. Rows are neuron cell subclasses. I, mCG fraction of all 6,138 CpG sites of *Nrxn3* gene with columns ordered by original genome coordinates (bottom colors are CpG clusters from heatmap II). II, mCG fraction of CpG sites re-ordered by their CpG clusters (bottom colors) based on subclasses methylation pattern. Heatmap III and Heatmap IV show the TPM of 14 highly variable transcripts and PSI of 38 highly variable exons of *Nrxn3*, quantified with the SMART-seq dataset. All values are z-score normalized across cell subclasses. The *Nrxn3* transcript structures and exon locations are indicated at the bottom plots. Red arrows point to beta-*Nrxn3* transcripts and one associated CpG cluster. Heatmap V shows the *Nrxn3* gene log(CPM) in scRNA-seq (10X) data. **b**, Compound heatmaps illustrate the similarity between the *Oxr1* intragenic methylation heterogeneity and alternative isoform expression patterns. Rows are neuron cell subclasses. I, mCG fraction of all 1,797 CpG sites of *Oxr1* gene with columns ordered by original genome coordinates (bottom colors are CpG clusters from heatmap II). II, mCG fraction of CpG sites re-ordered by their CpG clusters (bottom colors) based on

subclasses methylation pattern. Heatmap III and Heatmap IV show the TPM of 11 highly variable transcripts and PSI of 24 highly variable exons of *Oxr1*, quantified with the SMART-seq dataset. All values are z-score normalized across cell subclasses. The *Oxr1* transcript structures and exon locations are indicated at the bottom plots. Heatmap V shows the *Oxr1* gene log(CPM) in scRNA-seq (10X) data. **c**, Scatterplot shows the PCC between predicted PSI and true PSI for each highly variable exon (dot), using methylation features (left) and chromatin contact interactions (right) to predict. **d**, Scatterplot shows the delta PCC in mC models (x-axis) and m3C models (y-axis) for highly variable exons (dot). Top exons with large delta PCC are listed by their corresponding gene names. **e**. Genome browser view of intragenic epigenetic and isoform diversity of the *Nrxn3* gene in five cell subclasses (rows). The middle heatmaps are normalized contact strengths of the *Nrxn3* gene locus, with arrows pointing to strips over the beta-*Nrxn3* transcript body. The zoom-in panels show alpha-*Nrxn3*'s (left) and beta-*Nrxn3*'s (right) TSS region, with mCG fraction (green), mCH fraction (blue), and SMART RNA (bottom) expression tracks. **f**, Similar to **e**, showing the corresponding intragenic epigenetic and isoform diversity in the *Oxr1* gene.

# Reporting Summary

## Statistics

For all statistical analyses, confirm that the following items are present in the figure legend, table legend, main text, or Methods section.

| n/a | Confirmed | |
|---|---|---|
| ☐ | ☒ | The exact sample size (*n*) for each experimental group/condition, given as a discrete number and unit of measurement |
| ☐ | ☒ | A statement on whether measurements were taken from distinct samples or whether the same sample was measured repeatedly |
| ☐ | ☒ | The statistical test(s) used AND whether they are one- or two-sided<br>*Only common tests should be described solely by name; describe more complex techniques in the Methods section.* |
| ☐ | ☒ | A description of all covariates tested |
| ☐ | ☒ | A description of any assumptions or corrections, such as tests of normality and adjustment for multiple comparisons |
| ☐ | ☒ | A full description of the statistical parameters including central tendency (e.g. means) or other basic estimates (e.g. regression coefficient) AND variation (e.g. standard deviation) or associated estimates of uncertainty (e.g. confidence intervals) |
| ☐ | ☒ | For null hypothesis testing, the test statistic (e.g. *F*, *t*, *r*) with confidence intervals, effect sizes, degrees of freedom and *P* value noted<br>*Give P values as exact values whenever suitable.* |
| ☒ | ☐ | For Bayesian analysis, information on the choice of priors and Markov chain Monte Carlo settings |
| ☒ | ☐ | For hierarchical and complex designs, identification of the appropriate level for tests and full reporting of outcomes |
| ☐ | ☒ | Estimates of effect sizes (e.g. Cohen's *d*, Pearson's *r*), indicating how they were calculated |

*Our web collection on statistics for biologists contains articles on many of the points above.*

## Software and code

Policy information about availability of computer code

| | |
|---|---|
| Data collection | BD Influx Sortware v1.2.0.142 (flow cytometry), Freedom EVOware v2.7 (library preparation), Illumina MiSeq control software v3.1.0.13 and NovaSeq 6000 control software v1.6.0/RTA v3.4.4 (sequencing), Olympus cellSens Dimension 1.8 (image acquisition) |
| Data analysis | cemba-data, v1.6.8; cutadapt, v2.10; bismark, v0.20; bowtie2, v2.3; samtools, v1.9; Picard, v3.0.0;<br><br>Mapping pipeline for snmC-seq3 and snm3C-seq is available at https://hq-1.gitbook.io/mc/. Single-cell DNA methylome data analysis tools are available at ALLCools (v1.0.8) python package, https://lhqing.github.io/ALLCools/intro.html; Single-cell chromatin conformation data analysis tools are available at the scHiCluster (v1.3.2) python package, https://github.com/zhoujt1994/scHiCluster. Other codes and Jupyter Notebooks related to results and method sections are shared in this GitHub repository: https://github.com/lhqing/wmb2023.<br>Brain dissection 3D image created by ITK-SNAP (v4.0.0)<br>MERSCOPE data analysis were performed with MERSCOPE Instrument Software (v2023-01) |

For manuscripts utilizing custom algorithms or software that are central to the research but not yet described in published literature, software must be made available to editors and reviewers. We strongly encourage code deposition in a community repository (e.g. GitHub). See the Nature Portfolio guidelines for submitting code & software for further information.

## Data

Policy information about availability of data

All manuscripts must include a data availability statement. This statement should provide the following information, where applicable:

- Accession codes, unique identifiers, or web links for publicly available datasets
- A description of any restrictions on data availability
- For clinical datasets or third party data, please ensure that the statement adheres to our policy

The snmC-seq2/3 single-cell sequencing data and MERFISH dataset are accessible through the Neuroscience Multi-omic Data (NeMO) Archive (https://assets.nemoarchive.org/dat-sig83t9). The snm3C-seq single-cell sequencing data are accessible through NeMO (https://assets.nemoarchive.org/dat-sig83t9) and GEO (GSE213262). The MERFISH dataset will be accessible through GEO. The whole-brain snATAC-seq dataset is shared by Zu et al11. The whole-brain scRNA-seq MERFISH, and SMART-seq dataset is shared by Yao et al6. All the processed data related to results and method sections are shared in this GitHub repository: https://github.com/lhqing/wmb2023. Allen Brain Reference Atlas and Common Coordinate Framework is accessed through Wang et al3. A detailed description of the data availability is provided at https://mousebrain.salk.edu/download.

## Human research participants

Policy information about studies involving human research participants and Sex and Gender in Research.

| | |
|---|---|
| Reporting on sex and gender | This study did not involve human research participants. |
| Population characteristics | This study did not involve human research participants. |
| Recruitment | This study did not involve human research participants. |
| Ethics oversight | This study did not involve human research participants. |

Note that full information on the approval of the study protocol must also be provided in the manuscript.

# Field-specific reporting

Please select the one below that is the best fit for your research. If you are not sure, read the appropriate sections before making your selection.

☒ Life sciences ☐ Behavioural & social sciences ☐ Ecological, evolutionary & environmental sciences

For a reference copy of the document with all sections, see nature.com/documents/nr-reporting-summary-flat.pdf

# Life sciences study design

All studies must disclose on these points even when the disclosure is negative.

| | |
|---|---|
| Sample size | No statistical methods were used to predetermine sample size. We empirically determined to use two to three biological experiments for all single-cell epigenomic experiments to achieve minimum reproducibility for this large-scale project. For snmC-seq, at least 3,072 nuclei (eight 384-well plates) from each dissected region (1,536 nuclei from each replicate). For snm3C-seq, at least 6,144 nuclei (eight 384-well plates) from each dissected region (3,072 nuclei from each replicate). The sample size allowed us to obtain high coverage methylome and 3D genome for thousands of brain cell clusters, and perform confident downstream analyses. |
| Data exclusions | Primary quality control for DNA methylome cells was (1) overall mCCC level < 0.05; (2) overall mCH level < 0.2; (3) overall mCG level > 0.5; (4) total final reads > 500,000 and < 10,000,000; and (5) Bismarck mapping rate > 0.5. Additionally, we calculated lambda DNA spike-in methylation levels to estimate each sample's non-conversion rate. All samples demonstrated a low non-conversion rate (< 0.01). For the 3C modality in snm3C-seq cells, we also required cis-long-range contacts (two anchors > 2500 bp apart) > 50,000. |
| Replication | Each dissected region is represented by 2-3 replicates, obtained from pooling the same region from at least six animals. All replications were successful. |
| Randomization | Randomization is not applicable, since the cells collected are random by nature. |
| Blinding | Blinding is not applicable, since all data are collected from male C57BL/6J mice at the age of P56 when generating this reference brain atlas. |

# Reporting for specific materials, systems and methods

We require information from authors about some types of materials, experimental systems and methods used in many studies. Here, indicate whether each material, system or method listed is relevant to your study. If you are not sure if a list item applies to your research, read the appropriate section before selecting a response.

## Materials & experimental systems

| n/a | Involved in the study |
|-----|----------------------|
| ☐ | ☒ Antibodies |
| ☒ | ☐ Eukaryotic cell lines |
| ☒ | ☐ Palaeontology and archaeology |
| ☐ | ☒ Animals and other organisms |
| ☒ | ☐ Clinical data |
| ☒ | ☐ Dual use research of concern |

## Methods

| n/a | Involved in the study |
|-----|----------------------|
| ☒ | ☐ ChIP-seq |
| ☐ | ☒ Flow cytometry |
| ☒ | ☐ MRI-based neuroimaging |

# Antibodies

| | |
|---|---|
| Antibodies used | AlexaFluor488-conjugated anti-NeuN antibody (MAB377X, Millipore, A60, monoclonal, 1:500 dilution) |
| Validation | All antibodies have been previously published for use in immunohistochemistry and flow cytometry experiments. See vendor's page here: https://www.emdmillipore.com/US/en/product/Anti-NeuN-Antibody-clone-A60-Alexa-Fluor488-conjugated,MM_NF-MAB377X |

# Animals and other research organisms

Policy information about studies involving animals; ARRIVE guidelines recommended for reporting animal research, and Sex and Gender in Research

| | |
|---|---|
| Laboratory animals | Adult (P56) C57BL/6J male mice were purchased from Jackson Laboratories at seven weeks of age and maintained in the Salk animal barrier facility on 12-hour dark-light cycles with food ad-libitum for up to 10 days (Housing condition: Temperature: 21-23 C, relative humidity: 61-63%) |
| Wild animals | the study did not involve wild animals |
| Reporting on sex | We only used male mice, directly purchased from Jackson Laboratories. Sex difference was not considered in this study. |
| Field-collected samples | the study did not involve samples collected from the field |
| Ethics oversight | All experimental procedures using live animals were approved by the Salk Institute Animal Care and Use Committee under protocol number 18-00006. |

Note that full information on the approval of the study protocol must also be provided in the manuscript.

# Flow Cytometry

## Plots

Confirm that:

☒ The axis labels state the marker and fluorochrome used (e.g. CD4-FITC).

☒ The axis scales are clearly visible. Include numbers along axes only for bottom left plot of group (a 'group' is an analysis of identical markers).

☒ All plots are contour plots with outliers or pseudocolor plots.

☒ A numerical value for number of cells or percentage (with statistics) is provided.

## Methodology

| | |
|---|---|
| Sample preparation | Isolated nuclei were labeled by incubation with 1:1000 dilution of AlexaFluor488-conjugated anti-NeuN antibody (MAB377X, Millipore) and a 1:1000 dilution of Hoechst 33342 at 4°C for 1 hour with continuous shaking. Fluorescence-Activated Nuclei Sorting (FANS) of single nuclei was performed using a BD Influx sorter with an 85μm nozzle at 22.5 PSI sheath pressure. Single nuclei were sorted into each well of a 384-well plate preloaded with 2 μl of Proteinase K digestion buffer (1μl M-Digestion Buffer, 0.1μl 20 μg/μl Proteinase K and 0.9μl H2O). The alignment of the receiving 384-well plate was performed by sorting sheath flow into wells of an empty plate and making adjustments based on the liquid drop position. Single-cell (1 drop single) mode was selected to ensure the stringency of sorting. For each 384-well plate, columns 1-22 were sorted with NeuN+ (488 +) gate, and column 23-24 with NeuN- (488-) gate, reaching an 11:1 ratio of NeuN+ to NeuN- nuclei.<br><br>Detail experiment protocol for snmC/snm3C-seq is provided in Supplementary Information 1 |
| Instrument | BD Influx |

| Software | BD Influx Sortware v1.2.0.142 |
|---|---|
| Cell population abundance | We sort NeuN+ (488+) gate and NeuN- (488-) gate with an 11:1 ratio into each 384-well plate. |
| Gating strategy | Intact nuclei were first discriminated from debris by virtue of their bright DNA labeling (Hoechst Height signal) followed by light scattering profiles (Forward Scatter (FSC) Height vs Side Scatter (SSC) Height). Events with high Pulse Width measurements for FSC and SSC were then excluded as aggregates. Next, NeuN-AlexaFluor 488 positive or negative nuclei were selected, reaching an 11:1 ratio of NeuN+ to NeuN- nuclei. |

☒ Tick this box to confirm that a figure exemplifying the gating strategy is provided in the Supplementary Information.

