## [Peer Review File · Nature]

Manuscript Title: Single-cell DNA Methylome and 3D Multi-omic Atlas of the Adult Mouse Brain

Editorial Notes:

Reviewer Comments & Author Rebuttals

Reviewer Reports on the Initial Version:

Referees' comments:

Referee #1 (Remarks to the Author):

In this manuscript, Liu et al. presented a comprehensive single-cell sequencing study of the adult mouse brain, covering 117 regions and five modalities of nucleus profiles. They used single-nucleus methylome profiling and joint profiling of DNA methylation and chromatin conformation to analyze over 300K and 170K cells, respectively. They classified the cells into 261 subclasses based on gene expression, chromatin conformation, accessibility, gene body non-CpG cytosine methylation (mCH), and differentially methylated regions of mCG. They identified spatially expressed genes and regulatory elements that are specific to certain cell types and brain regions. They confirmed spatial epigenetic patterns of several neuronal genes and their DMRs, such as *Elavl2* and *Rasgrf2*, using spatially resolved single-cell transcriptomics analysis (MERFISH) generated in a companion manuscript. Moreover, they made a valuable contribution to the research community by publicly sharing their datasets and scripts, which enables other researchers to reproduce their results and to build on their data for further exploration. Overall, this study provides a rich, valuable, and open resource for the neuroscience and epigenetics research community. The authors demonstrate a comprehensive and rigorous approach to combine different modalities of single-cell data and spatial transcriptomics and how this integrative analysis can reveal novel insights into the cellular heterogeneity and spatial organization of the mouse brain. They also illustrate how spatial transcriptomics can be used to validate and refine the results from single-cell sequencing analyses, such as spatially restricted gene expression patterns. There are a number of aspects that can be improved and better clarified.

Hemispheric asymmetry in DNA methylation patterns is a phenomenon observed in the human brain (e.g., PMID: 32151270). This study provides a whole-brain scale dataset of mouse brain methylation patterns. Does the mouse brain also exhibit hemispheric asymmetry and are they correlated with any specific features, such as gene expression or chromatin confirmation?

The authors provide three examples (*Elavl2*, *Rasgrf2*, and *Negr1*) of spatial epigenetic patterns that can be validated in MERFISH. The study will be more impactful if the authors can perform more systematic integration of these datasets.

The manuscript would benefit from more clarity on the validation of the integration strategy. How did the authors assess the accuracy of their method using the joint profiling dataset?

The authors should justify why they chose 100-kb bins for analyses of the mCH and mCG datasets.

The authors used gene expression data from whole cells and other data from nuclei to integrate and analyze their results. However, previous studies have shown that there are differences between single-cell and single-nucleus sequencing data (e.g. PMID: 30586455, 32997994, and 37095394, and doi.org/10.1101/2022.11.07.515504). These differences could affect the data integration and the interpretation of the findings. Therefore, the authors should discuss how they

accounted for these discrepancies and how they might influence their conclusions.

The authors can add some discussion on the limitations of the study, for instance, the link between DNA methylation levels and chromatin structure is not well investigated.

Some of the figures or panels contain enormous details that are hard to read and interpret for readers and reviewers. The authors should provide a more user-friendly way to access their datasets for non-programmers, such as a UCSC Browser interface.

Referee #2 (Remarks to the Author):

This manuscript presents a whole-brain cell atlas from adult mouse tissues, and significantly extends on previous studies by i) integrating multiple molecular modalities, ii) increasing the quantity of cells analysed, and iii) placing cells within the 3D anatomy of the brain.

As well as generating an incredible data resource, this paper presents several interesting biological discoveries and innovative computational approaches. For example, the authors demonstrate that chromatin compartment boundaries are enriched at transcription start and end sites, and provide a novel framework to understanding the role of transcription factors in brain cell subclasses. The approach used to understand epigenetic control of isoform expression was also very innovative and interesting.

The authors are to be congratulated on an exceptional body of work presented to a very high standard. In particular, the authors' efforts to explain a complex data set, in accessible language and figures, is commendable. The revisions suggested below will resolve minor issues related to data interpretation and further improve on the study presentation.

Suggested Revisions:

1) Introduction: Most of the Introduction is a summary of the findings presented in this manuscript, which seems mis-placed. It is also difficult to understand the Introduction since the results are described before the methods have been explained. Could the authors instead use the introduction to describe in greater detail earlier brain methylome and atlas studies? This could help to emphasise the importance and applicability of their work.

2) Lines 379-382: The Authors report that regions with negative correlations between chromatin compartment score and CG methylation, show greater variability in compartment score across cell sub-classes. This is not surprising, since correlations require variability. To detect a correlation between mCG and compartment score, you need variability in both variables. If a compartment score were stable across cell subclasses, it would have no chance of being correlated with mCG. The authors should either i) rephrase their conclusion, or ii) perform additional analyses to account for the expected relationship between compartment score variability and Pearson correlation.

3) Lines 602-605: The data presented on the truncated beta-neurexin isoform is not very compelling. Hypomethylation is observed broadly across cell subclasses, whereas isoform expression is restricted to very few cell subclasses. Furthermore, some cell subclasses from other brain regions (eg sub-cortical exc neurons) have very high isoform expression, but no hypomethylation of the CpG clusters indicated. Also, it is difficult to understand how co-regulated CpGs that are not adjacent to one another could influence splicing. Can the authors provide a more compelling example of mCG regulated isoform expression? For example, in Fig 6b I there are some interesting hypomethylated regions in the CTX, HPF Exc sub-class. The isoform usage in this subclass also seems quite distinct. Do any of the DMRs from Fig 6b I CTX, HPF Exc sub-class correspond to the genomic co-ordinates of the exon used in only 2 transcripts, including the isoforms increased in the same cell sub-class?

4) Lines 620-621: The authors claim that differences between Pearson correlation co-efficients computed using true and shuffled data represent a "gain in predictability through adding intragenic features". This is a very interesting data analysis approach; however, the change in Pearson Correlation Co-efficient will also be influenced by several other factors. For example, a gene with

few intragenic features, or little difference in mCG signal across features will have no chance of producing a high delta PCC value. Did the authors consider correcting the delta PCC for gene length or mCG variance? These limitations should be mentioned briefly in the manuscript text.

5) The authors have generated an incredible data resource, which will be of great interest to neuroscientists. Do the authors intend to develop an online data exploration tool so that this data can be utilised globally?

Minor comments:

6) Suggestions to improve the clarity of text and data presentation are included in comments on the attached pdf file.

7) Statistical significance of Pearson Correlations is reported, but it is unclear whether the p-values have been adjusted for multiple tests.

8) A relevant recent study has not been cited: Herring CA*, Simmons RK*, Freytag S*, Poppe D*, Moffet JJD, Pflueger J, Buckberry S, Vargas-Landin BD, Clément O, Echeverría EG, Sutton GJ, Alvarez-Franco A, Hou R, Pflueger C, McDonald K, Polo JM, Forrest ARR, Nowak AK, Voineagu I, Martelotto L, Lister R# (2022) Human prefrontal cortex gene regulatory dynamics from gestation to adulthood at single-cell resolution. Cell 10.1016/j.cell.2022.09.039

Referee #3 (Remarks to the Author):

In this manuscript Ecker and colleagues presented a large and nearly complete cell atlas of mouse adult brain, based on single-cell methylome sequencing, chromosome conformation capture, and in situ multiplexed RNA spatial mapping. The study was carefully designed in terms of sample collection, brain dissection, CCF registration etc. Data generation was done on highly optimized experimental platforms, and the resulting data quality is high, representing the best that can be achieved by the current research community. The reported data set is unprecedented, in terms of data size and complexity. The authors went through very carefully quality checking and filtering, then created a set of innovative procedures for computational analysis. Even tasks like data storage, cell clustering, multi-omics integration require creative solutions for the data at this size. The authors then went on to perform a number of analyses, including cell type annotation, spatial registration, DMR calling, integration with spatial RNA data and chromatin accessibility data, characterization of chromosomal confirmation differences and changes across cell types and their relationship with DNA methylation (especially neuronal mCH fraction), identification of gene regulatory networks and key transcription factors, examination of splicing isoform diversity and intragenic epigenetic heterogeneity. This led to many novel findings and insights. With such a huge multi-modal data set, there are definitely numerous other directions to explore in terms of computational analysis. I do think the authors have covered some more important and obvious ones, and showcased the value and significance of this multi-omics atlas. Overall, I believe this is a landmark study that was very well designed and executed. There was a high level of rigor, also there were numerous innovative aspects in this study. I think it is appropriate for Nature.

I do not have any major concerns, but do have several questions that I hope the authors can clarify or address.

(1) Gene-body mCH in neuronal cells represents an attractive alternative for capturing gene expression, and it can even detect genes of low transcript abundance that sn-RNA-seq might miss. My question is whether gene-body mCH abundance has its own bias. Is it more sensitive on longer genes (exons + introns)?

(2) TF bindings in specific cell types were inferred by searching for the TF binding motifs in DMRs. This is a very unique and innovative aspect of this study. Most other studies do not have single-cell methylome data, so differential accessible regions(DARs) identified from scATAC-seq data were used for the search of TF binding footprints. Since there is a parallel scATAC-seq data set for the

same mouse brain regions, can the authors do a direct comparison to assess the strength/weakness of using DMRs versus DARs from scATAC-seq data for TF binding detection?

(3) Another strength of this data set is the chromosome conformation map, which allow the authors to link regulatory regions with transcriptional units at a higher level of confidence. In many other studies where chromosome conformation information is not available, a more naive approach of linking regulatory regions and genes based on the linear chromosomal distance was used. Can the authors make a comment on how many more confident linkage can be made by incorporating chromosome conformation information?

(4) For the spatial mapping of single-cell methylomes using MERFISH (LINE 323-325), is there a confidence score for the spatial assignment of each cell or cell type (or sub-class)? If so, what cutoff was used for retaining the confident assignments?

(5) I really like the part on GRN construction and the identification of key TFs. The strategy used here was quite innovative. On the other hands, this part ended with a long list of predictions with no information of the confidence, and no validations. Ultimate validations would be to experimentally perturb the TFs or the regulatory elements in mice, and examine the functional consequences. Obviously this would require very significant efforts beyond the scope of this study. Is there any computational analysis that the authors can do to estimate the confidence of these prediction, maybe though some computational perturbation predictions, such as CellOracle? Also many of the TFs/genes have been knocked out in mice in the past. What are the phenotypes? Can they lend supports to some of the predictions?

Author Rebuttals to Initial Comments:

Reply to Referee #1:

In this manuscript, Liu et al. presented a comprehensive single-cell sequencing study of the adult mouse brain, covering 117 regions and five modalities of nucleus profiles. They used single-nucleus methylome profiling and joint profiling of DNA methylation and chromatin conformation to analyze over 300K and 170K cells, respectively. They classified the cells into 261 subclasses based on gene expression, chromatin conformation, accessibility, gene body non-CpG cytosine methylation (mCH), and differentially methylated regions of mCG. They identified spatially expressed genes and regulatory elements that are specific to certain cell types and brain regions. They confirmed spatial epigenetic patterns of several neuronal genes and their DMRs, such as *Elavl2* and *Rasgrf2*, using spatially resolved single-cell transcriptomics analysis (MERFISH) generated in a companion manuscript. Moreover, they made a valuable contribution to the research community by publicly sharing their datasets and scripts, which enables other researchers to reproduce their results and to build on their data for further exploration. Overall, this study provides a rich, valuable, and open resource for the neuroscience and epigenetics research community. The authors demonstrate a comprehensive and rigorous approach to combine different modalities of single-cell data and spatial transcriptomics and how this integrative analysis can reveal novel insights into the cellular heterogeneity and spatial organization of the mouse brain. They also illustrate how spatial transcriptomics can be used to validate and refine the results from single-cell sequencing analyses, such as spatially restricted gene expression patterns. There are a number of aspects that can be improved and better clarified.

R1.1

We appreciate the reviewer's positive remarks on the significance and impact of our manuscript and the constructive suggestions for improvement.

Per the reviewer's recommendation, we have developed a user-friendly web application (<https://mousebrain.salk.edu>) to facilitate public access to the multi-omic whole-brain dataset. A step-by-step guide to using this browser can be found in section R1.8.

During this revision, we also aligned our cell annotations with the companion whole-brain scRNA-seq manuscript from the Allen Institute for Brain Science (AIBS)¹. This ensures consistent cell taxonomy across the Brain Initiative Cell Census Network (BICCN) consortium. Importantly, these updates do not affect our initial grouping of 4,673 cell groups and primarily involve renaming and minor regrouping at the subclass level. We use the updated taxonomy in our main manuscript and web application. Additionally, we integrated our work with the AIBS's whole-brain MERFISH dataset¹, expanding our spatial annotation of epigenome cells to encompass the entire brain (see section R1.3).

The updated results and downloadable data files are available on our web application and the NeMO archive (<https://mousebrain.salk.edu/download>). We address each comment in detail below.

Hemispheric asymmetry in DNA methylation patterns is a phenomenon observed in the human brain (e.g., PMID: 32151270). This study provides a whole-brain scale dataset of mouse brain methylation patterns. Does the mouse brain also exhibit hemispheric asymmetry, and are they correlated with any specific features, such as gene expression or chromatin conformation?

R1.2

We agree with the reviewer that understanding hemispheric asymmetry in human brain function is crucial, particularly for advancing research in neurological diseases². Investigating inter-hemispheric differences in the mouse brain is indeed an exciting avenue for future work.

However, our study design has limitations that preclude a focused examination of intra-hemispheric differences for two main reasons:

1. The granularity of our anatomical dissection, dissecting the brain into 117 small regions, is designed to enrich our single-cell dataset's metadata annotations. This granularity is vital for accurate cell cluster interpretation, particularly given the limited availability of large-scale spatial transcriptomics when our project began. Consequently, we aggregate data from both hemispheres and multiple mouse brains to ensure adequate tissue samples for nuclei preparation and downstream processing.
2. Our study shares dissected tissue nuclei with a parallel snATAC-seq study³, aiming for consistent and comparable cell population profiling across studies. The combinatorial-indexing approach of the snATAC-seq protocol requires many nuclei, further constraining our ability to separate data by hemisphere.

These methodological decisions, shaped by our primary objective to explore multi-omic cellular diversity across defined anatomical regions, limit our capacity to investigate inter-

hemispheric variations. We have acknowledged these limitations in the revised manuscript's discussion section:

“However, the comprehensive scope of this study presents challenges in addressing additional biological aspects such as intra-hemispheric differences, individual variability, and sex differences. Future research endeavors are anticipated to delve into these areas, contributing to a more comprehensive molecular atlas of the brain.”

The authors provide three examples (*Elavl2*, *Rasgrf2*, and *Negr1*) of spatial epigenetic patterns that can be validated in MERFISH. The study will be more impactful if the authors can perform a more systematic integration of these datasets.

R1.3

We appreciate the reviewer's suggestion and extend our integration analysis with the comprehensive whole-brain MERFISH dataset from the AIBS companion study¹. This dataset comprises 51 coronal slices of a male C57BL/6J mouse brain and includes 3,934,605 cells profiled for 500 genes. These cells are spatially registered to the mouse adult brain common coordinate framework (CCF)⁴ and annotated with detailed CCF anatomical structures based on this registration. Notably, the MERFISH dataset shares consistent cell taxonomy with our methylome dataset, as both are integrated with the AIBS's scRNA-seq dataset.

The integration approach is akin to the snmC-MERFISH integration section in our manuscript. We grouped snmC-seq data by coronal slices and integrated each with corresponding MERFISH data using the 500 profiled genes. Post-integration, we calculated the closest MERFISH neighbor for each methylation cell based on shared subclass labels and Euclidean distance in the integrated low-dimensional space. In alignment with our manuscript's findings, the spatial coordinates of methylation nuclei closely matched with dissected brain regions across multiple slices, confirming that methylation patterns correlate with spatial transcriptome heterogeneity (**Figure R1.3a**).

To assess the accuracy of imputed locations, we compared MERFISH-assigned CCF regions to dissection regions from the methylation cell metadata. For instance, a methylation cell from the primary motor cortex (MOp) that aligns with a MERFISH cell from the same CCF region is considered accurately assigned. On average, 70% of cells were correctly mapped to their dissection regions, demonstrating remarkably high spatial agreement given our dataset's complexities (**Figure R1.3b, c**). Notably, the MERFISH assignment allows us to evaluate the spatial methylation pattern of any gene and regulatory elements in the in-situ coordinates, which are now interactively visualized in our new web application (<https://mousebrain.salk.edu>, see R1.8 for more details). Here, we provide an example for visualizing the RNA level, mCH fraction and mCG fraction of *Cux2* on merfish slice 59, while also provide the dissection region, cell subclass, and CCF region information (See: <https://tinyurl.com/merfish-example>).

Moreover, the spatial imputation allowed us to assign cell subclasses to intricate anatomical structures smaller than our dissection regions (**Figure R1.3d**). For example, we accurately identified the spatial location of several cortical and hippocampal structures (“CA1-ProS Glut”, “DG Glut”, “L2/3 IT CTX Glut”, “L4/5 IT CTX Glut”, “L5 IT CTX Glut”, “L6 CT CTX Glut”), including a particular cell subclass, “L2/3 IT RSP Glut” that only locate at the superficial layer

of the retrosplenial cortex. We also captured many subcortical brain nuclei structures, such as the “STN-PSTN Pitx2 Glut” in the Subthalamic nucleus, “TH Prkcd Grin2c Glut” in the lateral and ventral thalamus, “ZI Pax6 Gaba” in the thalamic zona incerta region. Together, these results demonstrate brain-wide high accuracy in integrating our snmC-seq and MERFISH data.

In our revised manuscript, we added the following text “To extend our spatial annotation to the entire brain, we performed a comprehensive integration with the MERFISH dataset from the AIBS companion study, containing 51 coronal slices and 3.9M cells⁷. This integration helps us to position each nucleus from the methylome dataset into a specific spatial location, facilitating the interpretation of epigenetic profiles in brain-wide anatomical structures (Extended Data Fig. 8, Supplementary Table 9)”

Figure R1.3 | Integration of snmC-seq and whole-mouse-brain MERFISH datasets a, Imputed spatial locations for glutamatergic neurons colored by dissection regions. **b-c,** Example slices showing glutamatergic neurons color by dissection region (left) and CCF anatomical structures (right). **d,** MERFISH Slice 67 color by example cell subclasses.

The manuscript would benefit from more clarity on the validation of the integration strategy. How did the authors assess the accuracy of their method using the joint profiling dataset?

R1.4

We appreciate the reviewer's insights on validating our integration analysis. While a single-modality dataset is unsuitable for verifying cross-modality integration, we have previously addressed this concern. Specifically, we utilized a multi-omic snmCAT-seq dataset (4253 cells) from the human prefrontal cortex that profiles DNA methylome, transcriptome, and chromatin accessibility within the same cell⁵.

In this context, we evaluated five integration tools: 1) Scanorama⁶, 2) Harmony⁷, 3) Seurat⁸, 4) LIGER^{9,10}, and 5) SingleCellFusion⁵. Our benchmark analysis revealed that Seurat and SingleCellFusion performed best using default parameters. We opted for a Seurat-based framework for the current mouse whole-brain study but adapted it to accommodate multi-million atlas-level data.

For the completeness of our response, here we reproduce our integration benchmarking:

Method of Integration Benchmarking

For Scanorama, Harmony and Seurat, we used the top 2000 highly variable genes (HVG) identified by Seurat's FindVariableGenes function. For LIGER and SingleCellFusion, we employed the top 5000 genes with the highest RNA-mCH correlation. We reversed the methylation values ($\max(X) - X$, where X denotes the cell-by-gene methylation fraction matrix) to account for the negative correlation between mCH fraction and RNA expression.

All tools started with a per-cell normalized RNA-HVG and reversed mCH-HVG matrix. Post-integration, we used the decomposed matrix (PC matrix in all integration methods except H matrix in LIGER) for evaluation. Major parameters are listed below; for reproducibility, we have uploaded the steps and input files here:

- 1) For scanorama, we used these parameters ($\sigma=100$, $\alpha=0.1$, $knn=30$) to perform the integration and dimension reduction using Scanorama v1.7 on the scaled (via `scanpy.pp.scale`) mC and RNA matrix. We used the top 20 integrated PCs (`n_components = 20`) for integration evaluation.
- 2) Unlike scanorama, Harmony directly takes a dimension reduction matrix as input. Therefore, we first run PCA separately (`n_components = 20`) on the scaled mCH and RNA matrix and run Harmony (`pyharmony` from <https://github.com/jterrace/pyharmony>) with default parameters on the concatenated PCs. Harmony-integrated PCs were then used for evaluation.
- 3) For Seurat, we followed the Seurat (v4.0.0) vignette steps to perform integration (https://satijalab.org/seurat/articles/integration_introduction.html). When calculating integration anchors (FindTransferAnchors), we use the RNA matrix as the reference matrix, the mCH matrix as the query matrix, and CCA as the dimension reduction method. This allows us to transfer the mCH matrix to the RNA space using the anchors and run PCA (`n_components = 20`) on the concatenated (mCH and RNA) matrix.
- 4) For LIGER, we followed the tutorial from developers (http://htmlpreview.github.io/?https://github.com/welch-lab/liger/blob/master/vignettes/online_iNMF_tutorial.html). We used the online iNMF algorithm¹⁰ with default parameters to perform the integration. We used the normalized

matrix H (the cells' decomposed matrix from the online iNMF algorithm) for integration evaluation.

- 5) Finally, the SingleCellFusion analysis was described in Luo et al.⁵ We used the integrated PCs for evaluation.

Metrics For Integration Evaluation

We used two approaches to evaluate the integration results.

- 1) Co-embedding: We ran UMAP on the decomposed matrix from each tool to provide an overview of the integrated dataset.
- 2) Cell-level: We utilize the ground-truth information from the snmCAT-seq to calculate a self-radius at the single-cell level. Specifically, we first construct a nearest-neighbor index using Annoy (v1.17.0) on the decomposed matrix (euclidean distance). If the same cell's RNA vector is the mCH vector's Kth neighbor, we then use $d=K$ as the self-radius. The quality of the integration can be normalized by $d/2N$, where N is the total number of snmCAT-seq cells involved in the analysis. The value of $d/2N$ ranges from 0 to 1, with smaller values indicating good integration and larger values indicating inadequate integration of mC and RNA profiles of a cell.

Summary

Integration algorithms were first visually evaluated by overlapping different modalities on the UMAP embeddings (**Fig. R1.4.1**). Three tools (SingleCellFusion, Seurat, Harmony) generated well-overlapped UMAP embeddings with cluster-like structures, while the UMAP embedding from the other two tools (LIGER, Scanorama) didn't show clear overlap and structures, indicating inadequate integration.

Furthermore, the self-radius allows us to identify failure events at the cell level. In all five tools (**Fig. R1.4.1, 4th column; Fig. R1.4.2**), we found some failed cells that were either scattered in different clusters or formed some artificial structures (red arrows). We quantify these failed events by setting a threshold (0.3) on the self-radius and counting the cells having larger values. The Seurat (260 cells) and SingleCellFusion (222 cells) have the lowest failed events, while the Harmony (416 cells) have a larger amount of failure compared to them.

Overall, the Seurat and SingleCellFusion provide the best integration results. Harmony integrates most of the cells correctly but also creates artificial populations that do not exist in the original dataset, potentially caused by inaccurately assigned low-quality cells together into a separate cluster in its iterative soft-clustering steps⁷. Noticeably, the integration analysis involves multiple preprocessing steps. For all the tools we tested here, better results might be achieved with further fine-tuning parameters and steps. For example, integrating each cell class (excitatory, inhibitory, and non-neuronal cells) separately and using different input features could impact the results. Nevertheless, here, we only tested some possible parameter combinations. The performance on default settings of each tool also indicates their usability in general, as many real integration analyses do not have the ground truth provided by snmCAT-seq.

Fig. R1.4.1, methylome and transcriptome integration using different tools. Each row is the UMAP embeddings for one integration tool: (a) SingleCellFusion, (b) Seurat, (c) Harmony, (d) LIGER, and (e) Scanorama. From left to right, the UMAP embeddings were colored by 1) the cell's modality, 2) the methylome cells' cell type color, 3) the transcriptome cell's cell type color, and 4) The self-radius for each cell, a larger value (red) indicates the cell is far away from itself between the two integrated modalities.

Fig. R1.4.2. A summary barplot of self-radius in the integration test.

The authors should justify why they chose 100-kb bins for analyses of the mCH and mCG datasets.

R1.5

We thank the reviewer for this suggestion. Similar questions have been addressed in our previous publication¹¹ and are also included here for completeness. We evaluated the clustering performance of mCH and mCG using eight different bin sizes, ranging from 5 kb to 1 Mb, on the MOP-2 dataset, which comprises 2,386 nuclei from two replicas. We observed robust clustering results across these varying parameters.

In terms of data preprocessing, for each bin size, we adhered to the methodology outlined in the original paper, adapting cutoffs relative to the bin size under consideration (for example, the coverage cutoff is scaled linearly based on bin length). These adjustments are designed to maintain a comparable feature set to that of our 100 kb bin analysis. Post preprocessing,

we conducted PCA (**Fig. R1.5a, b**), as well as UMAP embedding on the PCA space (**Fig. R1.5c**), and performed clustering (**Fig. R1.5d**) for each bin size. Notably, the explained variance ratios for top principal components (PCs) and the shapes of PC1 and PC2 embeddings remain consistent across most bin sizes (**Fig. R1.5a, b**). Although varying bin sizes do result in different UMAP embeddings, the overall cell-type structures are preserved (**Fig. R1.5c**).

To quantitatively validate the robustness of our clustering approaches across bin sizes and methylation types, we employed the Adjusted Rand Score (ARS)¹². The ARS calculations affirm that both mCH and mCG offer consistent results across bin sizes (**Fig. R1.5d**).

In summary, our analysis indicates that bin size does not substantially impact the clustering efficacy of the snmC data. We opted for a 100 kb bin size to align with existing literature^{13,14}. While smaller bins could potentially offer more granularity at the enhancer level, our findings suggest that features across different bin sizes contain sufficient redundant information for effective clustering. Moreover, using 100 kb bins has the computational advantage of being order of magnitudes faster than smaller bins, such as 5 kb, allowing for more effective exploration for an atlas dataset analysis.

Fig. R1.5. Comparing the bin-size effect on embedding and clustering results. **a**, Explained variance ratio of top PCs calculated from mCH and mCG matrix. **b**, **c**, Scatter plot of PC1-PC2 and UMAP embeddings ($n=2,386$ for all plots). Each group is clustered separately, but all scatter plots are colored using subclass labels to allow easy visual comprehension. **d**, ARS comparing the clusters identified by each feature set with original subclass labels.

The authors used gene expression data from whole cells and other data from nuclei to integrate and analyze their results. However, previous studies have shown that there are differences between single-cell and single-nucleus sequencing data (e.g., PMID: 30586455,

32997994, and 37095394, and doi.org/10.1101/2022.11.07.515504). These differences could affect the data integration and the interpretation of the findings. Therefore, the authors should discuss how they accounted for these discrepancies and how they might influence their conclusions.

R1.6

We thank the reviewer for pointing out the discrepancies between single-cell RNA-seq (scRNA-seq) and single-nucleus RNA-seq (snRNA-seq) data. In our study, we used the scRNA dataset from the AIBS¹ and included all single nuclei for our snmC-seq3 and snm3C-seq datasets. It's important to note that epigenomic technologies focus solely on genomic DNA modalities and are always carried out on single nuclei. Thus, the main source of discrepancy between cell and nucleus pertains to transcriptome data, which influences cross-modality integration.

In our previous work, we investigated the differences between single-nucleus and single-cell methodologies in the primary motor cortex (MOp)¹⁵ of the mouse brain. Using this MOp dataset, we assessed how varying cell preparation sources of transcriptome datasets integrate with the methylome dataset within our current integration pipeline. This assessment was based on three key questions:

1. Which genes show differential expression (or detection biases) between snRNA-seq and scRNA-seq datasets?
2. Do these technology-biased genes exhibit high cell-type specificity, thus being included in gene lists essential for cross-modality integration?
3. How do integration results differ when combining the methylome dataset with either scRNA-seq or snRNA-seq datasets?

For the first question, our differentially expressed genes (DEG) analysis between scRNA-seq and snRNA-seq identified 531 cell-versus-nuclei DEGs (Adjusted p-value < 0.001; absolute log₂FoldChange > 3) (**Fig. R1.6a**). These DEGs include genes that have very restricted subcellular localization (**Fig. R1.6b**). For example, the nuclear-localized long non-coding RNA *Malat1* was enriched in the snRNA-seq dataset, while mitochondrial and ribosomal components' mRNAs, predominantly cytoplasmic, were more prevalent in the scRNA-seq dataset (**Fig. R1.6c**).

Next, using 9,602 nuclei from the MOp region in our snmC dataset, we carried out an integration analysis of the three datasets following the methodology described in our manuscript. This process began by identifying cluster-enriched genes (CEGs)¹⁶ through cluster comparisons within each dataset. Among the 3,959 CEGs selected from the three datasets, only 51 (1.3%) overlapped with the cell-versus-nuclei DEGs (**Fig. R1.6a**), in concordance with the little difference observed in the integrated UMAP embeddings (**Fig. R1.6d**). We further evaluated the overlap score¹⁷ of methylome and transcriptome subclasses when using different RNA datasets. This metric range from 0 to 1, with higher value indicates the individual cluster from mC and RNA dataset is more overlapped in the co-clustering post integration. The confusion matrix of the subclass overlap score indicates a highly consistent cluster match in both scRNA-mC and snRNA-mC integration. (**Fig. R1.6e**).

In summary, our findings suggest that the technical biases anticipated between single-cell and single-nuclei methodologies influence only a minor portion of the transcriptome used in the integration analysis. This small proportion appears to exert minimal impact on the final integration outcomes.

Fig R1.6. Evaluating the effect of single-nucleus and single-cell discrepancies in integration. **a**, Venn plot shows the overlap between cell-versus-nuclei DEGs and CEGs selected for integration. **b**, Gene Ontology (GO) term analysis for cellular components with cell-versus-nuclei DEGs. **c**, Scatterplot shows normalized and log-transformed UMI counts of each gene in scRNA-seq (x-axis) and snRNA-seq datasets (y-axis). Each dot is a gene. **d**, Integration UMAP of all MOp cells from the snmC-seq, scRNA-seq, and snRNA-seq datasets, colored by modality, gray dots on the background indicate other datasets. **e**, Confusion matrix of overlap score^{11,17} between annotated cell subclasses of snmC-snRNA (left) and snmC-scRNA integration (right). The overlap score ranges from 0-1, and the larger, the more overlapped.

The authors can add some discussion on the limitations of the study; for instance, the link between DNA methylation levels and chromatin structure is not well investigated.

R1.7

Our study offers a comprehensive resource covering the cellular-spatial and regulatory genome landscape of the entire mouse brain. In Figure 4, we delved deeply into the relationship between gene body methylation and surrounding chromatin conformation diversities. Specifically, our findings showed that chromatin compartment changes in the 3D genome overlap with genes predominant in the early development phase (**Fig. 4a-d**). Long neuronal genes' boundaries overlap with domain boundaries, and this boundary strength is correlated with gene body methylation (**Fig. 4e-h**). Individual highly variable interactions demonstrate significant correlations with adjacent gene body methylation, creating potential links between cis-regulatory elements and target genes (**Fig. 4i-m**).

However, we acknowledge the limited discussion on the relationship between chromatin conformation features and their underlying genomic region's methylation levels. This theme is elaborated more extensively in another of our publications¹⁸. In Summary, that research discerned a negative correlation between chromatin interaction strength and average methylation levels of interacting anchors. This was predominantly observed in the active compartment (A Compartment), suggesting a closer link between DNA methylation and genome conformation in euchromatin regions. Future research is required for understanding the mechanistic relationships here, such as the involvement of pivotal neuronal lineage determining transcription factors. One potential example is *Neurog2*, which has been found to modulate enhancer activity DNA demethylation, boost chromatin accessibility, and facilitate chromatin looping in the mouse cerebral cortex as highlighted by Noack et al.¹⁹

Lastly, the resolution of current single-cell chromatin conformation analysis remains a limitation, especially compared to bulk MicroC technologies²⁰. Our snm3C-seq analysis, conducted at a resolution of 10 Kb, might not capture the intricacies of individual regulatory elements' DNA methylation and their chromatin conformation. However, upcoming technological (e.g., single-cell MicroC-seq²¹) and computational advancements promise enhanced resolutions, paving the way for more nuanced analyses.

We have modified our discussion section as follows:

This approach yields unprecedented gene-specific chromatin conformation landscapes that offer predictions about the importance of individual chromatin interaction pixels at a 10-kb resolution. These results offer numerous intriguing candidate loci that can potentially elucidate the causal relationships between DNA methylation statuses and chromatin structures. It paves the way for employing advanced technologies like epigenetic editing²² and region-capture MicroC²⁰ in future investigations.

Some of the figures or panels contain enormous details that are hard to read and interpret for readers and reviewers. The authors should provide a more user-friendly way to access their datasets for non-programmers, such as a UCSC Browser interface.

R1.8

We appreciate the reviewer's feedback on the challenges of displaying extensive data in static figures. In response, we have developed a web application (<https://mousebrain.salk.edu>) to comprehensively visualize our whole-brain multi-omic dataset.

The application has two design principles:

1. **Versatility:** It allows users to visualize all molecular modalities (mCH, mCG, ATAC, chromatin conformation, RNA), metadata, and cell coordinates (tSNE, UMAP, or imputed MERFISH coordinates for methylome cells) in various scatter plots, as well as subclass genome tracks using the HiGlass genome browser.
2. **Modularity:** Users can customize layouts based on their needs and save them for future use. Notably, we've integrated ChatGPT, enabling users to craft complex layouts using natural language (examples provided below). This feature aims to enhance user experience, particularly for those without programming expertise.

Addressing the browser suggestion, we opted for the HiGlass browser over the UCSC genome browser. This decision was influenced by HiGlass's flexible programming environment and its better capabilities in visualizing 3D genome data. However, for those familiar with the UCSC browser, we've made all track files available for download (<https://mousebrain.salk.edu/download>) that are compatible with it.

For a deeper dive into our web application's capabilities, we provide a comprehensive walkthrough below.

General Application Structure

The primary interface of the application is divided into two key sections:

1. **User Input Area:** Located at the top, this is where users specify the type of graph or visualization they wish to see. And other functionalities to provide examples and download/upload current layouts.
2. **Graph Display Area:** Positioned below the input section, this area displays the visualizations. After users input their preferences, they must click "Add Panels" to generate the desired scatter plot or HiGlass browser view within this section.

Fig R1.8a General Application Structure

Creating Custom Graph Panels with Text Input

Users have the flexibility to craft various visualizations that reflect diverse facets of the brain atlas. A great starting point is the array of example buttons situated below the input area. These pre-set examples showcase the diverse capabilities and functionalities of the browser, as detailed below.

Fig R1.8b Demonstration of steps to draw a graph.

Available Graph Panel Types

We currently offer six distinct graph panel types to cater to varied visualization needs:

1. **Categorical Scatter Plot:** Creates a scatter plot using tSNE, UMAP, or predicted MERFISH coordinates of the methylome dataset, coloring dots by categorical cell metadata (e.g., cell subclass, dissection region).

Example-URL-1

Fig R1.8c t-SNE and MERFISH plot (slice 59) of snmC nuclei color in cell subclass

2. **Continuous Scatter Plot:** Creates a scatter plot with the same coordinate options, coloring dots by continuous variables like normalized gene body mCH fraction, mCG fraction, or RNA expression $\log(\text{CPM}+1)$ based on integrated RNA cell cluster pseudo bulk profiles.

Example-URL-2

Fig R1.8d From left to right, the snmC dataset t-SNE plot is colored by normalized mCH fraction, mCG fraction and RNA level of *Gad1*.

- Multi-2D HiGlass View:** Features a HiGlass browser view for multiple cell subclasses, displaying chromatin conformation, mCH/mCG fraction, ATAC, and SMART-seq tracks. This browser also incorporates helpful functions like gene annotation tracks.

Example URL-3:

Fig R1.8e From top to bottom, the left plot shows the gene location, genome browser view of mCH fraction (blue), mCG fraction (green), ATAC CPM (orange) and RNA CPM (purple) and normalized chromatin contact matrices around the *Lingo2* gene from “L2/3 CTX Glut”. The right plot shows the same information for “STR D2 Gaba”.

- Multi-1D HiGlass View:** Offers a HiGlass browser view focusing on several cell subclasses, displaying only 1-D tracks. Users can group and color tracks by cell subclass or modality.

Example Input-URL-4:

Fig R1.8f From top to bottom, genome browser view depicting modality-specific color-coded

tracks for mCH fraction (blue), mCG fraction (green), ATAC CPM (orange) and RNA CPM (purple) around *Sst* gene in “CA3 Glut” and “Sst Gaba” cell subclasses. Numbers on the right side display the data range.

5. **Two Cell Type Difference Comparison HiGlass View:** Designed for highlighting differences between two cell subclasses.

Example URL-5:

Fig R1.8g From top to bottom, the left plot shows the gene location, genome browser view of mCH fraction (blue), mCG fraction (green), ATAC CPM (orange) and RNA CPM (purple) and normalized chromatin contact matrices around the *Lingo2* gene from “L2/3 IT CTX Glut”. The right plot shows the same information for “STR D2 Gaba”. The middle plot indicates the log(left/right) values to highlight the differences between the two cell subclasses. Red color indicates the left cell subclass (“L2/3 IT CTX Glut”) has higher value, blue color indicates the right cell subclass (“STR D2 Gaba”) has higher value.

6. One Cell Type Loop Zoom-In HiGlass View: Focuses on a detailed view of one cell type loop.

Example URL-6:

Fig R1.8h From top to bottom, the left plot shows the gene location, genome browser view of mCH fraction (blue), mCG fraction (green), ATAC CPM (orange) and RNA CPM (purple) and normalized chromatin contact matrices between chr1:11550000-11720000 for CA3 Glut. The right plot shows this information for a zoomed-in region between chr1:12710000-12910000 (indicated by an interactive gray box in the center of the left plot).

Customizing Generated Graph Panels

Additionally, each graph panel type is customizable. Users can access each graph's control panel by clicking on the top tab named "Control". The customization includes modifying aspects like cell subclasses, color schemes, cell coordinates (including MERFISH slices), genome browser modalities or target genome regions.

Creating Custom Graph Panels with Natural Language

To enhance user accessibility, we've incorporated ChatGPT into our browser. This integration empowers users to input commands in intuitive natural language rather than relying on structured instructions.

Users can activate this feature by toggling the "Use ChatGPT" switch (**Fig 1.8i**). Upon activation, a new set of example buttons emerges (**Fig 1.8i**), presenting layout instructions articulated in natural language. For instance, the "Gene mCH" example transforms to: "Make a scatter plot for the *Gad1* gene body mCH". The "Multi-2D Higlass" example evolves into: "Show me a multi 2D HiGlass browser for three cell types: CA3 Glut, Sst Gaba, and STR D2 Gaba, centered on the *Nfia* gene locus".

It's crucial to clarify that while we employ the ChatGPT API to translate user input into structured instructions, all visualizations, graphs, and browsers display genuine experimental data produced in this study, with no artificial generation from ChatGPT is involved.

Fig 1.8i Demonstration of steps to draw a graph by using ChatGPT.

Enhanced User Experience: Plot Synchronization and Cell Clipboard Function

To enrich the usability of our platform when working with multiple graph panels, we've integrated a couple of essential features:

Plot Synchronization: When users open multiple scatter plots based on the same coordinates, movements and zoom levels are synchronized across these plots. This offers a seamless experience, especially when simultaneously visualizing cell metadata and multiple gene plots on the same page.

Example Input-7 (scroll your mouse on the plot to zoom in/out):

R1.8j Synchronized Zoom in view shows Cux2's RNA (middle plot) and mCH (right plot) level in a subset of cell subclasses (left plot).

Cell Clipboard Function: Recognizing the frequent need to delve deeper into specific cells or cell types in the genome browser, we've introduced the cell clipboard function. Whenever a user selects cells within a scatter plot, the relevant cell metadata is captured and saved to a 'cell card' in the clipboard. The clipboard can be toggled by clicking the "Open Cell Clipboard" button in the user input area. This facilitates easy copying of the cell subclass name and the subsequent loading of associated genomic tracks in a HiGlass panel.

Fig. R1.8k Demonstration of cell clipboard function

Preserving Your Workspace: Save and Load Layout

Upon finalizing their desired layout, users can save their current configuration. Simply click on the "Download Current Layout Config" button, located at the top right. To resume work or view this layout later, users can upload the saved config file via the browser's config upload box on a new page.

Fig. R1.8I Demonstration of config layout function

Reply to Referee #2:

This manuscript presents a whole-brain cell atlas from adult mouse tissues, and significantly extends on previous studies by i) integrating multiple molecular modalities, ii) increasing the quantity of cells analyzed, and iii) placing cells within the 3D anatomy of the brain. As well as generating an incredible data resource, this paper presents several interesting biological discoveries and innovative computational approaches. For example, the authors demonstrate that chromatin compartment boundaries are enriched at transcription start and end sites, and provide a novel framework to understanding the role of transcription factors in brain cell subclasses. The approach used to understand epigenetic control of isoform expression was also very innovative and interesting. The authors are to be congratulated on an exceptional body of work presented to a very high standard. In particular, the authors' efforts to explain a complex data set, in accessible language and figures, is commendable. The revisions suggested below will resolve minor issues related to data interpretation and further improve on the study presentation.

R2.1

We thank the reviewer for appreciating the significance and impact of our manuscript and providing helpful comments to improve our study. Per the reviewer's recommendation, we have developed a user-friendly web application (<https://mousebrain.salk.edu>) to facilitate public access to the multi-omic whole-brain dataset. A step-by-step guide to using this browser can be found in section R2.6.

During this revision, we also aligned our cell annotations with the companion whole-brain scRNA-seq manuscript from the Allen Institute for Brain Science (AIBS)¹. This ensures consistent cell taxonomy across the Brain Initiative Cell Census Network (BICCN) consortium. Importantly, these updates do not affect our initial grouping of 4,673 cell groups and primarily involve renaming and minor regrouping at the subclass level. We use the updated taxonomy in our main manuscript and web application. Additionally, we integrated our work with the AIBS's whole-brain MERFISH dataset¹, expanding our spatial annotation of epigenome cells to encompass the entire brain.

The updated results and downloadable data files are available on our web application and the NeMO archive (<https://mousebrain.salk.edu/download>). We address each comment in detail below.

Suggested Revisions:

1) Introduction: Most of the Introduction is a summary of the findings presented in this manuscript, which seems mis-placed. It is also difficult to understand the Introduction since the results are described before the methods have been explained. Could the authors instead use the introduction to describe in greater detail earlier brain methylome and atlas studies? This could help to emphasize the importance and applicability of their work.

R2.2

We appreciate the reviewer's feedback on the introduction section. We have revised this section to focus on providing a detailed context and background, particularly emphasizing earlier brain methylome and atlas studies, as suggested. We have also minimized the

premature presentation of our findings to ensure a coherent flow of information. Here is the revised version of our Introduction:

“

The mouse brain is a complex organ comprising millions of cells forming hundreds of anatomical structures and diverse cell types^{1,4,15,23–25}. Advances in single-cell transcriptome and epigenome technologies have illuminated the intricate molecular diversity of the mammalian brain, offering insights into epigenetic mechanisms that play a central role in orchestrating this biological diversity^{3,11,26–28}.

Cytosine DNA methylation (5mC), a covalent modification found in post-mitotic cells throughout their lifespan²⁹, is associated with neuronal function, behavior, and various diseases³⁰. While 5mC predominantly occurs at CpG sites (mCG) in mammalian genomes, non-CpG cytosine methylation (mCH, H denotes A, C, or T) is also prevalent in neurons^{29,31}. CpG and CpH methylation modulate transcription factor binding and gene transcription through dynamic occurrence at regulatory elements and gene bodies³². Both methylation types directly influence the DNA-binding of methyl CpG binding protein 2 (MeCP2)^{33–36}, a critical 5mC reader and the cause of Rett syndrome³⁷. Genome-wide differential methylation analysis can predict millions of regulatory elements, yielding a cellular taxonomy and a base-resolution genome atlas^{11,38}.

Furthermore, cis-regulatory elements in complex mammalian genomes can operate over long distances to regulate their target genes³⁹. Understanding the relationships between physical contact frequency of enhancers and promoters, and their collective impact on gene body epigenetics and transcriptomic status is crucial to decoding the molecular diversity of the mammalian brain. Our prior work employed single-nucleus methylome and chromatin conformation capture sequencing (snm3C-seq) for concurrent examination of these aspects⁴⁰. However, a detailed, brain-wide chromatin conformation map remains to be charted.

In this study, we employ enhanced single-nucleus methylation sequencing (snmC-seq3) and snm3C-seq technologies to analyze DNA methylomes and the 3D genome with unprecedented detail^{13,41}. We collect 301,626 methylomes and 176,003 m3C joint profiles from the entire mouse brain, yielding a dataset comprising 786 billion final methylation reads (snmC-seq3 + snm3C-seq) and 33 billion cis-long-range chromatin contacts (snm3C-seq). This rich dataset facilitated the identification of 4,673 cell groups, aligning coherently with data from the BRAIN Initiative Cell Census Network (BICCN)^{1,3}. The methylome clusters are annotated using the nomenclature from companion transcriptomic studies¹, offering a comprehensive multi-omic resource for further research.

Our analysis underscores the prevalence of spatial information in the epigenome, which was validated using a MERFISH dataset created with genes showing distinct gene-body methylation patterns across brain regions. We also deepen our inquiry into the regulatory landscapes of individual genes by examining thousands of aggregated epigenetic profiles. Notable connections emerge between chromatin conformation diversity and gene body methylation profiles across multiple genome scales. By intersecting this epigenetic dataset with a correlation-based analysis, we construct gene regulatory networks that connect transcription factors, differentially methylated regions (DMRs), and potential target genes. Finally, integration with a whole-brain full-length SMART-seq dataset¹ further illuminates the

interplay between epigenetic profiles and transcriptional dynamics within individual long neuronal genes.

To facilitate broader access to this invaluable resource, we introduce the mouse brain cellular and genomic browser (<https://mousebrain.salk.edu>), a user-friendly platform for data query and visualization. By unveiling the multifaceted complexities of the mouse brain's molecular architecture, our study paves the way for deeper insights into the epigenetic and transcriptomic intricacies underpinning brain function and diseases.

”

2) Lines 379-382: The Authors report that regions with negative correlations between chromatin compartment score and CG methylation, show greater variability in compartment score across cell sub-classes. This is not surprising, since correlations require variability. To detect a correlation between mCG and compartment score, you need variability in both variables. If a compartment score were stable across cell subclasses, it would have no chance of being correlated with mCG. The authors should either i) rephrase their conclusion, or ii) perform additional analyses to account for the expected relationship between compartment score variability and Pearson correlation.

R2.3

We apologize for the confusion of this statement. We were aiming to compare the negatively correlated bins versus the positively correlated bins in Fig. 4c. Without plotting the variation in the y-axis, both directions of correlation have bins that pass the significance, but the positively correlated bin has a much lower standard deviation compared to negatively correlated bins, even when they have similar absolute correlation values.

We clarified this statement in the main text:

Additionally, we observed that the compartment score of negatively correlated bins demonstrated greater variability across cell subclasses **than the positively correlated bins** (Fig. 4c, Extended Data Fig. 9b, c), suggesting that these negatively correlated bins exhibit dramatic activity change across a wide range of cell **subclasses**.

3) Lines 602-605: The data presented on the truncated beta-neurexin isoform is not very compelling. Hypomethylation is observed broadly across cell subclasses, whereas isoform expression is restricted to very few cell subclasses. Furthermore, some cell subclasses from other brain regions (eg sub-cortical exc neurons) have very high isoform expression, but no hypomethylation of the CpG clusters indicated. Also, it is difficult to understand how co-regulated CpGs that are not adjacent to one another could influence splicing. Can the authors provide a more compelling example of mCG regulated isoform expression? For example, in Fig 6b I there are some interesting hypomethylated regions in the CTX, HPF Exc sub-class. The isoform usage in this subclass also seems quite distinct. Do any of the DMRs from Fig 6b I CTX, HPF Exc subclass correspond to the genomic coordinates of the exon used in only 2 transcripts, including the isoforms increased in the same cell sub-class?

R2.4

We appreciate the reviewer's insightful comments and suggestions. We agree that Figure 6b was complex, potentially obscuring the key message of this section. To enhance clarity, we have relocated the previous Fig. 6b to Extended Data Figure 13a and simplified the main figure

to focus on the diversity of alpha- and beta-Neurexin and its association with single-base mCG fractions.

Methodological Notes:

We have updated our integration analysis of the snmC and SMART-seq dataset to align with the updated cell taxonomy used elsewhere in the study. Post-integration, each methylation cell subclass is paired with a transcript-specific profile, quantified from SMART-seq using Kalisto (see Methods). We employed “Transcript Per Million” (TPM) for absolute transcript expression levels and “Percent Spliced In” (PSI) for relative exon expression levels.

Detail response:

The alpha- and beta-isoforms of neurexin are pivotal in synaptogenesis and synapse recognition⁴². In our revised Figure 6b, c (and also **Fig. R2.4**), we illustrate the total *Nrxn3* gene expression (sum across all transcripts), representative alpha-*Nrxn3* (ENSMUST00000163134), and beta-*Nrxn3* (ENSMUST00000110130) TPM in the methylome t-SNE coordinates. As expected, *Nrxn3* is broadly expressed across neurons with quantitative isoform differences among cell subclasses.

In response to the reviewer’s suggestion, we performed a thorough correlation analysis between *Nrxn3* isoform TPMs and surrounding single CpG mC fractions. We aimed to unbiasedly display the relationships between isoform diversity and mCG methylation, searching for compelling examples. Interestingly, a generally negative correlation emerged between transcripts and CpGs within transcript bodies, with a stronger correlation close to the start (5’ end) of the transcript (**Fig. R2.4d** for alpha-*Nrxn3*, **Fig. R2.4e** for beta-*Nrxn3*). This result indicates a stronger co-regulatory interplay between the transcript expression and their adjacent transcript body CpG sites. Hypomethylation of whole-transcript-body CpG sites potentially provides a more active status to elevate the expression of the corresponding transcript.

We also observed several highly correlated small regions across the *Nrxn3* gene body. The two most correlated regions are located at chr12:88,724,000-88,728,000 and chr12:89,813,000-89,816,000 (indicated by red arrows 1 and 2 in **Fig R2.4d-g**). Both regions are downstream adjacent to the promoter and first-exon region of alpha- and beta-*Nrxn3* transcript, respectively. To plot these two regions in more detail, the genome browser view of five representative cell types is displayed in main Figure 6f and the interactive HiGlass browser is also provided (Region1 Link; Region2 Link).

These two regions displayed interesting patterns with the alpha- and beta-*Nrxn3* transcripts (**Fig. R2.4d-g**). Region 1 (chr12:88,724,000-88,728,000) is negatively correlated with alpha-*Nrxn3* TPM (**d**), beta-*Nrxn3* TPM (**e**), and alpha-*Nrxn3* first exon’s PSI (**f**), but is strongly positively correlated with beta-*Nrxn3* first exon’s PSI (**g**). This indicates that hypomethylation of Region 1 potentially activates the *Nrxn3* total gene expression, with a stronger effect on alpha transcript over the beta transcript. Region 2 (chr12:89,813,000-89,816,000) is negatively correlated with beta-*Nrxn3* TPM (**e**) and its first exon’s PSI (**g**), and positively correlated with alpha-*Nrxn3* TPM (**d**) and its first exon’s PSI (**f**). This indicates that hypomethylation of Region 2 potentially activates beta-*Nrxn3* and also negatively impacts the alpha-*Nrxn3* expression level.

A potential explanation for the interesting difference between Region 1 and Region 2 is that the alpha and beta-*Nrxn3* transcripts might compete for the limited transcription machinery proteins to initiate transcription at their corresponding promoter regions. The hypomethylation of Region 1 can elevate the total *Nrxn3* gene expression and, therefore, also elevate beta-*Nrxn3* TPM. However, the hypomethylation of Region 2 might be critical for the beta-*Nrxn3* promoter to gain more transcription initiation than the alpha promoter, thereby having a reverse effect on the two promoters. With only the correlation analysis, we cannot confirm whether the distal relationship between Region 2 and Region 1 is due to direct regulation or some indirect effects. These distal co-regulatory relationships would require further genome or epigenome editing experiments, which are beyond the scope of this study but open up opportunities for future research.

After reorganizing the figures, we modified the main text as follows:

“We observed that *Nrxn3* is broadly expressed across neurons, with its alpha- and beta-*Nrxn3* isoform showing diverse patterns among cell subclasses (Fig. 6b). Surprisingly, these isoform expression patterns also matched with the methylation fraction of single CpGs located around the *Nrxn3* gene body (Extended Data Fig. 13a), with two particularly highly correlated regions positioned downstream of the alpha- and beta-*Nrxn3*'s first exon, respectively (Fig. 6c, Region 1 and 2).”

Fig. R2.4 Relationships between *Nrnx3* isoform diversity and mCG methylation | a-c, Cell-group-centroids t-SNE plot colored by total *Nrnx3* gene expression TPM in SMART-seq (sum up all transcripts) (a), alpha-*Nrnx3* TPM (b) and beta-*Nrnx3* TPM (c). d-g, Each dot represents a single-CpG site located between chromosome 12 88.5 to 90.5 megabase pairs (Mbp). For (d), the x-axis represents the genomic position and the y-axis depicts the correlation between alpha-*Nrnx3* TPM and the methylation level of single-CpG sites among all neuronal subclasses. Positive correlations are colored in red, while negative correlations are represented in blue to assist visualization. Red arrows point to Region1 and Region2 respectively. Other scatter plots show the same layout for CpG mC fractions' correlation with beta-*Nrnx3* TPM (e), alpha-*Nrnx3* first exon's PSI (f), and beta-*Nrnx3* first exon's PSI (g).

4) Lines 620-621: The authors claim that differences between Pearson correlation coefficients computed using true and shuffled data represent a "gain in predictability through adding intragenic features". This is a very interesting data analysis approach; however, the change in Pearson Correlation Coefficient will also be influenced by several other factors. For example, a gene with few intragenic features or little difference in mCG signal across features will have

no chance of producing a high delta PCC value. Did the authors consider correcting the delta PCC for gene length or mCG variance? These limitations should be mentioned briefly in the manuscript text.

R2.5

The reviewer raised an excellent point that we didn't cover in our current manuscript. To address this point, we calculated various metrics in conjunction with delta PCC from transcript prediction models to assess the influence of these factors on the outcomes.

Notably, before proceeding with training and prediction, we applied filters to the genes based on criteria including their methylome, chromatin conformation, and isoform diversity (detailed in the Methods section). This filtration was necessary because analyzing genes with little diversity would not yield meaningful results. Consequently, our focus shifted to genes exhibiting heterogeneity in both epigenome and transcriptome. We excluded genes with notable isoform diversity but negligible epigenetic variations, such as short genes with minimal overlap with CpG sites or m3C bins.

We computed the following metrics for 4,064 genes included in our analysis:

1. Number of Methylome Features: We considered the gene's exon region and 300 bp exon flanking regions (5' and 3') as the methylome features (Methods), resulting in a number of features roughly three times the exon number of the gene.
2. Median Standard Deviation of Methylome Features' mCG Fractions: Within each gene and cell group, we calculated the standard deviation (STD) among the features' mCG fractions. The median STD across cell groups was taken as the gene's value, emphasizing variance among intragenic features within cell types.
3. Number of Chromatin Conformation Features: These are derived from the gene body overlapping 10Kb-by-10Kb interaction bins.
4. Median STD of Chromatin Conformation Features: Similar to the second parameter, we measured STD among features for each gene and cell group, then recorded the median STD across cell groups as the gene's value.

Metrics 1-4 primarily evaluate the input of the prediction models using methylome and chromatin conformation data. We introduced two additional metrics to assess the models' output space, utilizing the SMART-seq dataset:

5. Number of Transcripts: We counted the number of expressed transcripts (TPM > 1e-2) for each gene.
6. Median STD of Transcript TPMs: The STD among transcripts' $\log(\text{TPM} + 1)$ was calculated for each gene and cell group, with the median STD across groups determined as the gene's value.

We subsequently analyzed the relationship of these metrics with the mean delta PCC of each gene (average delta PCC across a gene's transcripts). For metrics 1-4, we identified minimal correlations with delta PCC (**Fig. R2.5a-d,g**), indicating that predictability is not merely dependent on the number or intragenic variance of these features. Besides, there is also little correlation between the number of transcripts and the mean delta PCC (**Fig. R2.5e**). A weak correlation was observed between the transcript TPMs' STD and the delta PCC (**Fig. R2.5f,g**,

PCC=0.15, FDR=1e-5), suggesting the transcript TPM diversity is also not a driver of the gain of predictability.

Overall, we observed a diverse range of delta PCC across different genes (**Figure R2.5a-f**, y-axis). However, this variance in the gain of predictability cannot be solely attributed to the number of features or the variability within the feature or target value spaces. Given the unique characteristics inherent to each gene, establishing a universal rule applicable to all genes proves challenging. Additionally, we do not anticipate that all isoform diversities can be comprehensively explained by epigenetic features alone. Nevertheless, our analysis identifies potential candidates for in-depth exploration into the roles of epigenetic regulation in isoform diversity.

Fig. R2.5. Evaluating the effects of various metrics on the transcript prediction models outcomes | a-f, For (a), scatterplot shows the PCC between the number of exon features (x-axis) and PCC between predicted TPM and true TPM (y-axis) for each highly-variable gene (dot). Other scatterplots show the same information for exon features' mCG STD (b), number of 3C features (c), 3C features' contact strength STD (d), Number of transcripts (e), and transcripts TPM's STD (f). **g**, Heatmap shows the PCC between different metrics and the delta PCC

5) The authors have generated an incredible data resource, which will be of great interest to neuroscientists. Do the authors intend to develop an online data exploration tool so that this data can be utilized globally?

R2.6

We greatly appreciate the reviewer for recognizing the value of our data resource. A data exploration tool is essential for such a complex dataset and has been required by multiple reviewers. In response, we have developed a web application (<https://mousebrain.salk.edu/>) for comprehensive visualization of our whole-brain multi-omic dataset.

The application has two design principles:

1. **Versatility:** It allows users to visualize all molecular modalities (mCH, mCG, ATAC, chromatin conformation, RNA), metadata, and cell coordinates (tSNE, UMAP, or imputed MERFISH coordinates for methylome cells) in various scatter plots, as well as subclass genome tracks using the HiGlass genome browser⁴³.
2. **Modularity:** Users can customize multi-panel layouts based on their needs and save them for future use. Notably, we've integrated ChatGPT, enabling users to craft complex layouts using natural language (examples provided below). This feature aims to enhance user experience, particularly for those without programming expertise.

For a deeper dive into our web application's capabilities, we provide a comprehensive walkthrough below.

General Application Structure

The primary interface of the application is divided into two key sections:

1. **User Input Area:** Located at the top, this is where users specify the type of graph or visualization they wish to see. And other functionalities to provide examples and download/upload current layouts.
2. **Graph Display Area:** Positioned below the input section, this area displays the visualizations. After users input their preferences, they must click "Add Panels" to generate the desired scatter plot or HiGlass browser view within this section.

Fig R2.6a General Application Structure

Creating Custom Graph Panels with Text Input

Users have the flexibility to craft various visualizations that reflect diverse facets of the brain atlas. A great starting point is the array of example buttons situated below the input area. These pre-set examples showcase the diverse capabilities and functionalities of the browser, as detailed below.

Fig R2.6b Demonstration of steps to draw a graph.

Available Graph Panel Types

We currently offer six distinct graph panel types to cater to varied visualization needs:

1. **Categorical Scatter Plot:** Creates a scatter plot using tSNE, UMAP, or predicted MERFISH coordinates of the methylome dataset, coloring dots by categorical cell metadata (e.g., cell subclass, dissection region).

Example-URL-1

Fig R2.6c t-SNE and MERFISH plot (slice 59) of snmC nuclei color in cell subclass

2. **Continuous Scatter Plot:** Creates a scatter plot with the same coordinate options, coloring dots by continuous variables like normalized gene body mCH fraction, mCG fraction, or RNA expression $\log(\text{CPM}+1)$ based on integrated RNA cell cluster pseudo bulk profiles.

Example-URL-2

Fig R2.6d From left to right, the t-SNE plot is colored by normalized mCH fraction, mCG fraction, and RNA level of *Gad1*.

- Multi-2D HiGlass View:** Features a HiGlass browser view for multiple cell subclasses, displaying chromatin conformation, mCH/mCG fraction, and ATAC tracks. This browser also incorporates helpful functions like gene annotation tracks.

Example URL-3:

Fig R2.6e From top to bottom, the left plot shows the gene location, genome browser view of mCH fraction (blue), mCG fraction (green), ATAC CPM (orange) and RNA CPM (purple) and normalized chromatin contact matrices around the *Lingo2* gene from “L2/3 CTX Glut”. The right plot shows the same information for “STR D2 Gaba”.

- Multi-1D HiGlass View:** Offers a HiGlass browser view focusing on several cell subclasses, displaying only 1-D tracks. Users can group and color tracks by cell subclass or modality.

Example Input-URL-4:

Fig R2.6f From top to bottom, genome browser view depicting modality-specific color-coded tracks for mCH fraction (blue), mCG fraction (green), ATAC CPM (orange), and RNA CPM (purple) around *Sst* gene in “CA3 Glut” and “Sst Gaba”. The numbers on the right side display the data range.

- Two Cell Type Difference Comparison HiGlass View:** Designed for highlighting differences between two cell subclasses.

Example URL-5:

Fig R2.6g From top to bottom, the left plot shows the gene location, genome browser view of mCH fraction (blue), mCG fraction (green), ATAC CPM (orange), RNA CPM (purple) and

normalized chromatin contact matrices around the *Lingo2* gene from “L2/3 CTX Glut”. The right plot shows the same information for “STR D2 Gaba”. The middle plot indicates the log(left/right) values to highlight the differences between the two cell subclasses. Red color indicates the left cell subclass (“L2/3 IT CTX Glut”) has higher value, blue color indicates the right cell subclass (“STR D2 Gaba”) has higher value.

7. **One Cell Type Loop Zoom-In HiGlass View:** Focuses on a detailed view of one cell type loop.

Example URL-6:

Fig R2.6h From top to bottom, the left plot shows the gene location, genome browser view of mCH fraction (blue), mCG fraction (green), ATAC CPM (orange) and RNA CPM (purple) and normalized chromatin contact matrices between chr1:11550000-11720000 for CA3 Glut. The right plot shows this information for a zoomed-in region between chr1:12710000-12910000 (indicated by an interactive gray box in the center of the left plot).

Customizing Generated Graph Panels

Additionally, each graph panel type is customizable. Users can access each graph's control panel to modify aspects like cell subclasses, color schemes, cell coordinates (including MERFISH slices), genome browser modalities, or target genome regions.

Creating Custom Graph Panels with Natural Language

To enhance user accessibility, we've incorporated ChatGPT into our browser. This integration empowers users to input commands in intuitive natural language rather than relying on structured instructions.

Activate this feature by toggling the “Use ChatGPT” switch (**Fig 2.6i**). Upon activation, a new set of example buttons emerges (**Fig 2.6i**), presenting layout instructions articulated in natural language. For instance, the "Gene mCH" example transforms to: “Make a scatter plot for the

Gad1 gene body mCH”. The “Multi-2D Higlass” example evolves into: “Show me a multi 2D HiGlass browser for three cell types: CA3 Glut, Sst Gaba, and STR D2 Gaba, centered on the *Nfia* gene locus”.

It's crucial to clarify that while we employ the ChatGPT API to translate user input into structured instructions, all visualizations, graphs, and browsers display genuine experimental data produced in this study, with no artificial generation from ChatGPT is involved.

Fig 2.6i Demonstration of steps to draw a graph by using ChatGPT.

Enhanced User Experience: Plot Synchronization and Cell Clipboard Function

To enrich the usability of our platform when working with multiple graph panels, we've integrated a couple of essential features:

Plot Synchronization: When users open multiple scatter plots based on the same coordinates, movements and zoom levels are synchronized across these plots. This offers a seamless experience, especially when simultaneously visualizing cell metadata and multiple gene plots on the same page.

Example Input-7 (scroll your mouse on the plot to zoom in):

R2.6j Synchronized Zoom in view shows *Cux2*'s RNA (middle plot) and mCH (right plot) level in a subset of cell subclasses (left plot).

Cell Clipboard Function: Recognizing the frequent need to delve deeper into specific cells or cell types in the genome browser, we've introduced the cell clipboard function. Whenever a user selects cells within a scatter plot, the relevant cell metadata is captured and saved to a 'cell card' in the clipboard. The clipboard can be toggled by clicking the “Open Cell Clipboard”

button in the user input area. This facilitates easy copying of the cell subclass name and the subsequent loading of associated genomic tracks in a HiGlass panel.

Fig. R2.6k Demonstration of cell clipboard function

Preserving Your Workspace: Save and Load Layout

Upon finalizing their desired layout, users can save their current configuration. Simply click on the "Download Current Layout Config" button, located at the top right. To resume work or view this layout later, users can upload the saved config file via the browser's config upload box on a new page.

Fig. R2.6I Demonstration of config layout function

Minor comments:

6) Suggestions to improve the clarity of text and data presentation are included in comments on the attached pdf file.

R2.7

We appreciate all the insightful suggestions and have made modifications to all the suggestions.

LINE 75: Somewhat contrary to the statement above: Cytosine DNA methylation (5mC) is a stable covalent modification that endures in post-mitotic cells for their entire lifespan

To avoid potential confusion, we refined the introduction and changed these two sentences to:

1. "Cytosine DNA methylation (5mC), a covalent modification found in post-mitotic cells throughout their lifespan, is associated with neuronal function, behavior, and various diseases."
2. "CpG and CpH methylation modulate transcription factor binding and gene transcription through dynamic occurrence at regulatory elements and gene bodies."

LINE 100-111: What is the meaning of methyl-diverse genes?

We changed methyl-diverse genes to "genes showing differential gene-body methylation patterns across brain regions".

LINE 143: How many were from the present study?

102,783 nuclei (34%) were originally featured in the previous publication. We also added this important information in the manuscript: “Collectively, we obtained 324,687 (301,626 passed QC) DNA methylome profiles, including 102,783 nuclei from previous research¹¹.”

Line 157: Were any new clusters identified relative to previous studies?

We identified 116 clusters in the previous paper and 4,673 clusters in this manuscript. More clusters are identified due to both the inclusion of more comprehensive brain regions and more rounds of iterative clustering to separate cells into the finest resolution of clusters. Among these 4,673 cell clusters, 1,799 included methylation nuclei from the previous publication. The rest of the 2,874 cell clusters were new in this manuscript and are shown to integrate well with the AIBS 10X RNA datasets, suggesting they are molecularly distinct (**Fig 2e**).

LINE 171: Were p-values adjusted for multiple testing?

We only did a single correlation calculation here to quantify the level of correlation for this particular example, the p-value is calculated with a permutation test without multi-testing adjustment.

LINE 192: Can you please specify in the text or figure legend that the numbers in Fig 2A refer to 31 cell types? Where/how were these cell classes specified? This will help to avoid confusion between the 4673 cell groups and the 31 named cell types.

Thanks to the reviewer for pointing this out. The 31 named cell classes were from our old version of annotation. To avoid confusion, we now change it into annotation at class level from AIBS taxonomy¹ to keep it consistent (**Fig 2a**).

LINE 309: Can you include a short definition of DMGs, as you did for DMRs above? From Fig. 3, it seems that DMG refers to genes with differential levels of mCH in the gene body. It would be helpful to state this in either the legend or main text (results).

We added the definition of DMG in the legend of Figure 3: “Spatial methylation patterns of DMGs (genes with differential mCH levels on gene body \pm 2kb among different brain regions)”.

LINE 328: meaning?

Arealization has been used to describe the cortical neurons forming functional areas. We added a citation in the manuscript.

LINE 334: Can you please refer to labeled features in the figure to help readers unfamiliar with the abbreviations given and/or brain anatomy?

We have modified the sentence to “For instance, laminar layer information was mapped among cortical excitatory cells (e.g. “L2/3 IT CTX Glut”, “L5 ET CTX Glut”, “L2/3 IT ENTI-PIR Glut”).”. This sentence specifically refers to the subclass labels shown in Fig. 3f. Additionally, we bolded these referred cell subclasses to highlight them in the figure panel.

LINE 391-392: Is there any 3C data from developing brains to test this idea?

We added two citations^{44,45}, which observed the remodeling of global chromatin conformation during early development in mouse granule cells and human hippocampus.

LINE 492-493: The data shown in Fig 5c does not seem related to this sentence.

To better facilitate readers to understand the Figure 5c, we added the DMR-Gene edge number to the title of the figure panel. Additionally, we labeled the gray dots between TSS and TTS as 27% edges connecting intragenic DMRs to genes.

LINE 566: Could you add a zoomed in view of a good example?

We added a zoom-in figure in Extended Data Fig. 12h and cited this figure in the manuscript.

LINE 599-600: This sentence is not very clear.

We modified the sentence to “**Note that these CpG clusters are created by grouping distal CpGs with similar methylation patterns across cell subclasses, making them distinct from traditionally described CpG islands.**”

LINE 611: What are "features" in this context? Is each Feature in Fig. 6c an exon?

We added in the legend for Fig. 6d (originally Fig. 6c) - “**For each gene, we used the exon, exon-flanking region, and intragenic DMRs as the mC features. The 3C features are all the intragenic highly variable interactions (Methods)**”

LINE 762: What was the source of this information? I can only see 4 colours (Glut, Gaba, Gly-Gaba, Dopa) in the neuro-transmitter (bottom) line. Presumably the other neuro-transmitters are so rare that they can't be seen in the plot. If that's the case, you may want to remove them from the colour legend, as its confusing to have colours in the legend that don't seem to be present in the plot, especially when the same colours are used in other sections of the plot (brain regions and cell subclasses).

We agree with the reviewer that some rare neurotransmitters were too faint on the bar plots. Thus, we removed them from the plots while keeping all neurotransmitter information in the metadata table (Supplementary Table 2). In the legend, we added, “**Rare neurotransmitters are not shown in the plot, but information is provided in Supplementary Table 2.**”.

LINE 763-764: If I understand correctly, all data sets are displayed in the same tSNE projection. That tSNE is shown 4 times, highlighting data points from each data set. Data points from other data sets are grey. The colour indicates the cell subclass that a data point was assigned to. The fact that the clusters are coloured in similar patterns between plots indicates that integration was successful - cell subclasses cluster together, even when data were obtained using different technologies.

That's the correct understanding of the plots. We added some explanation in the legend “**For each plot, the light gray cells in the background represent cells from the other three modalities.**”

LINE 769-770: It has taken me a long time to figure out that the tSNEs inset in panel f are related to subsets of the data. Please provide more explanation in the legend. Here's how I understand it: The top 2 plots show many (n=?) sub-clusters of cells falling within the MB-MY Glut-Sero group. The tSNE projection is the result of data integration. The left plot is coloured by methylome cell group and the right plot is coloured by transcriptome cluster. Part of my confusion came from the use of maroon and green dots to distinguish MB-MY Glut-Sero and L5 ET CTX Glut. Using similar colour scales to indicate different types of information on the same plot is confusing. Text may be more helpful.

We change the legend for Fig 2f to: “Integration t-SNE of “MB-MY Glut-Sero” or “L5 ET CTX Glut” cells from mC and RNA colored by intra-modality clusters and confusion matrix of overlap score between the intra-modality clusters”. To help readers comprehend this plot, we have replaced the colored dots with text and added the cell number to each floating panel.

LINE 772: The connection to Extended Data Fig 5 is difficult to understand because the tSNE projection differs from that shown in Fig 2f and the legend of Extended Data Figure 5 is very brief. Please add to the legend of Extended Figure 5 that the tSNE projection is based on the methylation-based clustering shown in Fig2A. It would also be helpful to state that the non-neuronal cell types were absent from the scRNA-seq data (if that's the case).

Thanks to the reviewer for this useful suggestion. In the legend of Extended Data Fig 5., we added “All t-SNE projection in this figure is based on the methylation-based clustering shown in Figure 2A. Gene expression of non-neuronal cell subclasses are not plotted here.”

LINE 774: It's not possible to discern differences in dot size, so I don't find this helpful. Typically size and colour would be used to convey different things (eg gene count, p-value in GO ORA). This plot would be easier to understand if only colour were used to indicate mCG/ATAC data. Or does dot size indicate the number of DMRs contributing to each cell-type specific regulatory element? +/- 500bp presumably means that you are displaying the average mCG over a region including 500bp flanks of the aggregated DMRs. Please specify or remove the text “+/- 500bp”.

We have improved Figure 2g by keeping dot sizes constant and using color to signify mCG Fraction and chromatin accessibility, with deeper color indicating these DMRs are more hypo-methylated or accessible in a subclass. Additionally, we replaced floating panels with a color bar for a more easily understandable figure.

LINE 806: What does the colour scale indicate? Darker colour = more interactions, right?

Yes, we are plotting the normalized chromatin contact matrices in Figure 4e, which means the darker the color, the more interactions there are.

LINE 838, 842, 845, 849: change all the color scale

Thanks to the reviewer for pointing these out. We carefully changed all the color scales in the figure.

LINE 851-852: The legend for this information is confusing. Should be: % cells with expression *little dot* 0, *big dot* 100

We rewrote the legend to better explain Fig. 5j: “Purple dots are colored by RNA CPM, sized by the percentage of cells in the subclass (column) with expression of the gene (row). Little dot means 0% and big dot means 100%.”

LINE 853: These subclass tSNEs could probably be removed. It isn't possible to interpret the colour code given the size of the image, so they add little information to the figure.

We changed Fig. 5k to t-SNE plots colored by the normalized PageRank Score of the six RFX family genes.

LINE 865: Please define this abbreviation in the legend.

We defined PSI in the legend as “Percent Spliced In” in the legend.

LINE 876: Check coloured bar to the left of these heatmaps. Should they be colored by subclass as in 6g?

We apologize for this mistake and have changed the colors to subclass colors.

LINE 879: Can you please clarify whether the scales for mCG fraction are 0-1 bottom to top, or top to bottom? It took me a while to figure this out, because you have previously indicated low mCG fractions with a darker green colour. Looking at this chart, I thought that the green bars were hypomethylated CGs and the white gaps were hypermethylated CGs (because gene bodies are typically hypermethylated). If you are using tall green bars for high mCG fraction, it is hard to tell the difference between low mCG fraction and absence of CG dinucleotides.

This panel shows a genome browser barplot of alpha and beta-Nrxn3 promoter regions. The color indicates modalities rather than quantitative values. This visualization is consistent with our other visualizations (e.g., Figure 1d). We have added "Genome Browser View" to the titles of these figure panels and also labeled the color range vertically to enhance clarity.

LINE 934: What is the reason for this? Are hypermethylated cells not present in the brain?

We apologize for this typo in this important information. We keep overall mCG level > 0.5, NOT "<", the global CG methylation levels for most cells range from 61.6% to 88.8%¹¹.

LINE 1602-1603: I would have expected an anti-correlation between mCG Fraction and ATAC CPM. Is there a mistake in the colour scale, or am I missing something? Can you explain which DMRs are in each mini heatmap? For example, the mini heatmap at the top right of the matrix:- Is this showing only DMRs between ABC_NN and VLNC_NN, or do all mini heatmaps show the same 1000 DMRs?- Does the green colour indicate the mC Fraction in ABC_NN or VLNC_NN? Or is it some kind of comparison/correlation between the two?

We apologize for the mistake in the color bar and reversed it for both Extended Data Fig 6c and 6d. In our manuscript, we have always used darker green to represent lower mCG methylation.

We modified the figure legend to clarify:

“Columns display hypo-DMRs of that cell subclass while rows show their mCG fraction/ATAC CPM values. Take the top-right mini heatmap as an example, rows represent VLNC_NN hypo-DMRs, with cell color indicating mCG fraction in ABC_NN.”

7) Statistical significance of Pearson Correlations is reported, but it is unclear whether the p-values have been adjusted for multiple tests.

R2.8

Thanks to the reviewer for pointing out the need for clarification on the adjustment of p-values for multiple tests. In the revised manuscript, we have corrected Line 171 (as the reviewer indicated in the attached PDF) to provide a clearer explanation of the statistical test performed. The PCC referred to at this line is calculated for individual examples as showcased in Fig. 1d. We employed a permutation test to compute the p-values, utilizing the `scipy.stats.pearsonr` function, where the p-value method is based on `PermutationMethod`.

For the remaining PCC calculations throughout the manuscript, we have adopted the Benjamini–Hochberg Method to adjust the p-values, effectively controlling the False Discovery Rate associated with multiple testing. We have undertaken a comprehensive review of the entire manuscript to ensure that our statistical methodologies, including the approach to p-value adjustment, are clearly and accurately presented.

8) A relevant recent study has not been cited: Herring CA*, Simmons RK*, Freytag S*, Poppe D*, Moffet JJD, Pflueger J, Buckberry S, Vargas-Landin BD, Clément O, Echeverría EG, Sutton GJ, Alvarez-Franco A, Hou R, Pflueger C, McDonald K, Polo JM, Forrest ARR, Nowak AK, Voineagu I, Martelotto L, Lister R# (2022) Human prefrontal cortex gene regulatory dynamics from gestation to adulthood at single-cell resolution. *Cell* 10.1016/j.cell.2022.09.039

R2.9

We have now cited this important reference in our introduction.

Reply to Referee #3:

In this manuscript, Ecker and colleagues presented a large and nearly complete cell atlas of mouse adult brain based on single-cell methylome sequencing, chromosome conformation capture, and in situ multiplexed RNA spatial mapping. The study was carefully designed in terms of sample collection, brain dissection, CCF registration, etc. Data generation was done on highly optimized experimental platforms, and the resulting data quality is high, representing the best that can be achieved by the current research community. The reported data set is unprecedented, in terms of data size and complexity. The authors went through very carefully quality checking and filtering, then created a set of innovative procedures for computational analysis. Even tasks like data storage, cell clustering, multi-omics integration require creative solutions for the data at this size. The authors then went on to perform a number of analyses, including cell type annotation, spatial registration, DMR calling, integration with spatial RNA data and chromatin accessibility data, characterization of chromosomal conformation differences and changes across cell types and their relationship with DNA methylation (especially neuronal mCH fraction), identification of gene regulatory networks and key transcription factors, examination of splicing isoform diversity and intragenic epigenetic heterogeneity. This led to many novel findings and insights. With such a huge multi-modal data set, there are definitely numerous other directions to explore in terms of computational analysis. I do think the authors have covered some more important and obvious ones and showcased the value and significance of this multi-omics atlas. Overall, I believe this is a landmark study that was very well designed and executed. There was a high level of rigor, also there were numerous innovative aspects in this study. I think it is appropriate for Nature.

R3.1

We are grateful to the reviewer for recognizing the importance and contributions of our manuscript, and for offering valuable insights to enhance our work. Per other reviewers' recommendations, we have developed a user-friendly web application (<https://mousebrain.salk.edu>) to facilitate public access to the multi-omic whole-brain dataset.

During this revision, we also aligned our cell annotations with the companion whole-brain scRNA-seq manuscript from the Allen Institute for Brain Science (AIBS)¹. This ensures consistent cell taxonomy across the Brain Initiative Cell Census Network (BICCN) consortium. Importantly, these updates do not affect our initial grouping of 4,673 cell groups and primarily involve renaming and minor regrouping at the subclass level. We use the updated taxonomy in our main manuscript and web application. Additionally, we integrated our work with the AIBS's whole-brain MERFISH dataset¹, expanding our spatial annotation of epigenome cells to encompass the entire brain.

The updated results and downloadable data files are available on our web application and the NeMO archive (<https://mousebrain.salk.edu/download>). We address each comment in detail below.

I do not have any major concerns but do have several questions that I hope the authors can clarify or address.

(1) Gene-body mCH in neuronal cells represents an attractive alternative for capturing gene expression, and it can even detect genes of low transcript abundance that sn-RNA-seq might

miss. My question is whether gene-body mCH abundance has its own bias. Is it more sensitive on longer genes (exons + introns)?

R3.2

Thank you for highlighting the potential bias in gene-body mCH abundance. To address this, we've expanded our analysis to investigate further the influence of gene length on the correlation between mCH and gene expression.

Methodology:

We calculated the Pearson Correlation Coefficient (PCC) between normalized gene body \pm 2Kb mCH fraction and RNA Count Per Million (CPM) across 212 neuronal cell subclasses, incorporating a permutation-based null distribution obtained by shuffling RNA and mCH values within each subclass. Genes were categorized into three groups: short (<2Kb, 3,805 genes), medium (2-100Kb, 23,879 genes), and long (>100Kb, 2,680 genes). We applied the Benjamini–Hochberg Method to adjust p-values from permutation tests and used an FDR < 0.05 to identify significant correlations.

Results:

After FDR filtering, the results revealed that longer genes exhibit a stronger negative correlation, with 72% of the >100Kb category showing significant correlations post-FDR adjustment (**Fig. R3.2a,b**). This underscores the efficacy of mCH in reflecting gene expression, especially in longer genes. However, we also observed significant correlations in 13% of short and 36% of medium-length genes (**Fig. R3.2a,b**), indicating that gene length is not the sole determinant of mCH-expression correlation.

A notable instance is the *Sst* gene, a short marker gene for the “Sst Gaba” cells, where significant correlation (PCC=-0.5, FDR= 1.2^{-8}) emerges only when including the \pm 2Kb flanking regions (**Fig. R3.2c**). This example showcases the necessity of including these regions when quantifying gene body methylation levels, as they likely house crucial regulatory elements. The advantage of being able to include regulatory flanking regions also enables the DNA methylome data to quantify the activity of short genes more accurately.

Additional mC quantification advantages:

Additionally, beyond measuring gene-level mCH fractions, the epigenetic dataset also extends to base resolution intragenic details. This granularity surpasses technologies like 10X scRNA-seq, constrained by 3'-bias and shallow coverage⁴⁶. An example of this advantage is in Figure 6, where the mCH dataset's depth unveils diverse intragenic epigenetic features indicative of isoform diversity in neuronal long genes, which is hard to observe with the 10X scRNA-seq data.

Fig R3.2. Analyzing the negative correlation between gene mCH and RNA. **a**, Density plot displays the PCC between the gene's mCH fractions and RNA expressions; Colored by different gene length groups. **b**, Number and % of genes showing significant positive or negative correlations in each gene category. **c**, Genome browser view of mCH (Blue) and RNA (Purple) level of the *Sst* gene and *Sst* gene \pm 2kb in five different cell subclasses (rows).

(2) TF bindings in specific cell types were inferred by searching for the TF binding motifs in DMRs. This is a very unique and innovative aspect of this study. Most other studies do not have single-cell methylome data, so differentially accessible regions (DARs) identified from scATAC-seq data were used for the search of TF binding footprints. Since there is a parallel scATAC-seq data set for the same mouse brain regions, can the authors do a direct comparison to assess the strength/weakness of using DMRs versus DARs from scATAC-seq data for TF binding detection?

R3.3

We express our appreciation to the reviewer for their insightful question. In our study, we identified 2.56 million non-overlapping CpG DMRs, providing insight into cell-type-specific cis-regulatory elements across various subclasses and major brain regions (Methods). Hypomethylation within these DMR regions typically signifies active regulatory status in adult brain

tissue, as depicted in **Fig. 2g**. A strong negative correlation between the mCG fraction and accessibility was observed for most identified DMRs (**Fig. 5a**).

Our approach to addressing the reviewer's query involved initiating an overlap analysis between the identified hypo-DMRs and ATAC peaks. We first categorize hypo-DMRs specific to cell subclasses, based on criteria where the average mCG fraction of the DMR fell below 0.3 of the robust mean of the cell subclass. The robust mean is defined as the average mCG fraction within the 25-75 percent range.

We then overlap these findings with the 1.3 million non-overlapping peaks from a parallel single-nucleus ATAC-seq (snATAC-seq) study³. The analysis revealed a significant overlap, with 51% of the hypo-DMRs corresponding with ATAC peaks and 93% of peaks aligning with DMRs (**Fig. R3.3a**). These findings underscore that a considerable portion of cCREs detected in snATAC-seq data can also be identified through methylation analysis. However, half of the DMRs defined in the methylation dataset do not exhibit overlap with ATAC peaks.

To delve deeper into the implications of these non-overlapping DMRs, we segmented the DMRs into five groups, based on their average accessibility across cell subclasses (**Fig R3.3b**). Each group, from I to V, displayed a progressive increase in accessibility.

The cell subclass specificity of each DMR group was examined by constructing a dendrogram of abundant cell subclasses utilizing either the accessibility or mCG fraction of each group (**Fig. R3.3c**). Cell subclasses included in this analysis contain more than 100 cells in both modalities. The mCG fraction revealed similar hierarchies across all groups, affirming the maintenance of cell-type specificity irrespective of accessibility levels (**Fig. R3.3c left**). In contrast, utilizing accessibility for this analysis only yielded clear cell-taxonomy organization in Groups IV and V (**Fig. R3.3c right**). Group I-III, with low accessibility signals, did not show meaningful cell type hierarchy. These results suggest that, although the relatively closed DMRs (Group I) have less chromatin accessibility information, the methylation level still preserves meaningful cell-type information and potentially represents some developmental vestigial information⁴⁷.

Furthermore, a motif enrichment analysis was undertaken for both Group I and V DMRs to discern their regulatory significance (**Fig R3.3d**). Compared to the genome background regions, both the Group I and V DMRs are enriched for motifs of different transcription factors. For instance, Group I DMRs exhibited enrichment for the myocyte enhancer factor-2 (Mef2) family, implicating their role in mouse fetal development⁴⁸. Group V DMRs, characterized by open chromatin, were enriched for CTCF, suggesting a more active involvement in chromatin conformation regulation of these regions.

In summary, our study not only corroborates the identification of most cCREs detectable by snATAC-seq but also illuminates the existence of cell-subclass specific DMRs of regulatory significance that are not apparent in accessibility information. These "closed DMRs" preserve cell-type-specific methylation patterns and potentially reveal interesting development information⁴⁷, which is not apparent in accessibility information.

Fig R3.3 Overlap between DMR and ATAC peaks | **a**, A Venn diagram shows the overlapping of cell-type specific hypo-DMRs and ATAC peaks. **b**, DMR Groups by mean accessibility level across brain cell subclasses. **c**, Major cell subclass dendrogram reconstructed with the accessibility profile or mCG fractions of each DMR group. Leafnode color by the cell subclasses. **d**, Motif enrichment was done by findMotifsGenome.pl function in Homer. Scatterplot shows motif enrichment in closed and open DMRs; Dot color exhibits the fold change of Target Sequences/Background Sequences with motif. Dot size shows $-\text{Log}(p\text{-value})$ of top enriched motifs in example cell subclasses.

(3) Another strength of this data set is the chromosome conformation map, which allows the authors to link regulatory regions with transcriptional units at a higher level of confidence. In many other studies where chromosome conformation information is not available, a more naïve approach of linking regulatory regions and genes based on the linear chromosomal

distance was used. Can the authors make a comment on how many more confident linkages can be made by incorporating chromosome conformation information?

R3.4

We appreciate the reviewer highlighting the value of the chromosome conformation map. This modality indeed facilitates a more confident linking of cis-regulatory elements (CREs) to their target genes, overcoming the limitations inherent in methods that rely solely on linear chromosomal distance.

As the reviewer points out, many studies are constrained to the assumption that genes and CREs located closer in linear distance are more likely to be functionally connected. However, in our analysis (**Figure 5**), a mere 25% of the TF-DMR-Gene edges link DMRs to their nearest genes based on linear distance. The majority, instead, establish connections not predicated on proximity, underscoring the limitations of the naïve distance-based method.

The intricate folding patterns of DNA molecules within the cell nucleus, characterized by features like chromosomal compartments, topologically associated domains, and chromatin loops, facilitate distal interactions that exceed expectations based on linear distance. We quantified these hyper-interactive linkages by first establishing the background contact strength for each genomic distance. This was achieved by calculating the mean and standard deviation from the diagonal values of normalized chromatin contact strength at a 10 kb resolution (**Fig. R3.4a-d**). Next, a z-test was employed to identify DMR-gene pairs in Figure 5 that exhibited interaction levels significantly above this background within specific cell types' 3C modality. After applying the Benjamini-Hochberg method, we classified linkages with a false discovery rate of less than 0.05 as significantly highly interactive (**Fig. R3.4a-d, colored dots**).

Our findings revealed $179,925 \pm 11,133$ (mean \pm std) highly interactive linkages among the 1.6M total edges (**Fig. R3.4e**). We also observed a low overlap ($42\% \pm 10\%$) of significant linkages between any given pair of cell subclasses (**Fig. R3.4f**), underscoring the cell-type-specific insights provided by chromatin conformation data. This method, therefore, offers a critical supplement to linear distance-based approaches, unveiling cell-type-specific interactions that would otherwise remain obscured.

In conclusion, the incorporation of chromosome conformation data allows for the identification of a significantly higher number of confident CRE-gene linkages and provides essential cell-type-specific information. This modality, therefore, provides an indispensable role compared to the naïve linear distance-based methods in enhancing our understanding of the link between regulatory regions and target genes.

Fig. R3.4 Comparison between DMR-Gene links identified by linear chromosomal distance and chromatin conformation | a-d, Each dot represents a DMR-gene linkage. Red dots indicate statistically significant DMR-gene linkages within that subclass, while gray dots represent insignificant linkages. Scatterplot displays the genome distance between linkages (x-axis) and normalized contact strength (y-axis) in CA3 Glut (**a**), L2/3 IT CTX Glut (**b**), MY Lhx1 Gly-Gaba (**c**) and STR D2 Gaba (**d**). The blue line represents the mean and light gray area represents standard deviation of normalized contact strength at each genomic distance. **e**, Histogram shows the distribution of the number of significant linkages in each cell subclass. **f**, Histogram depicts the distribution of significant linkages overlapping ratios between all subclass-subclass pairs.

(4) For the spatial mapping of single-cell methylomes using MERFISH (LINE 323-325), is there a confidence score for the spatial assignment of each cell or cell type (or sub-class)? If so, what cutoff was used for retaining the confident assignments?

R3.5

We appreciate the reviewer's question for evaluating confidence in imputed spatial coordinates. In the revised manuscript, we extend the mC-MERFISH integration analysis using a comprehensive whole-brain MERFISH dataset from the AIBS companion study¹. This dataset comprises 51 coronal slices of a male C57BL/6J mouse brain and includes 3,934,605 cells profiled for 500 genes. These cells are spatially registered to the mouse adult brain common coordinate framework (CCF)⁴ and annotated with detailed CCF anatomical structures based on this registration. Notably, the MERFISH dataset also shares consistent cell

taxonomy with our methylome dataset, as both are integrated with the AIBS's scRNA-seq dataset.

The integration approach is akin to the snmC-MERFISH integration section in our manuscript. We grouped snmC-seq data by coronal slices and integrated each with corresponding MERFISH data using the 500 profiled genes. Post-integration, we calculated the closest MERFISH neighbor for each methylation cell based on shared subclass labels and Euclidean distance in the integrated low-dimensional space.

To assess the uncertainty of the assigned mC nuclei spatial coordinates, we examined the distances among the five nearest MERFISH neighbors of each methylome nucleus by calculating their average distance to the center. In simpler terms, if the five neighbors were closer to each other on the MERFISH slice, it bolstered our confidence in the assigned spatial coordinates (in global microns) for the mC nuclei. Our findings revealed that neuronal cells exhibit a smaller neighboring distance between nuclei compared to non-neuronal cells (**Fig R3.5a**), exhibiting a more clustered distribution on the MERFISH slice (**Fig R3.5b**). Despite cell subclasses, we also notice that larger brain regions show larger average neighbor distance in the positioning of mC nuclei (**Fig R3.5c-d**). Therefore, establishing a strict cutoff point for determining confident assignments proves challenging, given the inherent variability in size and shape across diverse brain regions.

Fig R3.5. Confidence in mC nuclei spatial assignment | **a**, Violin plot shows the neighbor distance among neurons and non neurons. **b**, Imputed spatial locations for glutamatergic neurons, other neurons and non-neurons colored by cell subclasses. **c-d**, Imputed spatial locations of specific cell subclasses (**c**) and by dissection regions (**d**).

(5) I really like the part on GRN construction and the identification of key TFs. The strategy used here was quite innovative. On the other hand, this part ended with a long list of predictions with no information of confidence, and no validations. Ultimate validations would be to experimentally perturb the TFs or the regulatory elements in mice and examine the functional consequences. Obviously, this would require very significant efforts beyond the scope of this study. Is there any computational analysis that the authors can do to estimate the confidence of these predictions, maybe through some computational perturbation predictions, such as CellOracle?

R3.6a

We appreciate the reviewer's comment regarding the novelty of our approach. Our constructed final GRNs incorporate edges derived from three distinct analyses: (1) TF-DMR edges, (2) DMR-gene edges, and (3) TF-gene edges. The identification of each edge involved rigorous considerations to guarantee confidence and statistical significance, as described in the current manuscript. In the final assembly of our GRNs, an intersection strategy is applied to synthesize these diverse linkages (**Figure 5g**), with the intent of merging varied biological insights to refine the identification of key regulatory connections. We acknowledge the absence of a direct confidence assessment in this integrative phase, which requires extensive experimental perturbation. However, recent advancements in virus-based perturb-seq^{49–51} illuminate paths for future functional validations of our predictions. Our resource and analysis framework are instrumental in paving the way for these forthcoming high-throughput experimental endeavors.

Besides, we value the reviewer's perspective but must respectfully dissent on the applicability of computational perturbation approaches like CellOracle⁵² for validating or assessing the confidence of our GRNs. The in-silico perturbation analysis from CellOracle is essentially rooted from their TF-Gene GRN construction, which is in principle similar to our analysis, as delineated in **Figure R3.6a**.

CellOracle initiates with the formation of preliminary CRE-promoter links via a co-accessibility based approach⁵³. We find parallels with our initial steps, where the correlation between gene body mCH fraction and DMR mCG fraction crafts preliminary CRE-gene connections. Our approach, however, is further enriched by the incorporation of variable and correlated 3C interactions, offering an intricate layer of information unattainable through linear genome distance alone (as elucidated in response R3.4).

In the subsequent phase of TF assignment to each CRE-gene linkage, CellOracle's approach is through motif scanning, rooted exclusively in DNA sequence. We, contrastingly, adopt a multifaceted approach. We first used the pyCisTarget⁵⁴ package to perform motif scanning. We then added correlation analyses between TF mCH and DMR mCG fractions, showcasing the correspondence between epigenetic measurements and motif enrichments (Fig. 5f, Extended Data Fig. 11a-c). The combination of DNA sequence and modification diversity enhances the confidence and intricacy of TF-DMR relationships.

Next, our universal GRN (or the so-called Base-GRN in CellOracle) maintains the dynamic interplay among TF, DMR, and genes. In contrast, CellOracle's model, post motif scanning, retains only the TF-gene connections, obscuring the regulatory elements. Therefore, when introducing the cell-type-specific information, our model leverages both methylome and chromatin conformation information in the CRE and genes, offering an epigenetic view in contrast to CellOracle's reliance solely on gene expression data.

In essence, given the similar goal of constructing GRN and the correlative nature of different modalities, running a CellOracle analysis can hardly be considered as an independent assessment of our approach and results. Besides, after having the GRN structure, either PageRank algorithm or in-silico perturbation (modifying and propagating node weights) can be performed on both frameworks, which are not unique aspects for either methods. Nevertheless, our study generates new resources and proposed additional computational steps to potentially enrich other GRN frameworks like CellOracle. Eventually, all the GRN constructions underscore the need for more extensive and high-throughput experimental validations^{49–51}.

Fig. R3.6a, a step-by-step comparison of GRN construction frameworks between this study and Cell Oracle package⁵².

Also, many of the TFs/genes have been knocked out in mice in the past. What are the phenotypes? Can they lend support to some of the predictions?

R3.6b

To address the reviewer's question in a comprehensive manner, we turned to the International Mouse Phenotyping Consortium (IMPC) database (version 19.1, 2023-06-27)⁵⁵. This source offered insights into the phenotypes associated with 8483 total knockout genes, with 3,181 of them annotated as influencing "behavior/neurological phenotype" upon heterozygous or homozygous knockout.

We endeavored to annotate these phenotypic data with our TF-DMR-Gene linkages, as presented in Figure R3.6b. However, the endeavor yielded a marginal correlation (PCC=0.09), underscoring the challenge of directly linking individual transcriptional regulations to holistic animal phenotypes.

Despite this, there are isolated instances where pronounced phenotype overlaps align with our GRN. For instance, *Neurod2*, linked to *Efr3b* through five DMRs, shares seven phenotypes including "decreased locomotor activity" and "increased anxiety-related response." Similar overlaps are observed with *Neurod2*'s connections to *Strbp* and *Cpe*. Another example is the *Egr1* and *Junb* gene linkage, sharing five phenotypes including "abnormal forebrain, midbrain, and hindbrain development", showcasing the potential of regulatory linkages to reflect phenotypic realities.

However, the complexity of animal phenotypes and the absence of tissue-specific and temporal considerations in knockout effects are a huge challenge for us to use IMPC to evaluate detailed molecular regulations. This underscores the necessity for nuanced, targeted experimental validations to corroborate the insights offered by our GRN predictions.

Fig. R3.6b, Annotation of number of shared phenotypes between linked gene and TFs in the GRN. The phenotype information downloaded from IMPC database, where only the “behavior/neurological phenotype” terms are used.

References

1. Yao, Z. *et al.* A high-resolution transcriptomic and spatial atlas of cell types in the whole mouse brain. *bioRxiv* (2023) doi:10.1101/2023.03.06.531121.
2. Li, P. *et al.* Hemispheric asymmetry in the human brain and in Parkinson's disease is linked to divergent epigenetic patterns in neurons. *Genome Biol.* **21**, 61 (2020).
3. Zu, S. *et al.* Comprehensive single-cell analysis of chromatin accessibility in the adult mouse brain. *bioRxiv* (2023).
4. Wang, Q. *et al.* The Allen Mouse Brain Common Coordinate Framework: A 3D Reference Atlas. *Cell* **181**, 936–953.e20 (2020).
5. Luo, C. *et al.* Single nucleus multi-omics identifies human cortical cell regulatory genome diversity. *Cell Genomics* **2**, (2022).
6. Hie, B., Bryson, B. & Berger, B. Efficient integration of heterogeneous single-cell transcriptomes using Scanorama. *Nat. Biotechnol.* **37**, 685–691 (2019).
7. Korsunsky, I. *et al.* Fast, sensitive and accurate integration of single-cell data with Harmony. *Nat. Methods* **16**, 1289–1296 (2019).
8. Stuart, T., Srivastava, A., Lareau, C. & Satija, R. Multimodal single-cell chromatin analysis with Signac. *Cold Spring Harbor Laboratory* 2020.11.09.373613 (2020) doi:10.1101/2020.11.09.373613.
9. Welch, J. D. *et al.* Single-Cell Multi-omic Integration Compares and Contrasts Features of Brain Cell Identity. *Cell* **177**, 1873–1887.e17 (2019).
10. Gao, C. *et al.* Iterative Refinement of Cellular Identity from Single-Cell Data Using Online Learning. *bioRxiv* 2020.01.16.909861 (2020) doi:10.1101/2020.01.16.909861.
11. Liu, H. *et al.* 6-DNA methylation atlas of the mouse brain at single-cell resolution. *Nature* **598**, 120–128 (2021).
12. Hubert, L. & Arabie, P. Comparing partitions. *J. Classification* **2**, 193–218 (1985).
13. Luo, C. *et al.* Single-cell methylomes identify neuronal subtypes and regulatory elements in mammalian cortex. *Science* **357**, 600–604 (2017).

14. Liu, H. *et al.* Single-cell DNA Methylome and 3D Multi-omic Atlas of the Adult Mouse Brain. *bioRxiv* (2023) doi:10.1101/2023.04.16.536509.
15. Yao, Z. *et al.* A transcriptomic and epigenomic cell atlas of the mouse primary motor cortex. *Nature* **598**, 103–110 (2021).
16. La Manno, G. *et al.* Molecular architecture of the developing mouse brain. *Nature* **596**, 92–96 (2021).
17. Hodge, R. D. *et al.* Conserved cell types with divergent features in human versus mouse cortex. *Nature* **573**, 61–68 (2019).
18. Tian, W. *et al.* Epigenomic complexity of the human brain revealed by single-cell DNA methylomes and 3D genome structures. *bioRxiv* 2022.11.30.518285 (2022) doi:10.1101/2022.11.30.518285.
19. Noack, F. *et al.* Multimodal profiling of the transcriptional regulatory landscape of the developing mouse cortex identifies Neurog2 as a key epigenome remodeler. *Nat. Neurosci.* **25**, 154–167 (2022).
20. Goel, V. Y., Huseyin, M. K. & Hansen, A. S. Region Capture Micro-C reveals coalescence of enhancers and promoters into nested microcompartments. *Nat. Genet.* **55**, 1048–1056 (2023).
21. Wu, H., Zhang, J., Tan, L. & Xie, X. S. Extruding transcription elongation loops observed in high-resolution single-cell 3D genomes. *bioRxiv* 2023.02.18.529096 (2023) doi:10.1101/2023.02.18.529096.
22. Tarjan, D. R., Flavahan, W. A. & Bernstein, B. E. Epigenome editing strategies for the functional annotation of CTCF insulators. *Nat. Commun.* **10**, 4258 (2019).
23. Yao, Z. *et al.* A taxonomy of transcriptomic cell types across the isocortex and hippocampal formation. *Cell* (2021) doi:10.1016/j.cell.2021.04.021.
24. Zhang, M. *et al.* A molecularly defined and spatially resolved cell atlas of the whole mouse brain. *bioRxiv* 2023.03.06.531348 (2023) doi:10.1101/2023.03.06.531348.
25. Langlieb, J. *et al.* The cell type composition of the adult mouse brain revealed by single cell and spatial genomics. *bioRxiv* 2023.03.06.531307 (2023)

doi:10.1101/2023.03.06.531307.

26. Li, Y. E. *et al.* An atlas of gene regulatory elements in adult mouse cerebrum. *Nature* **598**, 129–136 (2021).
27. Herring, C. A. *et al.* Human prefrontal cortex gene regulatory dynamics from gestation to adulthood at single-cell resolution. *Cell* (2022) doi:10.1016/j.cell.2022.09.039.
28. Armand, E. J., Li, J., Xie, F., Luo, C. & Mukamel, E. A. Single-Cell Sequencing of Brain Cell Transcriptomes and Epigenomes. *Neuron* **109**, 11–26 (2021).
29. Lister, R. *et al.* Global epigenomic reconfiguration during mammalian brain development. *Science* **341**, 1237905 (2013).
30. Zoghbi, H. Y. & Beaudet, A. L. Epigenetics and Human Disease. *Cold Spring Harb. Perspect. Biol.* **8**, a019497 (2016).
31. He, Y. & Ecker, J. R. Non-CG Methylation in the Human Genome. *Annu. Rev. Genomics Hum. Genet.* **16**, 55–77 (2015).
32. Luo, C., Hajkova, P. & Ecker, J. R. Dynamic DNA methylation: In the right place at the right time. *Science* **361**, 1336–1340 (2018).
33. Guo, J. U. *et al.* Distribution, recognition and regulation of non-CpG methylation in the adult mammalian brain. *Nat. Neurosci.* **17**, 215–222 (2014).
34. Gabel, H. W. *et al.* Disruption of DNA-methylation-dependent long gene repression in Rett syndrome. *Nature* **522**, 89–93 (2015).
35. Lagger, S. *et al.* MeCP2 recognizes cytosine methylated tri-nucleotide and di-nucleotide sequences to tune transcription in the mammalian brain. *PLoS Genet.* **13**, e1006793 (2017).
36. Chen, L. *et al.* MeCP2 binds to non-CG methylated DNA as neurons mature, influencing transcription and the timing of onset for Rett syndrome. *Proc. Natl. Acad. Sci. U. S. A.* **112**, 5509–5514 (2015).
37. Tillotson, R. & Bird, A. The Molecular Basis of MeCP2 Function in the Brain. *J. Mol. Biol.* (2019) doi:10.1016/j.jmb.2019.10.004.
38. He, Y. *et al.* Spatiotemporal DNA methylome dynamics of the developing mouse

fetus. *Nature* **583**, 752–759 (2020).

39. Kim, S. & Wysocka, J. Deciphering the multi-scale, quantitative cis-regulatory code.

Mol. Cell **83**, 373–392 (2023).

40. Lee, D.-S. *et al.* Simultaneous profiling of 3D genome structure and DNA methylation in single human cells. *Nat. Methods* **16**, 999–1006 (2019).

41. Luo, C. *et al.* Robust single-cell DNA methylome profiling with snmC-seq2. *Nat. Commun.* **9**, 3824 (2018).

42. Südhof, T. C. Synaptic Neurexin Complexes: A Molecular Code for the Logic of Neural Circuits. *Cell* **171**, 745–769 (2017).

43. Kerpedjiev, P. *et al.* HiGlass: web-based visual exploration and analysis of genome interaction maps. *Genome Biol.* **19**, 125 (2018).

44. Tan, L. *et al.* Lifelong restructuring of 3D genome architecture in cerebellar granule cells. *Science* **381**, 1112–1119 (2023).

45. Heffel, M. G. *et al.* Epigenomic and chromosomal architectural reconfiguration in developing human frontal cortex and hippocampus. *bioRxiv* 2022.10.07.511350 (2022) doi:10.1101/2022.10.07.511350.

46. Ashton, J. M. *et al.* Comparative Analysis of Single-Cell RNA Sequencing Platforms and Methods. *J. Biomol. Tech.* **32**, (2021).

47. Hon, G. C. *et al.* Epigenetic memory at embryonic enhancers identified in DNA methylation maps from adult mouse tissues. *Nat. Genet.* **45**, 1198–1206 (2013).

48. Gorkin, D. U. *et al.* An atlas of dynamic chromatin landscapes in mouse fetal development. *Nature* **583**, 744–751 (2020).

49. Jin, X. *et al.* In vivo Perturb-Seq reveals neuronal and glial abnormalities associated with autism risk genes. *Science* **370**, 791525 (2020).

50. Santinha, A. J. *et al.* Transcriptional linkage analysis with in vivo AAV-Perturb-seq. *Nature* (2023) doi:10.1038/s41586-023-06570-y.

51. Zheng, X. *et al.* Massively parallel in vivo Perturb-seq reveals cell type-specific transcriptional networks in cortical development. *bioRxiv* 2023.09.18.558077 (2023)

doi:10.1101/2023.09.18.558077.

52. Kamimoto, K. *et al.* Dissecting cell identity via network inference and in silico gene perturbation. *Nature* (2023) doi:10.1038/s41586-022-05688-9.
53. Pliner, H. A. *et al.* Cicero Predicts cis-Regulatory DNA Interactions from Single-Cell Chromatin Accessibility Data. *Mol. Cell* **71**, 858–871.e8 (2018).
54. Bravo González-Blas, C. *et al.* SCENIC+: single-cell multiomic inference of enhancers and gene regulatory networks. *Nat. Methods* (2023) doi:10.1038/s41592-023-01938-4.
55. Dickinson, M. E. *et al.* High-throughput discovery of novel developmental phenotypes. *Nature* **537**, 508–514 (2016).

Reviewer Reports on the First Revision:

Referees' comments:

Referee #1 (Remarks to the Author):

The authors were very responsive to previous comments with substantial revisions. The integration of their dataset and other datasets in the collection of papers, such as Merfish dataset, is excellent. The creation of user friendly website of the large dataset for general access (with integration of ChatGPT) is also great. The manuscript is significantly improved.

One minor point:

The authors provide an explain on the limitation of their sample preparation to examine hemispheric asymmetry originally observed in the human brain. However, they have now integrated their dataset with the Merfish dataset. Can the authors identify any hemispheric asymmetry based on such integration?

Referee #2 (Remarks to the Author):

This revised manuscript has improved the presentation of an incredibly complex data set. The development of a user-friendly web interface will greatly increase the impact of this manuscript, and the illustration of alternatively spliced genes in Figure 6 is much more intuitive than the previous version.

All reviewer comments have been carefully considered and appropriately addressed.

I strongly support the immediate publication of this high-impact manuscript.

Dr Heather Lee

Referee #3 (Remarks to the Author):

The authors have significantly improved the quality and rigor of their manuscript in this revision. They have fully addressed the comments by me and (in my view) other reviewers. I also appreciate their effort on sharing their results on an interactive web portal. I'm fully supportive of publishing this manuscript on Nature.

Author Rebuttals to First Revision:

Response to reviewer comments:

Referee #1 (Remarks to the Author):

The authors were very responsive to previous comments with substantial revisions. The integration of their dataset and other datasets in the collection of papers, such as Merfish dataset, is excellent. The creation of user friendly website of the large dataset for general access (with integration of ChatGPT) is also great. The manuscript is significantly improved. One minor point:

The authors provide an explain on the limitation of their sample preparation to examine hemispheric asymmetry originally observed in the human brain. However, they have now integrated their dataset with the Merfish dataset. Can the authors identify any hemispheric asymmetry based on such integration?

We appreciate the insightful feedback from the reviewer. It is crucial to emphasize that the MERFISH slices are from thin sections measuring 10 μ m in thickness. This limited thickness complicates the precise alignment between the left and right coronal planes. Consequently, the observed transcriptome differences between the left and right hemispheres may result from both genuine hemispheric disparities and variations due to coronal plane rotation. This complexity further obscures the identification of hemispheric DMR (Differentially Methylated Region) signals. Given these challenges, we reasoned that assessing hemispheric asymmetry using the MERFISH data remains infeasible.

Referee #2 (Remarks to the Author):

This revised manuscript has improved the presentation of an incredibly complex data set. The development of a user-friendly web interface will greatly increase the impact of this manuscript, and the illustration of alternatively spliced genes in Figure 6 is much more intuitive than the previous version.

All reviewer comments have been carefully considered and appropriately addressed.

I strongly support the immediate publication of this high-impact manuscript.

Dr Heather Lee

We thank the reviewer for the positive feedback!

Referee #3 (Remarks to the Author):

The authors have significantly improved the quality and rigor of their manuscript in this revision. They have fully addressed the comments by me and (in my view) other reviewers. I also appreciate their effort on sharing their results on an interactive web portal. I'm fully supportive of publishing this manuscript on Nature.

We thank the reviewer for the positive feedback!